

Reconciling the total carbon budget for boreal forest wildfire emissions using airborne
observations
Katherine L. Hayden[1]*, Shao-Meng Li[2], John Liggio[1], Michael J. Wheeler[1], Jeremy J.B. Wentzell[1], Amy
Leithead[1], Peter Brickell[1], Richard L. Mittermeier[1], Zachary Oldham[1,6], Cris Mihele[1], Ralf M. Staebler[1],
Samar G. Moussa[1], Andrea Darlington[1], Alexandra Steffen[1], Mengistu Wolde[3], Daniel Thompson[4] , Jack

7        Chen[1], Debora Griffin[1], Ellen Eckert[1], Jenna C. Ditto[5], Megan He[5] and Drew R. Gentner[5]

[1]{Air Quality Research Division, Environment Canada, Toronto, ON, Canada}
[2]{College of Environmental Sciences and Engineering, Peking University, Beijing, China}
[3]{National Research Council of Canada, Ottawa, ON, Canada}
[4]{Canadian Forest Service, Natural Resources Canada, Edmonton, AB, Canada}
[5]{Yale University, New Haven, CT, USA}
[6]{University of Waterloo, Waterloo, ON, Canada}
*Correspondence to: Katherine Hayden (katherine.hayden@ec.gc.ca)
**Abstract**

18        Wildfire impacts on air quality and climate are expected to be exacerbated by climate change with the

most pronounced impacts in the boreal biome.  Despite the large geographic coverage, there is a lack of
information on boreal forest wildfire emissions, particularly for organic compounds, which are critical
inputs for air quality model predictions of downwind impacts.  In this study, airborne measurements of
250 compounds from 15 instruments, including 228 non-methane organics compounds (NMOG), were
used to provide the most detailed characterization, to date, of boreal forest wildfire emissions.  Highly
speciated measurements showed a large diversity of chemical classes highlighting the complexity of
emissions.  Using measurements of the total NMOG carbon (NMOG$_T$), the ΣNMOG was found to be 46.2
% of NMOG$_T$, of which, the intermediate- and semi-volatile organic compounds (I/SVOCs) were
estimated to account for 7.4 %.  These estimates of I/SVOC emission factors expand the volatility range
of NMOG typically reported.  Despite extensive speciation, a substantial portion of NMOG$_T$ remained
unidentified (46.4 %), with expected contributions from more highly-functionalized VOCs and I/SVOCs.



The emission factors derived in this study improve wildfire chemical speciation profiles and are
especially relevant for air quality modelling of boreal forest wildfires. These aircraft-derived emission
estimates were further linked with those derived from satellite observations demonstrating their combined
value in assessing variability in modelled emissions. These results contribute to the verification and
improvement of models that are essential for reliable predictions of near-source and downwind pollution
resulting from boreal forest wildfires.



## 1 Introduction

Wildfires play a natural role in maintaining forest health and diversity through the release of nutrients, seed germination, removal of aging vegetation, and reducing the spread of forest diseases. Wildfires are, however, one of the largest global sources of trace gases and aerosols to the atmosphere (Andreae, 2019; Yu et al., 2019) and can have deleterious impacts on human health (Cascio, 2018; Cherry and Haynes, 2017; Reid et al., 2016; Finlay et al., 2012), air quality (Landis et al., 2018; Miller et al., 2011; Rogers et al., 2020), ecosystems (Kou-Giesbrecht et al., 2019; Campos et al., 2019; Kallenborn et al., 2012; Johnstone et al., 2010) and climate (Randerson et al., 2006). Not only can wildfire pollutants fumigate local source areas, they can be transported over long distances resulting in degraded air quality in locations far from fire sources (Miller et al., 2011; Rogers et al., 2020), and pose threats to downwind ecosystems through wet and dry deposition processes (Kou-Giesbrecht et al., 2019; Kallenborn et al., 2012; Campos et al., 2019).

Wildfire impacts on air quality and climate are expected to be exacerbated by climate change (Bush and Lemmen, 2019; Seidl et al., 2017; Whitman et al., 2019) and such impacts are expected to be most pronounced in the boreal biome (Seidl et al., 2017; Whitman et al., 2019). The boreal forest zone is the most northerly of all forest biomes accounting for 1.2 billion ha of mostly coniferous forest and comprising about 30 % of the global forest area, or 11 % of the earth's land surface. On a global basis, boreal forest wildfires are responsible for an estimated 20 % of yearly global biomass burning emissions (van der Werf et al., 2006). Canada's boreal forests account for ~30 % of the global boreal zone area and encompasses 75 % of Canada's 347 million ha of forested land (Fig. S1). In the past decade, Canada has experienced unprecedented fire seasons, with large numbers of evacuations, major property damage, poor air quality and significant economic impacts (NRCan, 2018; Landis et al., 2018; McGee et al., 2015). Model predictions have suggested that Canadian fire occurrences will increase by 25 % by 2030 from a 1975 to 1990 baseline scenario (Wotton et al., 2010).

To adequately assess and mitigate the risks of wildfire emissions to human and ecosystem health, reliable pollutant predictions are required which depend on accurate and detailed fire emissions data.



Such emissions data are developed by multiplying emission factors and ratios with the mass of biomass
burned (Chen et al., 2019). In Canada, Environment and Climate Change Canada (ECCC) provides
predictions of particulate matter (PM) (<2.5 µm in diameter) from wildfire smoke to the public using the
FireWork modelling system that combines forecast meteorology, emissions inputs (e.g. emission factors),
forest fire and fuel data (e.g. fuel maps, plume height parameterization), and a regional air quality model,
GEM-MACH (details in Chen et al., 2019). FireWork is also used for air quality research studies with
significantly more complex chemical mechanisms for emissions characterization and detailed physical
processes. Wildfire field studies, as well as prescribed burns and laboratory work, have resulted in
valuable global databases of fire emission factors covering a broad range of ecosystems and geographic
areas (e.g. Andreae, 2019; Akagi et al., 2011), however, they are primarily concentrated on the temperate
forests of the American mid-west and savannah/grasslands of Africa (e.g. Andreae 2019; Permar et al.,
2021; Palm et al., 2020; Lindaas et al., 2020; Roberts et al., 2020; Juncosa-Calaharrano et al., 2021;
Coggon et al., 2019; Koss et al., 2018; Hatch et al., 2017). Due to a lack of emission data specific for
boreal wildfires, air quality models for northern regions face significant challenges resulting in uncertain
predictions of emissions, exposure and associated impacts.

In the summer 2018, a research aircraft was deployed to measure emissions and subsequent

transformation processes from an active boreal forest wildfire in western Canada (Fig. 1; Fig. S1). In this
paper, detailed emissions information is provided from an active, near-field boreal forest wildfire using a
detailed measurement suite of over 200 gas- and particle-phase compounds. Emissions of highly
speciated non-methane organic gases (NMOG) are characterized by broad chemical classes and
volatilities extending from VOCs to SVOCs. Speciated NMOGs, along with concurrent total NMOG
carbon ($NMOG_T$) measurements, provides a unique opportunity to reconcile the total carbon budget.
Emission factors are derived for all measured compounds resulting in more relevant emissions
information for boreal forest wildfires and improved emission quantification and chemical speciation
representations in air quality models. Combining aircraft-derived emissions with those from satellite
observations demonstrates usefulness to evaluate modelled emissions variability. The emissions





information in this work will contribute to verification and improvements of models that are essential for
reliable predictions of boreal forest wildfires pollutants.

**2 Methods**
**2.1 Aircraft measurements**

The NRC's Convair-580 research aircraft was deployed on June 25, 2018 to sample a wildfire

detected to the east of the Alberta/Saskatchewan border (56.4°N, 109.7°W) (Fig. 1).  Measurements of a
comprehensive suite of trace gases, particles and meteorology were made with high time resolution.
Meteorological measurements including relative humidity, temperature, wind direction and speed, as well
as aircraft state parameters such as altitude (masl) and geographic coordinates were conducted at 1 sec
intervals.  A detailed description of the various measurements methods with references is provided in the
supporting information (SI Sect. 2.1, Table S1), with only a brief description provided here.
**2.1.1 Trace gas measurements**  In-situ measurements of NO, $NO_2$, $NO_y$, $O_3$ and $SO_2$ were conducted
using commercial instruments (Thermo Scientific Inc.) modified to measure at 1 sec time resolution.
Ammonia ($NH_3$) measurements were made at 1 sec time resolution using a Los Gatos Research (LGR)
$NH_3$/$H_2S$ Analyzer, model 911-0039.  Calibrations were conducted periodically throughout the
measurement study using NIST–certified standards.  Instrument zeros were performed for all these
instruments 3-5 times per flight for a duration of ~3-5 minutes each time at the beginning, during and
after each flight.  Gas phase elemental Hg (GEM) was measured with a Tekran 237X instrument (Tekran
Instruments Corporation) modified to allow a reduced sampling time of 2 min (McLagan et al., 2021;
Cole et al., 2014).  CO, $CO_2$ and $CH_4$ were measured with a Cavity Ring Down spectroscopy instrument
(Picarro G2401-m).  A second Picarro G2401-m instrument was used to measure Total Carbon (TC, in
units of ppmC) by passing the sample air through a platinum catalyst (Shimadzu) which was placed at the
external rear-facing inlet assembly and maintained at 650 °C, adapted from Stockwell et al. (2018) and
Veres et al., (2010).  Total non-methane organic gases ($NMOG_T$), in mixing ratios units of ppmC, were





quantified by subtracting the ambient $CH_4$, CO and $CO_2$ measurements (instrument without the upstream
catalyst) from the TC measurements.
Individually speciated NMOGs (as well as some inorganic species) were measured with a
Chemical Ionization Mass Spectrometer (CIMS), a Proton Transfer Time-of-Flight Mass Spectrometer
(PTRMS), and through whole air sampling using canisters (Advanced Whole Air Sampler; AWAS). In
addition, integrated cartridge-based samples were taken. The CIMS (a modified Tofwerk/Aerodyne Api-
ToF) was operated using iodide as the reagent ion providing 1 sec time resolved measurements for 30
compounds (Table S2). The PTRMS (Ionicon Analytik GmbH, Austria) used chemical ionization with
$H_3O^+$ as the primary reagent ion providing 1 sec measurements for a suite of organic compounds. For
those compounds with no available gas standard, a relative response factor was calculated with reaction
rate constants using the method described in Sekimoto et al. (2017) and guided by the work of Koss et al.
(2018) ('calculated' compounds). Integrated 'grab' samples (20-30 sec) were collected from the aircraft
using the Advanced Whole Air Sampler (AWAS) with offline analysis. The AWAS provided speciated
measurements of hydrocarbons (<C10), but no oxygenates. Overlapping compounds/isomers that were
measured by both the PTRMS and AWAS, as well as between the PTRMS and CIMS were handled as
described in SI Sect. 2.1.4. Integrated gas phase samples were collected using an automated adsorbent
tube (i.e. cartridge) sampling assembly with offline analysis (Ditto et al., 2021; Sheu et al., 2018; Khare et
al., 2019). These samples provided targeted measurements of gas-phase compounds ranging in volatility
from $C_{10}$ volatile organic compounds (VOCs) to $C_{25}$ semivolatile organic compounds (SVOCs) including
hydrocarbons (CH), and functionalized compounds containing 1 oxygen atom ($CHO_1$), and 1 sulfur atom
($CHS_1$).
**2.1.2 Particle measurements**
Particle chemistry was obtained with a high resolution aerosol mass spectrometer (AMS)
(Aerodyne) providing mass concentrations of particle species including total organics (OA), $NO_3$, $SO_4$
and $NH_4$ for particles less than ~1 µm. Particle size distributions were measured between 60 and 1000
nm at 1 sec time resolution using the Ultra High Sensitivity Aerosol Spectrometer (UHSAS; Droplet





Measurement Technologies). Refractory black carbon (rBC) was measured using a single particle soot
photometer (SP2; Droplet Measurement Technologies).

**2.2 Flight and fire description**
A wildfire located near Lac La Loche in Saskatchewan (56.40°N 109.90°W) was detected by
satellite on June 23 (Fig. 1; Fig. S1). The fire was ignited by lightning on June 23, 2018 at 19:45 UTC
and lasted 50 hrs to June 25 21:41 UTC burning an estimated 10,000 ha before being extinguished by
rain. The area burned was mostly mature Jack pine and boreal spruce forest with a smaller fraction of
boreal mixed-wood forest. Satellite images from the VIIRS spectroradiometer on the Suomi NPP and
NOAA-20 satellites taken on June 25 showed merged fire hot spots with a visible smoke plume moving
in a north-westerly direction (Fig. 1; see SI Sect. 2.2 for more details).
Lagrangian flight tracks were flown downwind of the wildfire to follow the fire plumes. Multiple
horizontal transects, vertically stacked and perpendicular to the plume direction were made at different
altitudes from 640 to 1460 m asl (~220 – 1040 m agl, based on 420 m asl at Lac La Loche) forming
virtual screens. Five screens were completed over two flights with the closest screen ~10 km and the
farthest screen 164 km downwind of the fire, with the screens spaced such that the instruments sampled
the same air parcels as they were transported downwind. A vertical profile which typically reached
~2500 m asl was conducted in the plume at each screen to gather information on its vertical structure and
the height of the plume. As demonstrated by the elevated CO mixing ratios in Fig. 2, two distinct plumes
were identified - a south plume (SP) and north plume (NP), that were transported in parallel in a
northwesterly direction. The SP is estimated to be ~42 min old based on the measured wind speed at
Screen 1 and the distance from the closest edge of the VIIRS fire hot spots (~10 km). The NP is
estimated to be an additional 30 min older than the SP (further details in SI Sect. 2.2). For the purposes
of this investigation, only data from Screen 1 are used to characterize the direct emissions from this fire.
Evaluation of emissions of photolabile species could be influenced by photochemical and depositinal
losses that may take place between the time of emission and the time of measurement. However, at 10





km (<1 hr) away from the fire source, Screen 1 measurements represent some of the freshest emissions
ever measured under wildfire conditions. There are no other significant anthropogenic sources impacting
the Screen 1 measurements. Plume evolution during transport from Screen 1 to downwind Screens 2 to 5
is discussed in other papers (Liu et al., 2022; Ditto et al., 2021; McLagan et al., 2021).

**2.3 Emission ratios, emission factors and combustion efficiency**
**Emission ratios** Emission ratios were calculated using an integration method (e.g. Yokelson et al.,
2009) with the in-plume measurements for the SP and NP. The integration method was carried out by
first subtracting a background from the in-plume measurements. Background measurements were defined
as the average over short time segments (~30 sec) outside and at the same altitude as inside the plume,
and typically selected at the ends of the horizontal transects. The background-subtracted plume
measurements yielded enhanced plume values (e.g. $\Delta X(t)$) which were then integrated using the plume
start and end times guided by when CO mixing ratios were above the CO background. Nominal plume
time periods are indicated by the vertical grey bars in Fig. 3 which shows time series for CO, NMOG, OA
and acetonitrile for the first 4 of 5 transects on Screen 1. Integrated pollutant values were subsequently
normalized by the integrated values of CO (Eq. 1) to account for changes due to dilution producing
emission ratios (ER) for the SP and NP for each transect on Screen 1.

$$ER = \frac{\int_{start}^{end} \Delta X(t)\,(dt)}{\int_{start}^{end} \Delta CO(t)(dt)} \hspace{4cm} (1)$$

CO is known to be a suitable dilution tracer as it has a long atmospheric lifetime of 1-4 months (Seinfeld
and Pandis, 1998), is unreactive on the time scale of the measurements, and is a particularly good tracer
for smoldering fires (e.g. Simpson et al., 2011). In this study, ERs were calculated using CO as it was
well above background for the plumes measured, there were no other significant CO sources in the study
area, and co-varied well with the majority of measurements.





**Emission factors**  Emission factors (EFs) were determined as the mass of species X emitted per unit mass
of dry fuel burned in g kg⁻¹ assuming that all of the carbon in the fuel was released into the atmosphere
and measured (Ward and Radke, 1993; Yokelson et al., 2007), and that the mass fraction of carbon in the
fuel is constant.  EFs were determined using Eq. 2 where $F_c$ is the mass fraction of carbon in the fuel and
estimated to be 0.5 (de Groot et al., 2009 and references therein), $mm_x$ is the molar mass of the compound
of interest, and $mm_c$ is the molar mass of carbon, 12 g mol⁻¹, $\Delta X$ is the background-subtracted mixing
ratio or concentration of the species of interest, $\Delta TC$ is the background-subtracted total carbon.  Total
Carbon (TC) (see Sect. 2.1) was directly measured and includes all the carbon mass in $CO_2$, CO, $CH_4$, and
$NMOG_T$, as well as that from particulate black carbon (rBC) and particulate organic carbon (OC) (which
were added to the TC), for a complete accounting of all the emitted carbon.  For species measured in mass
concentration units, Eq. 2 was modified by converting TC to mass concentrations using the measured
temperature and pressure, and removing the molar mass ratio term.

$$EF\left(\frac{g}{kg\,f}\right) = F_c \; x \; 1000 \left(\frac{g}{kg}\right) x \; \frac{mm_X}{mm_C} \; x \; \frac{\Delta X}{\Delta TC}$$   (2)

EFs were determined for the SP and NP for each transect averaged to obtain screen-averaged EFs for the
SP and the NP, as well as for both plumes together.  There is a potential for inherent uncertainties with
this approach for calculating EFs and ERs as the ratios derived this way represent the average plume
composition and ignore the spatial heterogeneity in wildfire plumes (Liu et al., 2022; Decker et al., 2021;
Peng et al., 2020; Garofalo et al., 2019), chemical transformation processes, and can also be affected by
changing background levels.

**Combustion efficiency**  Combustion efficiency (CE) is a useful indicator of the relative proportion of
flaming vs smoldering stages of combustion which has a significant influence on the chemical
composition of the smoke (see SI Sect. 3.1 for further details).  Flaming fires have CE >0.90 (Yokelson et




al., 1996) and smoldering fires are typically ~0.8 with a range of 0.65 to 0.85 reported in the literature
(Akagi et al., 2011; Yokelson et al., 2003).  A modified combustion efficiency (MCE) is commonly
calculated assuming that $CO_2+CO$ adequately represents all of the fuel carbon that has been volatilized
and detected in ambient air.  Here, as the TC in the plume was directly measured, $\Delta TC$ was used in Eq. 3
to improve on the estimation of the CE by accounting for all the sources of carbon.  $\Delta CO_2$ and $\Delta TC$ in Eq.
3 are the integrated, background-subtracted mixing ratios.

$CE = \frac{\Delta CO_2}{\Delta TC}$                                                            (3)

**3 Results and Discussion**
**3.1 Fire combustion state**

The plume-averaged CE for the SP (transects 1 to 4) was 0.84±0.04 and for the NP (transects 1 to

3) 0.82±0.01.  Transect 4 was excluded from the calculations for the NP because only a portion of the
plume was detectable at this altitude (Fig. 3).  The derived CE indicates that the fire was predominantly in
a smoldering phase which is consistent with the satellite-derived fire intensities during the flight (see Fig.
10) and ground-based meteorological observations, and may reflect some residual smoldering combustion
(RSC).  It is estimated that emissions from this fire were sampled 14 hrs post flaming.  Other chemical
measurements from this flight also support that the fire was largely smoldering including the detection of
elevated $C_2H_4O_2^+$ (levoglucosan fragment from the AMS), low $NO_x$ levels (Lapina et al., 2008) (Fig. S2),
and no detectable $K^+$ (from the AMS) (Lee et al., 2010).  Significant spatial variability in the
concentrations of many of the measured species were observed closest to the fire source, while the plumes
became more well-mixed as they were transported downwind (Fig. S3).  This highlights the complexities
of assessing wildfire combustion processes (Ward and Radke, 1993), and in particular, boreal forests have
been observed to exhibit greater variability in combustion efficiencies than for other vegetation types
(Urbanski et al., 2009).



## 3.2 General plume features

Most pollutants were strongly concentrated in the fire plumes with the exception of several sulphur-containing compounds and a few other VOCs (Table S6). In Fig. 3, the in-plume portions are highlighted by the grey vertical bars and the SP and NP are indicated as the aircraft flew at increasing altitudes to complete five horizontal transects. The lowest 4 transects showed enhanced pollutant levels while the 5th transect (not shown) was predominantly above the height of the plumes. Higher concentrations were generally observed in the SP compared to the NP, possibly because of some plume dilution in the NP resulting from a change in wind direction prior to sampling. The SP and NP were distinctly separated from each other, with pollutants typically dropping to background levels between the plumes. $NMOG_T$ mixing ratios varied between 100 ppbv to near 10 ppmv in-plume. CO and acetonitrile, often used as tracers of biomass burning (e.g. Wiggins et al., 2021; Landis et al., 2018; Simpson et al., 2011; de Gouw et al., 2006), reached 6.6 ppmv and 20 ppbv, respectively in the SP, while maximum OA concentrations reached 276 µg m$^{-3}$, above a background level of ~9.5 µg m$^{-3}$. OA was the largest contributor to particulate mass (PM) comprising over 90 % of the measured submicron mass with remaining portion comprised of BC, $NO_3$, $NH_4$, and $SO_4$ (Fig. S4). Integrated filter samples taken from the aircraft across Screen 1 also showed the presence of a diverse set of functionalized particle-phase organic compounds (Ditto et al., 2021).

The most abundant reactive nitrogen compounds ($N_r$) were in the forms of reduced nitrogen (85 %) with $NH_3$ comprising 41.7 % of $\Sigma N_r$ (Fig. 4) and substantially lower nitrogen oxides i.e. $NO_x$ < 1 ppbv. A large portion of unmeasured nitrogen-containing compounds found in these plumes was likely dominated by peroxyacetyl nitrate (PAN) (Liu et al., 2022). These observations are consistent with emissions from smoldering fires (Burling et al., 2011; Goode et al., 2000; McMeeking et al., 2009; Yokelson et al., 1996). Dominant proportions of reduced nitrogen in biomass burning emissions were also reported previously (Lindaas et al., 2020; Burling et al., 2011; Yokelson et al., 1996). Alkyl nitrates have been identified in biomass burning emissions, but their contributions to total $N_r$ appeared to be small (Juncosa-Calahorrano





et al., 2021; Roberts et al., 2020; Lindaas et al., 2020; Simpson et al., 2011; Alvarado et al., 2010; Singh
et al., 2010).

**3.3 Total carbon budget**
**3.3.1 NMOG chemical classes – PTRMS, CIMS, AWAS**

In-plume mixing ratios and the relative contribution of individually measured NMOG species to

the sum of those species (ΣNMOG) are shown for 13 chemical classes in Fig. 5.  (See Fig. S5 for separate
SP and NP chemical classes).  The largest chemical classes include carbonyls (acids, aldehydes and
ketones), alcohols, hydrocarbons (alkanes, alkenes, alkynes), aromatics (including furans, phenol,
benzene and toluene), and nitriles.  Hydrocarbons (i.e. $C_xH_y$) are responsible for just over half of the
ΣNMOG (52.8 %) (Fig. S6), with 27.2 % identified as alkenes such as ethene, propadiene, and propene,
19.3 % alkanes, predominantly ethane, and 3.1 % alkynes, almost entirely acetylene.  Non-aromatic
oxygenates account for an additional 36.2 % of the ΣNMOG with roughly equal contributions (10.1 to
11.0 %) from acids, aldehydes and alcohols, and a smaller fraction from ketones (4.8 %).  Including other
oxygenated compounds such as furanoids and phenol/phenol derivatives, all oxygenates ($C_xH_yO_z$)
comprise 41.4 % (Fig. S6), of the ΣNMOG.

A similar range of compound classes has been observed in previous field and laboratory studies,

noting that the measured compound suite between studies varies to some extent.  For example, some
hydrocarbons, like 1-butene, ethane, propane, and isobutene measured in the present study were not
included in Koss et al., (2018) results.  Other studies have also found oxygenates to be a large portion of
NMOG emissions across multiple fuel types, including those similar to the current study, ranging from 51
– 68 % (Permar et al., 2021; Koss et al., 2018; Gilman et al., 2015; Akagi et al., 2011) with a range of 25
– 55 % reported in Hatch et al. (2017).   Comparisons between studies are influenced by differences in
study measurement suites and variations in fuel composition.  The fraction of NMOG oxygenates in the
present study (41.4 %) was closer to those reported in Hatch et al. (2017) when only the most relevant
fuel types of pine and spruce were considered (55 % and 43 %, respectively).  Similar to previous work





(Koss et al., 2018, Stockwell et al., 2015; Hatch et al., 2015), emissions of substituted oxygenates like
furanoids (furans+derivatives) and phenolic compounds were observed.  Furanoids contributed 4 % of the
ΣNMOG mostly due to furfural, furan and methyl furan while phenolic compounds eg. guaiacol, methyl
guaiacol, contributed 0.5 % of the ΣNMOG (Fig. S7).  Although their emissions were less abundant in the
present study, they represent important OH reactants (Coggon et al., 2019; Koss et al., 2018; Gilman et
al., 2015) with phenols being implicated as precursors to brown carbon formation in secondary organic
aerosol (SOA) (Palm et al., 2020).

Biogenic emissions of terpenoids including isoprene, monoterpenes, carvone, sesquiterpenes,

camphor/isomers and terpine-4-ol/cineole/isomers were elevated in the plumes collectively reaching 2.4
ppbv, and contributing ~1 % to the ΣNMOGs (Fig. S7).  Isoprene was ~70 % of these compounds with an
additional 29 % from monoterpenes.  Emissions of isoprene from biomass burning has been observed
from a wide range of fuel types (Hatch et al., 2019).  As isoprene is not stored by plants and the
measurements were taken ~14 hrs post flaming, it was likely emitted as a combustion product.

In this study, furfural was the most abundant oxygenated aromatic compound, whereas Hatch et

al. (2015) and Koss et al. (2018) found that phenol emissions were slightly larger than that of furfural for
all fuels tested.  As phenol emissions are associated with lignin pyrolysis (Stockwell et al., 2015;
Simoneit et al., 1999), the lower emissions in the current study could be because the lignin content in the
fuel mixture was lower than fuels used in previous studies or that most of the phenolic compounds were
emitted during the earlier phases of the fire.  Several modelling studies have indicated that aromatics and
terpenes are insufficient to explain SOA formation in biomass burning plumes (e.g. Hodshire et al., 2019)
suggesting the importance of inclusion of other aromatic species such as phenolics and furanoid
compounds.  However, models typically do not include reactions involving phenolic and furanoids
species, especially substituted compounds like furfural, guaiacol, and methyl guaiacol.  Box model
simulations have also shown that incorporation of OH oxidation of furan, 2-methyfuran, 2,5-
dimethylfuran, furfural, 5-methylfurfural, and guaiacol, leads to 10 % more $O_3$ formed (Coggon et al.,

2019).


### 3.3.2 Intermediate-volatility and semivolatile organic compounds (I/SVOCs)


Offline analysis of cartridge samples showed a wider range of hydrocarbons and functionalized
gas-phase organic compounds not observed in the PTRMS, CIMS, and AWAS measurements, including
I/SVOC compounds in the wildfire plume. ERs (Table S7) for species containing carbon, sulfur and
oxygen (i.e. CH (hydrocarbons), $CHS_1$ and $CHO_1$ type molecules) accounted for a sizeable fraction of
carbon in this range, with expected contributions from more highly functionalized organics in the gas (and
particle) phase not reflected in the CH, $CHO_1$, and $CHS_1$ compound classes (e.g., gas-phase species with
multiple oxygen atoms like vanillic acid or acetovanillone, and gas-phase species containing
combinations of oxygen and nitrogen atoms (CHON) (Ditto et al., 2021; 2022). ERs in the plume varied
across the carbon number range; in general, the highest ratios were observed for the complex mixture of
hydrocarbons (i.e. CH compounds) broadly peaking at $C_{20}$-$C_{25}$ in the SVOC range, with a larger
contribution from $C_{10}$ compounds including monoterpenes. By comparison, the complex mixture of
$CHO_1$ compounds was slightly lower in abundance than CH with contributions from $C_{10}$ monoterpenoid
emissions or oxidation products. $CHS_1$ IVOC-SVOCs were the lowest abundance species quantified.
$CHN_1$ compounds represent another observed contributor of IVOCs-SVOCs; the sum of all $CHN_1$
compound ion abundances was two orders of magnitude smaller than the sum of all $CHO_1$ species. We
note that for $CHN_1$, this qualitative comparison is in terms of ion abundances only, given a lack of
appropriate standards to calibrate for the complex mixture of reduced nitrogen I/SVOCs.
EFs were estimated for CH, $CHO_1$, and $CHS_1$ I/SVOCs based on Table S7 ERs (to CO) and the
average EF of CO ($115.7 \pm 7.5$ g kg$^{-1}$, Table A1). It was not possible to directly calculate EFs due to the
lack of a background sample upwind of the fire. EFs were estimated to be $1.6 \pm 0.04$ g kg$^{-1}$ for CH,
$0.9 \pm 0.09$ g kg$^{-1}$ for $CHO_1$, and $0.1 \pm 0.003$ g kg$^{-1}$ for $CHS_1$ species, for a total EF of $2.6 \pm 0.14$ g kg$^{-1}$ (Table
A1). These estimates accounted for $C_{11}$-$C_{25}$ species and focused on I/SVOCs to avoid double counting
the monoterpenes and $C_{10}$ monoterpenoid species. It is noted that the concentrations estimated for the
cartridge samples may be sensitive to variations in sampling efficiency within the under-wing sampling
pod across $C_{10}$-$C_{25}$ (Ditto et al., 2021). These emission estimates expanded the characterized spectrum of





organic species to include IVOC/SVOCs in boreal forest fire emissions, which until now, had only been
available from laboratory measurements (Hatch et al., 2018). However, the observed emissions of the
complex mixture of hydrocarbons and functionalized species may include contributions from the re-
volatilization of compounds previously emitted from upwind oil sands operations and deposited in the
forest ecosystem, as noted in Ditto et al. (2021).

**3.3.3 Accounting for the observed carbon**
Measurements of TC, along with the speciated measurements from the PTRMS, CIMS, AWAS
and cartridges, provided a unique opportunity to reconcile the TC budget in a wildfire. Fig. 6 shows the
TC partitioning based on derived EFs (Sect. 3.5); overlapping compounds from the individual
measurement methods were handled as described in SI Sect. 2.1.4. The total EF for all carbon-containing
compounds was 1651.5 g C kg$^{-1}$ and, as expected, $CO_2$ was the dominant contributor comprising >90 %
of TC. CO contributed 7.0 % followed by a contribution from $NMOG_T$ of 1.9 % with even smaller
contributions observed from $CH_4$ (0.5 %) followed by OC and BC (not shown) at <0.5 %. The magnified
pie chart (right side) displays the ΣNMOG EFs (for PTRMS+CIMS+AWAS measurements) totalling
14.4±3.2 g C kg$^{-1}$ which accounted for 46.2 % of the $NMOG_T$ EF of 31.2±3.8 g C kg$^{-1}$ (refer to Fig. S8 for
the individual SP and NP breakdowns). The cartridge data showed the presence of a range of larger
molecular weight I/SVOC compounds between $C_{10}$ and $C_{25}$ representing an additional 2.3±0.08 g C kg$^{-1}$
and 7.4 % of $NMOG_T$. Together, all of the speciated NMOG measurements in this study accounted for
53.6 % of $NMOG_T$. The remaining carbon mass was unidentified comprising 46.4 % of $NMOG_T$.
Despite using four state-of-the-art measurement techniques resulting in an extensive measurement suite,
almost half of $NMOG_T$ remained unidentified. This is consistent with previous work estimating ~50 % of
$NMOG_T$ by mass as unidentified (Akagi et al., 2011). It is noted, however, that the magnitude of the
unidentified portion is partly affected by uncertainties in the speciated measurements. For example, many
of the 'calculated' PTRMS compounds are uncertain by a factor of ~2 (SI Sect. 2.1.1, Table S1).
Nevertheless, a portion of the unidentified species likely consisted of challenging-to-measure-VOCs and



larger I/SVOCs that were highly functionalized or contained molecular features like reduced nitrogen
groups (e.g. amines) that have been observed in the gas and particle phase at various sites (Ditto et al.,
2020; Ditto et al., 2022).  While a complex mixture of I/SVOCs were observed from this fire (Table S7),
it is likely that other functionalized gas-phase species containing nitrogen and/or multiple oxygens (e.g.
$CHO_{>1}$, CHON, CHN) were also emitted, similar to particle-phase observations in the fire plume via
tandem MS in Ditto et al. (2021).  The presence of I/SVOCs in biomass burning emissions has been
previously observed in laboratory experiments (e.g. Koss et al., 2018; Hatch et al., 2018; Hatch et al.,
2017; Bruns et al., 2016) with smoldering more likely to emit a higher fraction of compounds with low
volatility than higher temperature processes (Koss et al., 2018).  Advancing analytical techniques to
expand the suite of NMOG speciation will enable further reconciliation of the TC budget which is
important for assessing secondary formation processes in the atmosphere.

**3.3.4 Volatility distribution of NMOG**

Volatility distributions can help track the full range of organic species to assess their partitioning

between the condensed and gas phases (Donahue et al., 2011).  Fig. 7 shows the fractional sum of all
NMOG EFs within each volatility bin in terms of saturation concentration ranges ($\log_{10}C_o$, µg m$^{-3}$).  $C_o$
values were estimated using the parameterization developed by Li et al. (2016).  NMOG emissions from
this fire spanned a large range of volatilities from $\log_{10}C_o$ of -2 to 10 µg m$^{-3}$ across SVOCs to VOCs
categories.  The bin-averaged O/C ratio based on the measurements increased with reduced volatility
reflecting the presence of compounds with additional oxygen-containing functional groups.  The highest
fraction of emissions was present as VOCs with 63.3 % having $\log_{10}C_o > 6$ µg m$^{-3}$, and 11.6 % as IVOCs
having $4 < \log_{10}C_o$ µg m$^{-3} < 6$ µg m$^{-3}$ and 7.9 % as SVOCs having $\log_{10}C_o < 3$ µg m$^{-3}$.  These results align
with laboratory studies showing that oxygenates comprised more than > 75 % of IVOCs across a range of
biomass types with IVOCs accounting for ~11 % of the ΣNMOG (Hatch et al.; 2018).  Fig. 7
encompasses the range of volatilities based on all the identified NMOGs in this study that is expected to
represent initial emission conditions for modelling downwind chemistry.  However, improved speciation,



particularly of lower volatility compounds, are needed to further expand the range of volatilities and
advance knowledge in gas to particle partitioning processes.

**3.4 Emission factors and comparisons with other studies**
Emission factors (EF) (and emission ratios (ER)) in this study are derived for 250 compounds
from 15 instruments of which 228 are NMOG species (Table A1).  This dataset represents the most
extensive range of field-based EFs ever determined for a wildfire in the boreal forest ecosystem.  In Fig. 8
average EFs are shown for compounds grouped by a) particles, b) gas-phase inorganics, and c) gas-phase
organics.  Separate EFs and ERs for the SP and NP are shown in the SI (Figs. S9 to S11).  In Fig. 9a-c,
EFs are compared with those from other relevant studies.  Fig. 9a shows a comparison with boreal forest
field measurements largely taken from a compilation by Andreae (2019) referred to as BFF19, as well as
values from Akagi et al. (2011) and Liu et al. (2017).  This results in a comparison for 50 compounds (35
organics and 15 inorganics/particulate species) with the largest suite of EFs from one study conducted in a
similar boreal region as the present study (Simpson et al., 2011).  EFs are also compared with laboratory-
derived EFs for lodgepole pine Koss et al. (2018; referred to as LAB18) (Fig. 9b), a dominant fuel in the
current study, with a total of 99 NMOGs and 3 inorganics in common.  In Fig. 9c, EFs are compared with
those recently reported in Permar et al. (2021) (referred to as TFF21) based on aircraft measurements of
temperate forest wildfires which provides the closest suitable comparison with similar speciated NMOGs
under wildfire conditions.  Comparisons include 111 NMOGs, and 4 inorganics/black carbon.  While the
Permar et al. (2021) study was conducted in a temperate forest region, it was at high elevation locations
with similar vegetation types as the current study.

**3.4.1 Particle species**  The $PM_1$ EF ($6.8\pm1.1$ g kg$^{-1}$) represents the total of all particle component species
as measured by the AMS.  OA has the largest EF, accounting for 90 % of $PM_1$, with comparatively lower
EFs for $pNO_3$, rBC, $pNH_4$, and $pSO_4$ (Fig. 8a, Fig. S4).  This reflects the dominant particle-phase organic
carbon content of the burned fuel and correspondingly lower fractions of nitrogen and sulphur-containing



compounds. Similar high organic fractions have been previously observed in biomass burning emissions
(Liu et al., 2017; May et al., 2014; Hecobian et al., 2011). ERs similarly highlight the dominant OA
emissions. The magnitude of EFs and ERs are generally similar between the SP and NP. EFs and ERs
for particle species derived in this study represent the first such measurements under boreal forest wildfire
conditions. In Fig. 9a, EFs for chemically speciated compounds are not found in BFF19, but when
compared with available values for U.S. temperate forest wildfires (Liu et al., 2017) are found to be lower
for OA, $SO_4$, $NO_3$ and $NH_4$ by factors of 2.7, 5, 5.3, and 3.1, respectively. The lower particulate
emissions in the present study may reflect differences in fuel elemental composition between temperate
and boreal forest ecosystems. Differences in fuel composition is inferred through comparisons of $NO_x$
and $SO_2$ emissions. For example, the average $NO_x$ and $SO_2$ EFs for boreal forests, are lower than the
average EFs for temperate forests by factors of 2.5 and 3.0, respectively. The lower $NO_x$ and $SO_2$
emissions from boreal vs temperate forest wildfires are likely reflective of the reduced S and N content in
boreal biomass (Bond-Lamberty et al., 2006) relative to conifer (Misel, 2012) fuels in the western U.S., as
well as the possible influence of lower anthropogenic sources of nitrogen and sulphur atmospheric
deposition in boreal forests (Jia et al., 2016). The $PM_1$ EF of $6.85\pm1.09$ g kg$^{-1}$ derived in the present study
is a factor of 2.8 lower than the $PM_{2.5}$ EF of $18.76\pm15.90$ g kg$^{-1}$ that is available for BFF19 (Fig. 9b). The
lower PM emissions in the present study, despite accounting for particle diameter differences (Sect.
2.1.2), is somewhat surprising given emissions of PM are typically higher from smoldering compared to
flaming fires (Liu et al., 2017; Akagi et al., 2012). However, there are few PM EFs for BFF19 (n=5) over
a limited range of MCEs (i.e. 0.89 to 0.93) showing significant variability. The $PM_1$ EF derived in the
present study falls within the range previously observed for boreal forest wildfires and underscores the
significant variability in PM emissions.

**3.4.2 Gas-phase inorganic species** The largest average EFs for inorganic gases (Fig. 8b) were from
reduced nitrogen compounds dominated by $NH_3$ ($0.63\pm0.19$ g kg$^{-1}$) and followed by HCN ($0.31\pm0.028$ g
kg$^{-1}$), with lower EFs for oxidized nitrogen compounds such as $NO_2$ ($0.11\pm0.037$ g kg$^{-1}$) and HONO





$(0.01\pm0.008$ g kg$^{-1}$).  This is consistent with previous work identifying elevated emissions of NH$_3$ and
HCN during smoldering conditions, whereas emissions of HONO and NO$_x$ are primarily associated with
flaming combustion (e.g. Roberts et al., 2020; Akagi et al., 2013; Yokelson et al., 1997; Griffith et al.,
1991).  The EFs for CO$_2$ and CO from the present study are very close to that previously reported for
BFF19 (Table A1).  However, EFs for most other gaseous inorganic species were lower than the BFF19
EF average including NH$_3$, HONO, SO$_2$ (n=2) and NO$_x$ (n=11), by factors of 3.9, 41, 4.7 and 14.9,
respectively (Fig. 9a).  There are only a limited number of studies reporting EFs for these compounds in
the BFF19 category.  For example, the HONO EF can only be compared with one other BFF19 study, but
is also lower compared to LAB18 (Fig. 9b).  There are also only 4 previously reported BFF19 EFs for
NH$_3$ ($2.46\pm1.75$ g kg$^{-1}$) showing a large range of values indicating a strong sensitivity towards factors like
fire intensity and chemical reactivity.  In contrast, EFs for HCN derived in the current study ($0.31\pm0.028$
g kg$^{-1}$) compare fairly well with BFF, LAB18 and TFF21, (Figs 9a, b, c, respectively) and do not vary
widely suggesting that HCN may be less sensitive to burning characteristics.  HCN is of concern due to
its impacts on human health particularly since biomass burning emissions are responsible for the majority
of the global HCN (Moussa et al., 2016 and references therein).

**3.4.3 Gas-phase organic species**  In Fig. 8c, the top 25 average EFs for gas-phase organic species are
shown in decreasing order of magnitude.  The most abundant emissions were from the lower molecular
weight compounds; such trends are generally in agreement with previous field-based measurements for a
range of fuel types (e.g. Permar et al., 2021; Andreae, 2019; Liu et al., 2017; Simpson et al., 2011;
Urbanski et al., 2009).  Excluding CH$_4$, the largest EFs were associated with methanol, followed by
ethene, ethane, acetic acid, C$_5$ oxo-carboxylic acids, acetaldehyde, formaldehyde, and acetone ranging
from $1.9\pm0.25$ g kg$^{-1}$ to $0.82\pm0.088$ g kg$^{-1}$ for these compounds.  Noting some variations related to
differences in measurement methods, other studies have identified many of these same species as
dominating biomass burning emissions (e.g. Permar et al., 2021; Simpson et al., 2011; Akagi et al., 2011).
For example, Simpson et al. (2011) found that 5 of the same compounds in the present study including



formaldehyde, methanol, ethene, ethane and acetone were in the top 10 NMOG EFs from aircraft-based
measurements made of boreal forest wildfires in northern Saskatchewan, Canada, and within ~300 km of
the current study. In the present study, the top 24 NMOG compounds accounted for just over half (57 %)
of the ΣNMOG by total molecular mass with lower lower emissions from the remaining measured
compounds. In western U.S. wildfires, small emissions from 151 species were found to account for
almost half of ΣNMOG (Permar et al., 2021).

To compare the total NMOG derived in the present study with those from previous studies that

typically sum up their speciated measurements i.e. ΣNMOG, estimates were made using two methods: 1.
increasing the ΣNMOG to account for the unidentified portion of $NMOG_T$; and 2. adjusting the $NMOG_T$
to reflect the total molecular mass (not just the carbon portion). For method 1, the ΣNMOG EF in this
study (25.8±3.2 g kg$^{-1}$) was increased by 46.4 % (Fig. 6) equalling 37.8 g kg$^{-1}$. This estimate assumes that
the carbon distribution is the same as the identified, speciated measurements. For method 2, based on the
speciated measurements, the average molecular mass was 100 g mol$^{-1}$ and the average carbon number was
6 resulting in ~28 % of the molecular fraction represented by atoms other than carbon. Adjusting the
$NMOG_T$ of 31.2±3.8 g C kg$^{-1}$ upwards by 28 % to reflect the additional molecular mass results in a
$NMOG_T$ of 39.9 g kg$^{-1}$. The resulting estimated $NMOG_T$ in this study of 37.8 to 39.9 g kg$^{-1}$ lies between
the estimated average of 58.7 g kg$^{-1}$ for the BFF19 (Fig. 9a) and those estimated from the ΣNMOG EFs of
25.0 g kg$^{-1}$ (LAB18) (Fig. 9b), and 26.1±6.9 g kg$^{-1}$ (TFF21) (Fig. 9c) derived from laboratory- and field-
based studies. In contrast to the current work, previous estimates of $NMOG_T$ are likely to underestimate
total NMOG emissions as they typically represent the sum of measured species only. Some studies have
attempted to account for $NMOG_T$ by including the sum of measured plus estimates of 'unknown' portions
of NMOGs (ΣNMOGs) (Permar et al., 2021; Koss et al., 2018; Stockwell et al., 2015; Gilman et al.,
2015). The BFF19 EF was recently doubled from 29.3±10.1 g kg$^{-1}$ to 58.7 g kg$^{-1}$ to account for
unidentified NMOGs where the ΣNMOGs were measured by FTIR, GC and PTRMS (Andreae, 2019;
Akagi et al., 2011). These results support that doubling the ΣNMOG provides a reasonable estimate the
$NMOG_T$. It is noted, however, that the average BFF19 NMOG EF is ~1.5 times higher than that derived





in the present study, however, this may reflect variability in NMOG emissions even within the same
boreal biome.

Although it is known that acidic compounds are emitted from biomass burning, few studies have

quantified their emission, particularly under field conditions (Andreae, 2019; Veres et al., 2010; Yokelson
et al., 2009; Goode et al.; 2000).  In this study, EFs for 31 organic acidic compounds were derived (Table
A1) representing the most detailed set of organic acid EFs from biomass burning for any ecosystem
(Andreae, 2019).  The largest EFs for these compounds include acetic acid, C5 oxo-carboxylic acids, C4
oxo-carboxylic acids, and pyruvic acid, all of which are found among the top 24 NMOGs (Fig. 8c).  For
those measurements that are available for comparison, EFs in the present study were lower for formic
acid and acetic acid, than in BFF19, and were also lower than in LAB18, and TFF21, ranging from factors
of 1.7 to 8.8 (Figs. 9c, d).  A total of nine organic acids that were in common with TFF21 and LAB18
(Table A1) have lower EFs, with the exception of pyruvic acid, which was substantially higher (> factor
of 37) in the present study.  Emissions for an additional 23 organic acids, as well as several inorganic
acids including nitrous acid, isocyanic acid, and peroxynitric acid, are included in Table A1.  These acids,
representing 10.3 % of the ΣNMOGs (Fig. 5), are an important class of oxygenates as they can form
additional PM (Reid et al., 2005) and influence the hygroscopicity of smoke particles (Rogers et al., 1991;
Kotchenruther and Hobbs, 1998).

Isoprene and monoterpenes, with similar EFs ~0.40±0.10 g kg$^{-1}$, represented 17[th] and 20[th],

respectively, of the top 24 NMOG EFs in this study.  Terpenes are known to be emitted from a range of
biomass burning fuels (Andreae, 2019 and references therein), but there have been few measurements in
boreal forest wildfire plumes (Simpson et al., 2011; Andreae, 2019).  It is noted that PTRMS
measurements of IVOCs like sesquiterpenes likely represent lower limits as they tend to be easily lost to
sample inlet lines due to their low volatility.  The isoprene EF of 0.41±0.10 g kg$^{-1}$ was more than a factor
of 5 higher, while the monoterpenes EF, 0.39±0.034 g kg$^{-1}$, was substantially lower than the only reported
EF for boreal forest wildfires (Simpson et al., 2011).  As the present study and the Simpson et al. (2011)
study were conducted in similar locations (i.e. boreal forest region within ~300 km of each other), with



similar average MCEs, and comparable background levels, these differences are likely driven by fire
stage sampled. The majority of monoterpenes are stored in plant tissues (resin stores) for long periods of
time, but isoprene is synthesized and immediately released by plants, and can also be emitted as a
combustion product (Ciccioli et al., 2014; Akagi et al., 2013). Hatch et al. (2019) found that a wide range
of terpenoids are released across a variety of biomass types with variable emissions that were dependent
on plant species, and specifically related to their fuel resin stores. In the present study, monoterpenes may
have 'boiled-off' through distillation processes in the early stages of the fire resulting in lower
monoterpenes emissions at the aircraft sampling time, ~14 hrs post-flaming. In contrast, the Simpson et
al. (2011) study sampled comparatively earlier and more intense fire stages where higher monoterpene
emissions were likely released from live or recently fallen trees that still contained significant resin stores.
The monoterpenes EF reported by Simpson et al. (2011) was likely even higher given only two
monoterpenes were speciated and emissions of other terpenes were likely (Hatch et al., 2019). Higher
isoprene emissions in the present study compared to Simpson et al. (2011) could be related to the
comparatively larger smoldering component. Although limited data exist on the release of isoprene as a
function of fire intensity, negative relationships between isoprene and MCE were observed in Australian
temperate forest fires (Guérette et al., 2018) and wheat fields (Kumar et al., 2018).
Several furanoid compounds also exhibited significant emissions (Fig. 8c) including furfural,
furan, and methyl furan ranking 12[th], 19[th], and 23[rd] of the top 24 NMOG compounds, respectively.
Emissions of furanoids have been observed for a wide range of fuel types (Hatch et al., 2017; Simpson et
al., 2011). Fairly good agreement was found with BFF19 for furfural, and furan (Fig 9a). The EFs for
furan ($0.39\pm0.028$ g kg$^{-1}$) and furfural ($0.65\pm0.08$ g kg$^{-1}$) were also similar to that in LAB18 (Fig. 9b), and
TFF21 (Fig. 9c), as well as other ecosystems (Andreae, 2019) suggesting their emissions were relatively
insensitive to fire intensity and fuel mixture. Overall, the comparisons in Fig. 9 indicate that for the
higher emitting species, the current results are fairly similar, but for the lower emitting species, these
results are lower than previous reported values.



**3.5 Evaluation of emissions models**

**3.5.1 Comparison of EFs with the model emissions speciation profile**

EFs derived in the present study are compared with those that are currently incorporated into the emissions component of the FireWork modelling system using the Forest Fire Emissions Prediction System (CFFEPS). CFFEPS uses EFs allocated for 3 combustion states (flaming, smoldering and residual) and for 8 species including lumped non-methane hydrocarbons (NMHC) based on United States vegetation data compiled in Urbanski et al. (2014) (Table 3 in Chen et al., 2019). Fig. 9d (bolded compounds) shows that the smoldering EFs in the present study were comparable for CO and $CH_4$, but lower for $PM_1$ ($PM_{2.5}$), $NH_3$, $SO_2$ and $NO_x$ by factors of 3.4, 2.4, 6.6 and 17, respectively. In the present study, additional mass between $PM_1$ and $PM_{25}$ accounted for only an additional 10 % of aerosol mass (SI Sect. 2.1.2). The lower EFs for these species implies that the CFFEPS EFs would not adequately capture their total emissions under smoldering conditions for the boreal fuel in the current measurement study.

For incorporation into numerical air quality models, total organic gas (TOG=NMOG+$CH_4$) emissions are typically split into detailed chemical components using chemical mass speciation profiles, and converted to lumped chemical mechanism species. In the FireWork modelling system, the smoldering combustion TOG is split into components based on EPA's SPECIATEv4.5 profile (#95428) (US EPA 2016, Urbanski et al.; 2014 - supplement Table A.2, Boreal Forest Duff/Organic soil). This profile is ultimately compiled using laboratory data from Yokelson et al. (2013), Bertschi et al. (2003), and Yokelson et al. (1997) based entirely on U.S fuel types. EFs in the present study were found to be generally lower than the laboratory-based EFs for 74 species in common ranging from factors of 1.7 to 8.5 including for monoterpenes, formic acid, phenol, furan and acetonitrile (Fig. 9d). The largest differences (factors of 49-57) were observed for sesquiterpenes, benzofuran, and naphthalene. A few species including furfural, propane nitrile and ethyl styrene are comparable, while isoprene, pyruvic acid, acetylene and cyclohexene are notably higher by factors 2 to 5.3.

For a research version of the FireWork system, the component speciation is mapped to the SAPRC-11 chemical mechanism species (Carter and Heo, 2013) with detailed oxygenated compounds


and aromatic species, largely to better represent SOA formation processes.  For comparison with the
measurement derived speciation profile in this study, EFs were first mapped to SAPRC-11 species and
then normalized to obtain mass fractions of relevant model mechanism species (Table S9).  Comparing
the normalized mass fractions for similar mechanism species (Fig. S12) showed much lower fractions of
reactive alkenes (ALK5) and aromatics (ARO2) and a slightly higher acetic acid group (CCOOH).  The
mass fraction of $CH_4$ is also different with 13 % of TOG in this study compared to 4 % from the SAPRC-
11 profile.  The measurement derived chemical speciation profile is expected to be slightly different from
the average speciation profile from EPA's SPECIATEv4.5 due to fuel type, chemical species
identification and mechanism mapping scheme.  The emissions profile developed in the present study is
considered a more representative smoldering emissions profile specific to the wildfire characterization for
the Canadian boreal forest fuel.

**596    3.5.2 Linking aircraft and satellite observations to evaluate modelled emissions diurnal variability**

Wildfires generally exhibit a diurnal cycle with fire intensities maximizing late afternoon and

diminishing at night having important implications for fire emissions.  Evaluating modelled emissions
throughout the diurnal cycle with observations is a critical step in verifying smoke predictions.  Emissions
models mostly parameterize diurnal fire emissions with prescribed profiles that distribute daily total
emissions to hourly.  In CFFEPs, a diurnal profile is applied to allocate daily burn area to hourly intervals,
with highest activity in the late afternoon.  The actual fuel consumed, and thus, hourly emissions, is then
calculated with depth of burn estimates driven by hourly meteorology (Chen et al., 2019).  In Fig. 10, for
the wildfire in the present study, the hourly CFFEPS-predicted emissions (orange dots) for selected
compounds are shown between 2018-06-24 17:00 UTC and 2018-06-25 21:00 UTC, spanning the aircraft
sample time (red arrow at 15:00 UTC).  The burning phases are outlined in the figure where flaming
(light pink background) is assumed to occur when the atmospheric conditions alongside fire behaviour
and emissions model outputs infer a fireline intensity >4,000 kW m$^{-1}$, and a smoldering fire (blue
background) for intensity <4000 kW m$^{-1}$.  The fire intensity distinction between flaming and smoldering





roughly aligns with the observed minimum for this particular fire with the fire radiative power (FRP, grey
dots) retrieval from the GOES-16 satellite sensor of 500 MW where smoldering occurs <500 MW and
flaming for >500 MW.  The 500 MW threshold over the approximately 1,700 ha of actively smoldering
area observed by overnight VIIRS thermal detections gives an estimated energy density of 0.29 MW ha$^{-1}$.
This FRP per unit area corresponds with observed FRP for flaming combustion of >0.4 MW ha$^{-1}$ from
lower intensity flaming fires by O'Brien et al. (2015).  The FRP represents the sum over all hotspots of
this fire for each 15-min observation period.  Emission rates in metric tonnes per hour (t h$^{-1}$) were derived
from selected aircraft measurements using a mass balance method (Gordon et al., 2015) and estimated to
be 29±2.1 t h$^{-1}$ for PM$_1$, 433±26.7 t h$^{-1}$ for CO, 0.65± 0.03 t h$^{-1}$ for NO$_x$ (as NO), and 2.7±0.16 t h$^{-1}$ for
NH$_3$ (red arrows).  Emission rates were also derived from satellite observations (black arrows) for CO,
NO$_x$, and NH$_3$.  Emissions of CO were estimated using a flux method as described in Stockwell et al.
(2021) using TROPOMI satellite observations yielding 1670±670 t h$^{-1}$ at 19:06 UTC and 4050±1620 t h$^{-1}$
at 20:48 UTC.  NO$_x$ emissions (9.1±3.4; scaled to t NO h$^{-1}$ at 19:06 UTC (not enough high-quality
observations for the 20:48 UTC overpass) were derived from the TROPOMI NO$_2$ dataset using an
Exponentially Modified Gaussian approach (Griffin et al., 2021).  NH$_3$ emission rates (5.6±3.9 t h$^{-1}$) were
derived from CRIS satellite observations at the satellite overpass time of 19:00 UTC by applying a flux
method (Adams et al., 2019).

The aircraft measurements were taken when the FRP was low reflecting a smoldering surface

fire.  However, the satellite overpass occurred ~4 hrs later than the aircraft measurements close to the
FRP daily maximum, after which rain passed through the area.  The CFFEPS model, exhibiting a
prescribed diurnal pattern, captures the increase in NO$_x$ and NH$_3$ emissions between that derived from the
aircraft and satellites transitioning from a smoldering to predominantly flaming fire; NO$_x$ emissions
increased by a factor >10, whereas the NH$_3$ emissions increased by a factor of approximately 2.  This is in
agreement with recent laboratory measurements that found that the release of NO$_x$ is favoured during the
flaming stage and the release of reduced forms of nitrogen, such as NH$_3$, is favoured during the
smoldering phase (Roberts et al., 2020) (also see Fig. 4).  However, the CFFEPS CO emission rates do





not track the increase in CO emissions between the aircraft-derived value and the two TROPOMI values,
indicating that the CO EF for flaming is low in the model. This highlights the need to validate model
emission rates with measurements to adjust and update the EFs accordingly.

Using the aircraft- and satellite-derived emission rates relative to FRP (in units of t h$^{-1}$ MW$^{-1}$) to

represent the the two end burning states ie. smoldering and flaming conditions, estimates of total
emissions from this fire were made for CO, NO$_x$ and NH$_3$. Total emissions were estimated by integrating
the GOES FRP over the period June 24, 2018 17:00 UTC to June 25 23:00 UTC, and applying the
derived smoldering and flaming emission ratios. It was assumed that flaming occured for FRP >500 MW
and smoldering for FRP < 500 MW. Emission rates were estimated with respect to the FRP for the
flaming and smoldering phases of the fire. The CO emission rates are roughly twice as large during
smoldering compared to flaming. For the satellite emission estimates from the two overpasses during the
flaming phase of the fire, the CO emission rates are very similar and well within the uncertainties (19:00
UTC ER$_{CO}$ = 4.7 t MW$^{-1}$; 2000 UTC ER$_{CO}$ = 43 t MW$^{-1}$). The ratio for NO$_x$ is also twice as large for
flaming compared to smoldering, and for NH$_3$, the ratio is ~5 times larger for smoldering than flaming.
Assuming that the fire went out when GOES did not observe any hot spots, total emissions for this fire of
CO, NO$_x$ and NH$_3$ are estimated at 21,808, 104.1, and 83.7 tonnes, respectively. If the fire is assumed to
have continued burning when GOES did not detect any fire hot spots (between 22:00 - 04:00 UTC and
07:00 - 15:00 UTC, with an FRP of 150 MW (~GOES detection limit; Roberts et al., 2015), the emissions
increase to 23,986, 106.4 and 97.7 tonnes, respectively, providing an upper limit of emissions. The
combination of aircraft and satellite-derived emission estimates for multiple species helps to obtain the
diurnal variability of emissions and to obtain more complete details on the emission information across
different burning stages.
**4. Summary and Implications**
This study provides detailed emissions information for boreal forest wildfires under a smoldering
combustion process. Highly speciated airborne measurements showed a large diversity of chemical
classes highlighting the complexity of emissions. Despite extensive speciation across a range of NMOG



volatilities, a substantial portion of $NMOG_T$ remained unidentified (46.4 %) and is expected to be
comprised of more highly functionalized VOCs and I/SVOCs. Although these compounds are
challenging to measure, their characterization is necessary to more fully understand particle-gas
partitioning processes related to the formation of SOA. Methodological advancements to achieve higher
time resolution speciated measurements of I/SIVOCs would move towards further $NMOG_T$ closure and
span a more complete range of volatilities. A detailed suite of EFs that were derived in this study can be
used to improve chemical speciation profiles that are relevant for air quality modelling of boreal forest
wildfires. Aircraft-derived emission estimates were paired with those from satellite observations
demonstrating their combined usefulness in assessing modelled emissions variability. As satellite
instrumentation and methodologies advance, linking emissions derived from aircraft (and ground)
observations for additional compounds will improve the ability to simulate and predict the diurnal
variation in wildfire emissions.

Although the measurements from this study provide a detailed characterization of a wildfire, the

results represent only one smoldering boreal forest wildfire. Additional measurements are needed under a
variety of fire conditions (combustion state, fire stage, biomass mixtures, time of day, etc) in order to
elucidate the major controlling factors and improve statistical representation for constraining and
modelling these sources. For example, measurements are needed to assess dark chemistry reactions in
biomass burning emissions which have been shown to be important in the formation of OA (Kodros et al.,
2020) and brown carbon (Palm et al.; 2020). In addition, reduced actinic flux associated with high
particle loadings in biomass burning emissions can influence plume chemistry (e.g. Juncosa-Calahorrano
et al., 2021; Parrington et al., 2013). The emissions information in this work will contribute to the
evaluation and improvements of models that are essential for reliable predictions of boreal forest wildfire
pollutants and their downwind chemistry.





**Acknowledgements**

The authors acknowledge the significant technical and scientific contributions towards the success of this study from the AQRD technical and data teams, the NRC team, and excellent program management by Stewart Cober. JCD, MH, and DRG acknowledge support from the National Science Foundation (AGS1764126) and GERSTEL for their collaboration with the thermal desorption unit used as part of this study, and MH also acknowledges the Goldwater Scholarship Foundation. S.-M.L. acknowledges the support of the Ministry of Science and Technology of China (Grant 2019YFC0214700).

**Author contribution**

KH, SML, JL, MJW, JJBW, AL, PB, RLM, CM, AS, RMS, SM, AD, and MW all contributed to the collection and analyses of the aircraft observations in the field. JCD, MH, and DRG analysed the cartridge samples. ZO contributed to the analyses and created many of the figures. DT contributed to the analyses of the physical and combustion state of the wildfire fire. DG and EE provided the satellite observations and DG wrote the satellite comparison section. JC contributed to the comparisons with the model emission speciation profile. KH wrote the paper with input from all co-authors.

**Competing interests**

The authors declare that they have no substantive conflicts of interest, but acknowledge that DRG and JL are associate editors with Atmospheric Chemistry and Physics.

**Data availability**

All data used in this publication are available upon request.





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



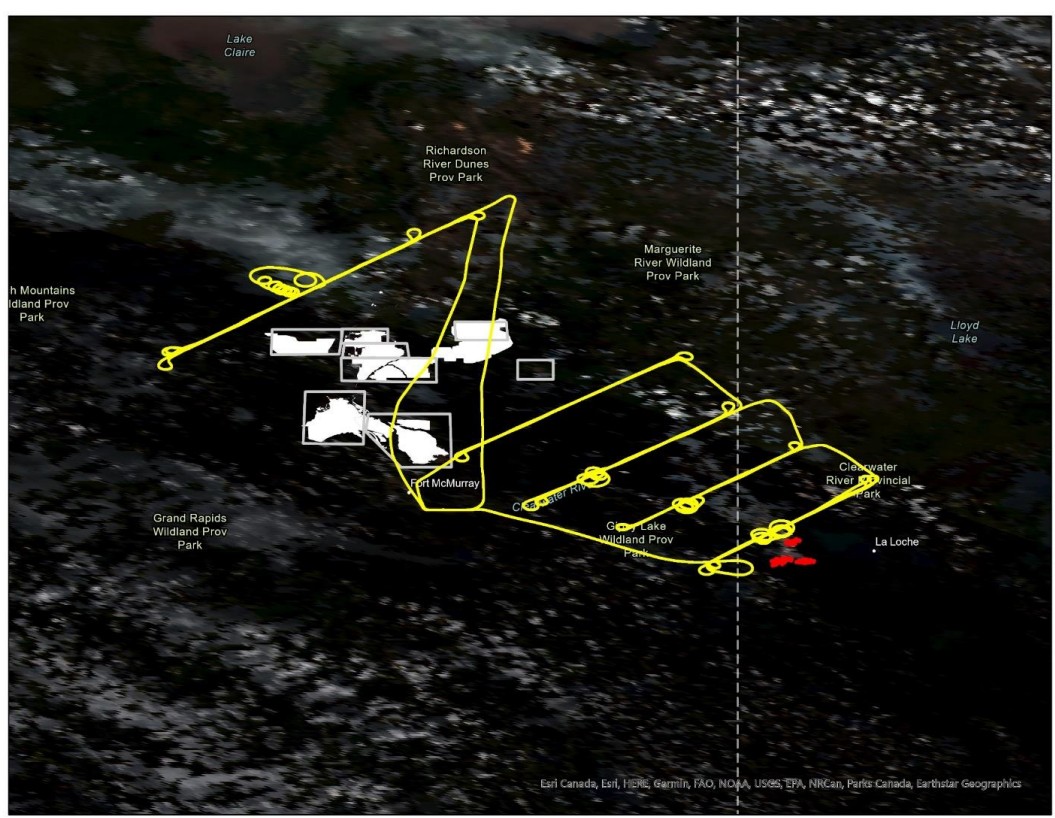

Figure 1. Corrected reflectance satellite image from the VIIRS spectroradiometer on the Suomi
NPP and NOAA-20 satellites taken on June 25, 2018. The fire hot spots for the wildfire of
interest are indicated by the red dots. Flight tracks were flow at Lagrangian distances downwind
of the wildfire. Multiple transects at varying altitudes perpendicular to the plume direction
formed virtual screens. Plume direction of travel is indicated by the large arrow. The location of
the Alberta oil sands mining facilities are shown in white.

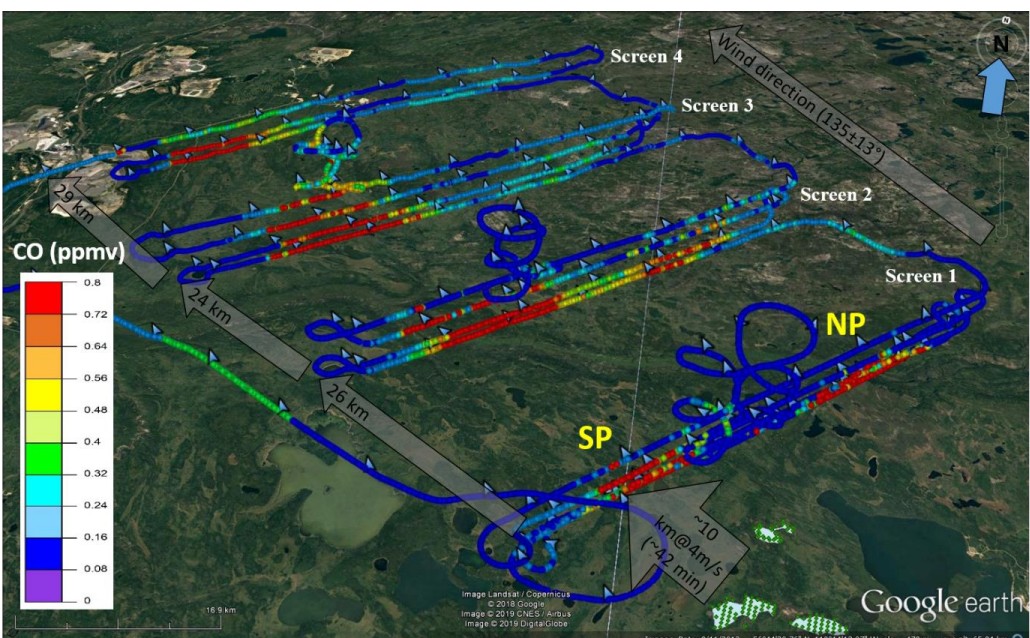

Figure 2. Flight tracks coloured by CO mixing ratio (ppmv) for Screens 1 to 4. The two plumes are identified as south plume (SP) and north plume (NP). The fire perimeter surrounding the detected MODIS-derived 'hot spots' on June 25, 2018 is shown in the green hatched area. The source of the NP is expected to be the same hot spots as the SP but ~ 30 min older; see SI Sect. 2.2. The small blue arrows along the flight tracks indicate the aircraft measured wind direction with the average wind direction depicted with the large gray arrow. Distances between screens are shown in the grey arrows.



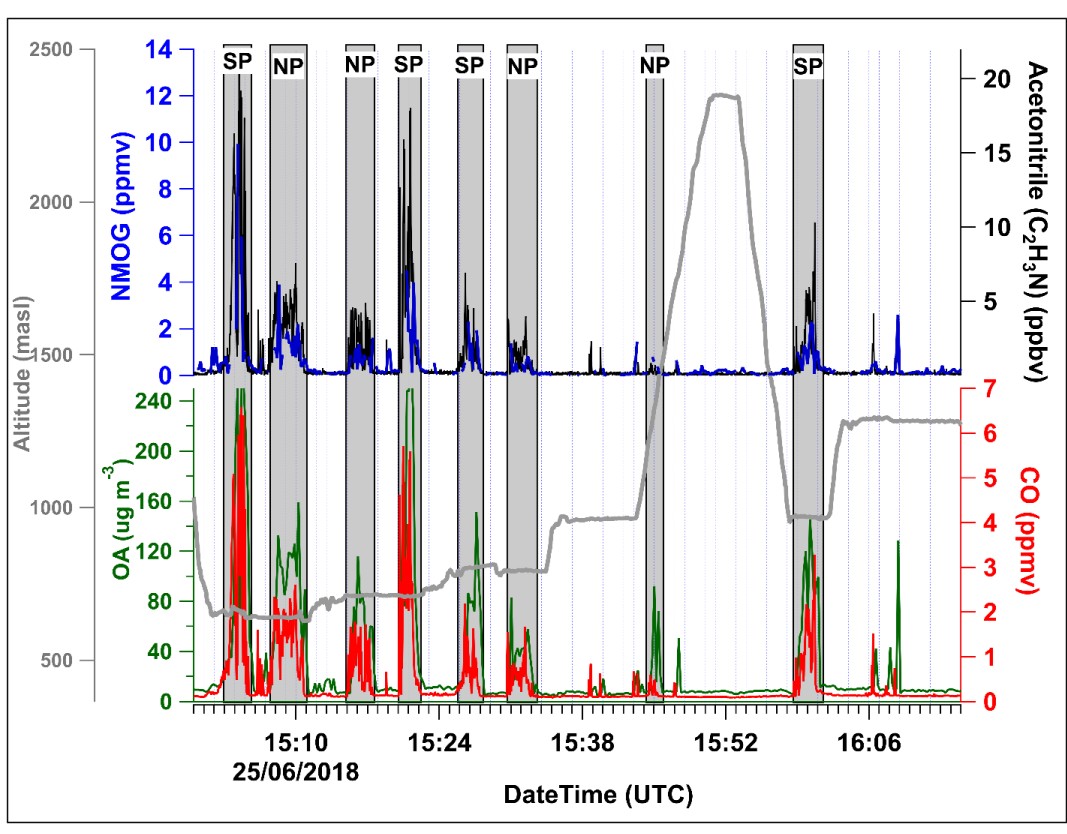

Figure 3.  Time series of NMOGs, acetonitrile ($C_2H_3N$) and CO mixing ratios, as well as OA
concentrations and altitude for Screen 1.  The in-plume portions are indicated by the vertical grey
bars.  The aircraft flew back and forth across the plumes at increasing altitudes to complete five
transects; a transect represents one pass across the SP and NP at the same altitude.



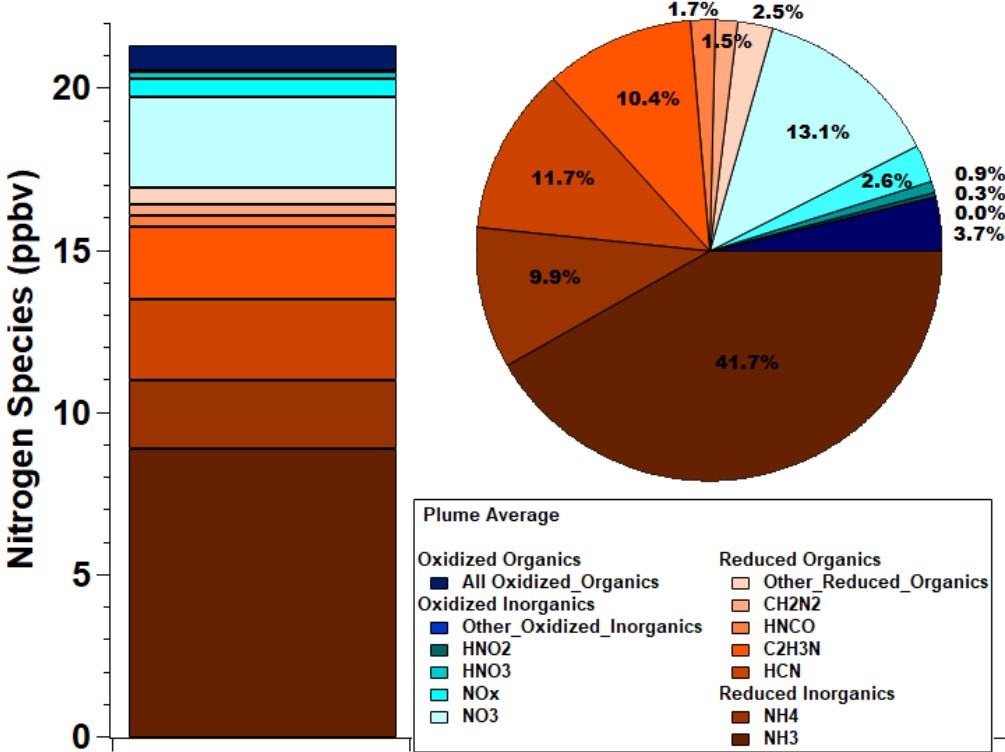

Figure 4.   Background-subtracted average Screen 1 in-plume mixing ratios of measured gas- and
particle-phase N-containing species ($N_r$) and their fractional contribution to the total summed $N_r$
species.  The $N_r$ species are grouped into categories of reduced inorganics, reduced organics,
oxidized inorganics and oxidized organics with reduced species in shades of red and oxidized
species in shades of blue.

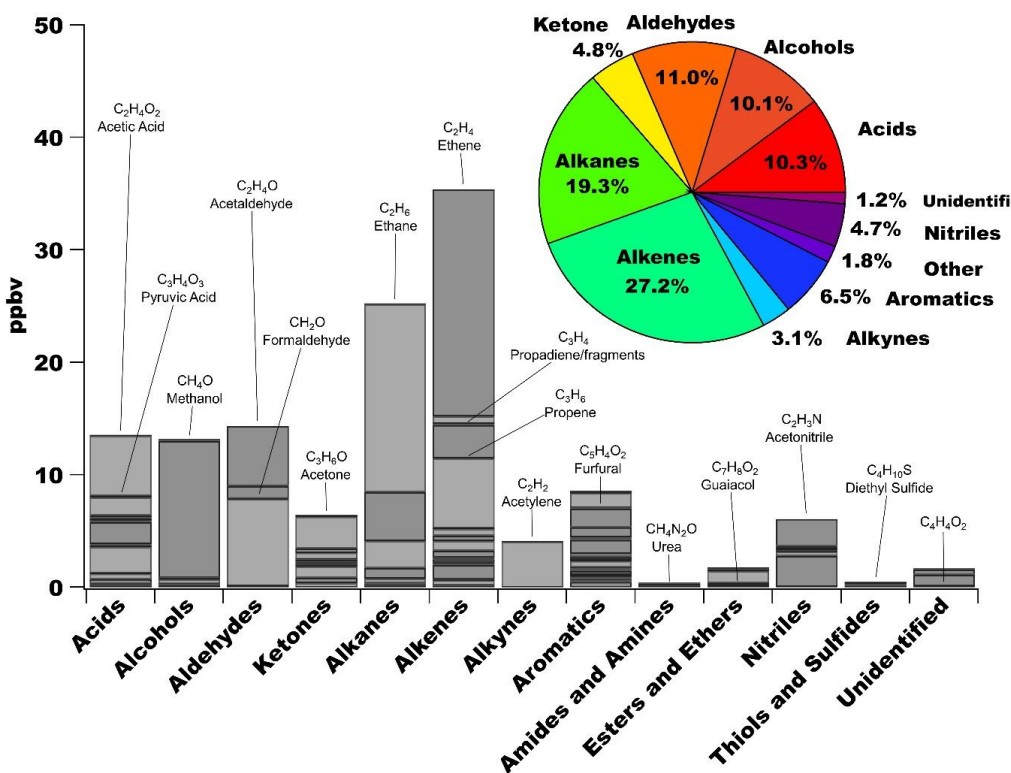

Figure 5. Background-subtracted average mixing ratios of individually measured NMOGs
shown for thirteen chemical classes. In some cases, compounds are double- (or triple-) counted if
they can be identified in more than one category. For example, phenol is an alcohol + an
aromatic; guaiacol is an alcohol + an ether + an aromatic. In the pie chart, the *Other* category
includes amides, amines, ethers, thiols and sulfides. The unidentified category contains
molecular formulas detected but the compound(s) could not be identified.

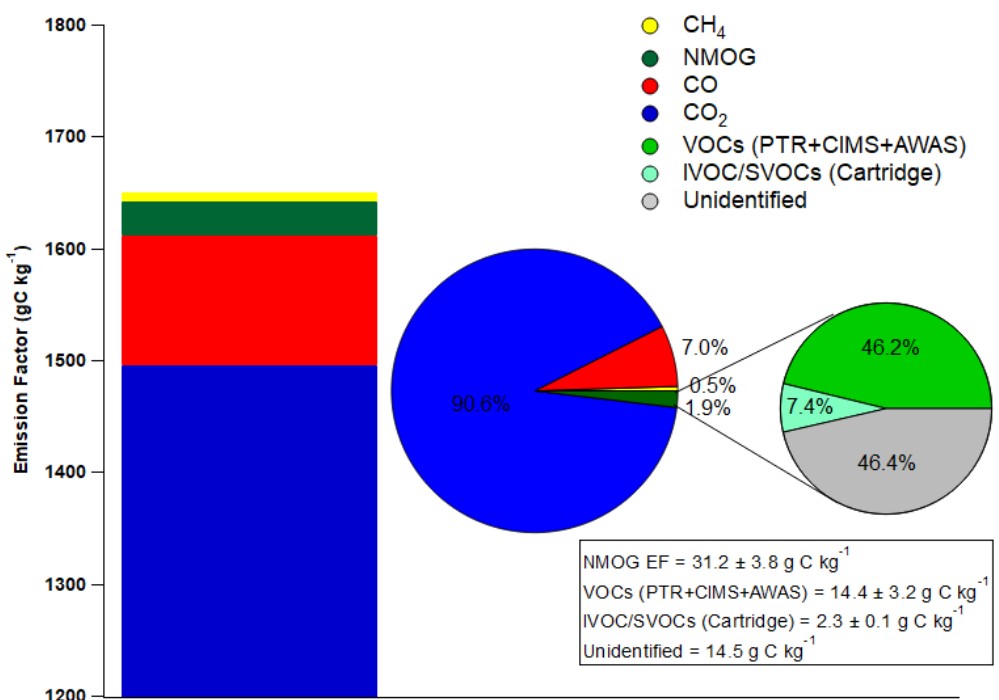

Figure 6. Total carbon (TC) partitioning based on EFs (carbon fraction). The bar chart shows
the stacked EFs for carbon-containing compounds with the middle pie chart showing their
percent contributions to the TC. The pie chart on the right show the percent breakdown of the
measured NMOGs with the remaining unidentified portion in terms of g C kg$^{-1}$. Note that all the
EFs shown in Table A1 were converted to g C kg$^{-1}$.

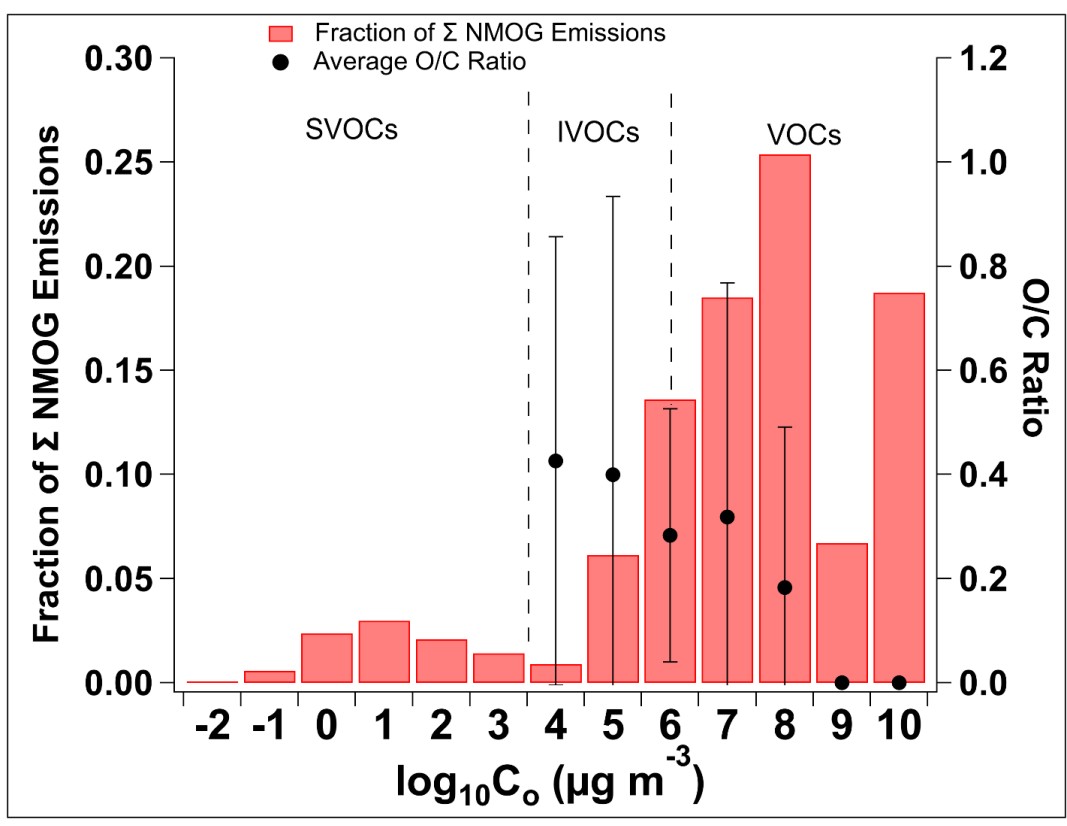

Figure 7. Fraction of total NMOG emissions in each volatility bin, as well as the bin-averaged O/C ratio spanning VOCs, IVOCs and SVOCs. Data is included from PTRMS, CIMS, AWAS and cartridge measurements. The O/C ratio is derived for only the PTRMS, CIMS and AWAS measurements and the errors bars indicate the standard deviation of the average O/C ratio.

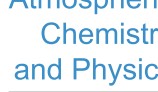
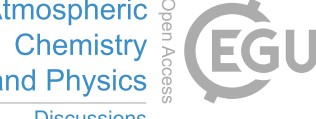

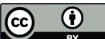

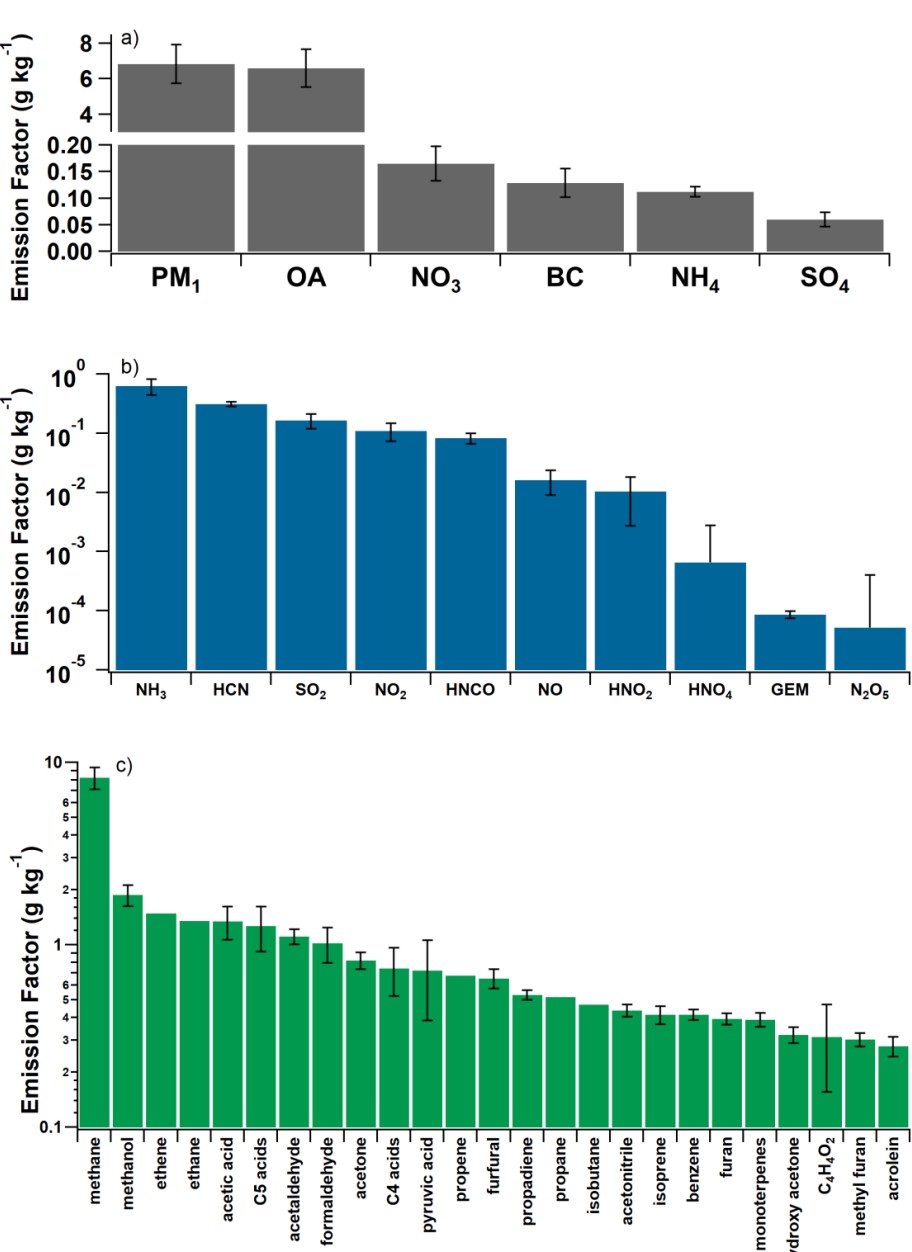

Figure 8. Average emission factors (g kg$^{-1}$) of a) particle species; b) inorganic gas-phase species, and c)
the top 25 measured gas-phase organic species. C5 acids = C5 oxo-carboxylic acids; C4 acids = C4 oxo-
carboxylic acids; propadiene = fragments/propadiene; hydroxy acetone = hydroxy acetone/ ethyl formate.
Organic species measurements are from the PTRMS, CIMS and AWAS.



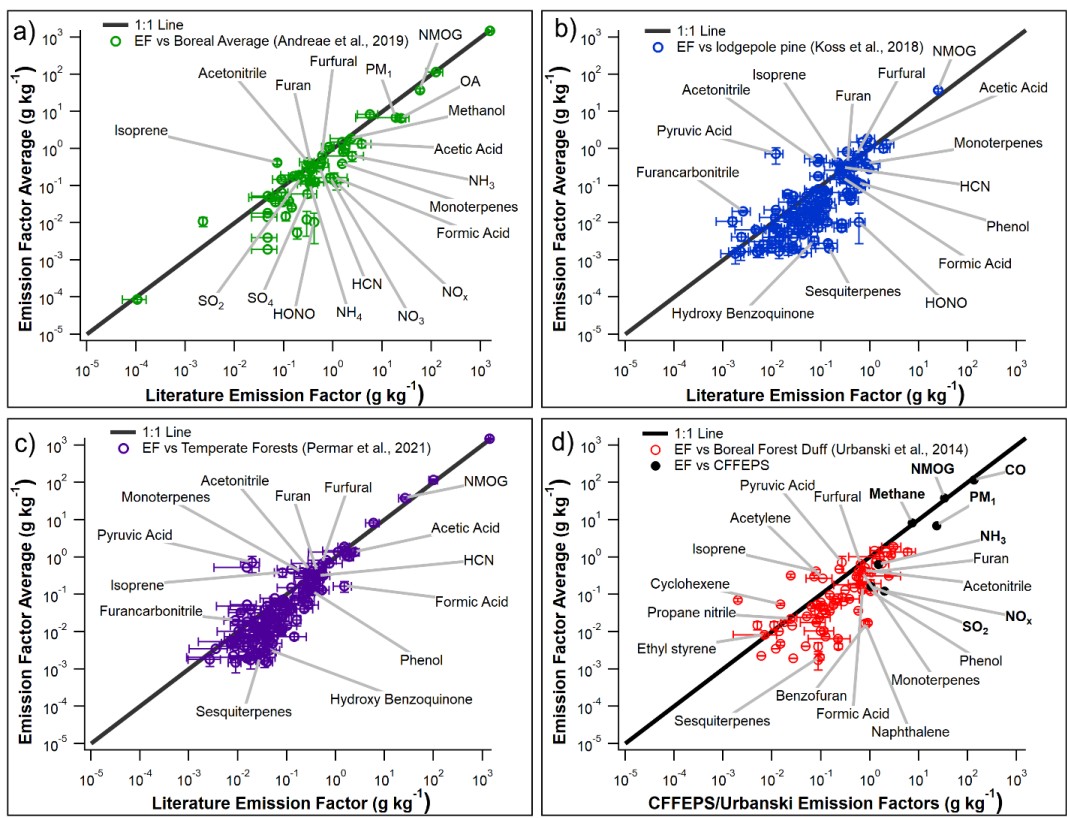

Figure 9.  Comparison of averaged emission factors with a) boreal forest field-based
measurements (Andreae, 2019; Akagi et al., 2011; Liu et al., 2017), b) laboratory-based
measurements of lodgepole pine (Koss et al., 2018), c) temperate forest field-based
measurements (Permar et al., 2021), and d) those used in CFFEPS (Urbanski et al., 2014). See
Table S8 for compound comparisons that don't have exact matches.



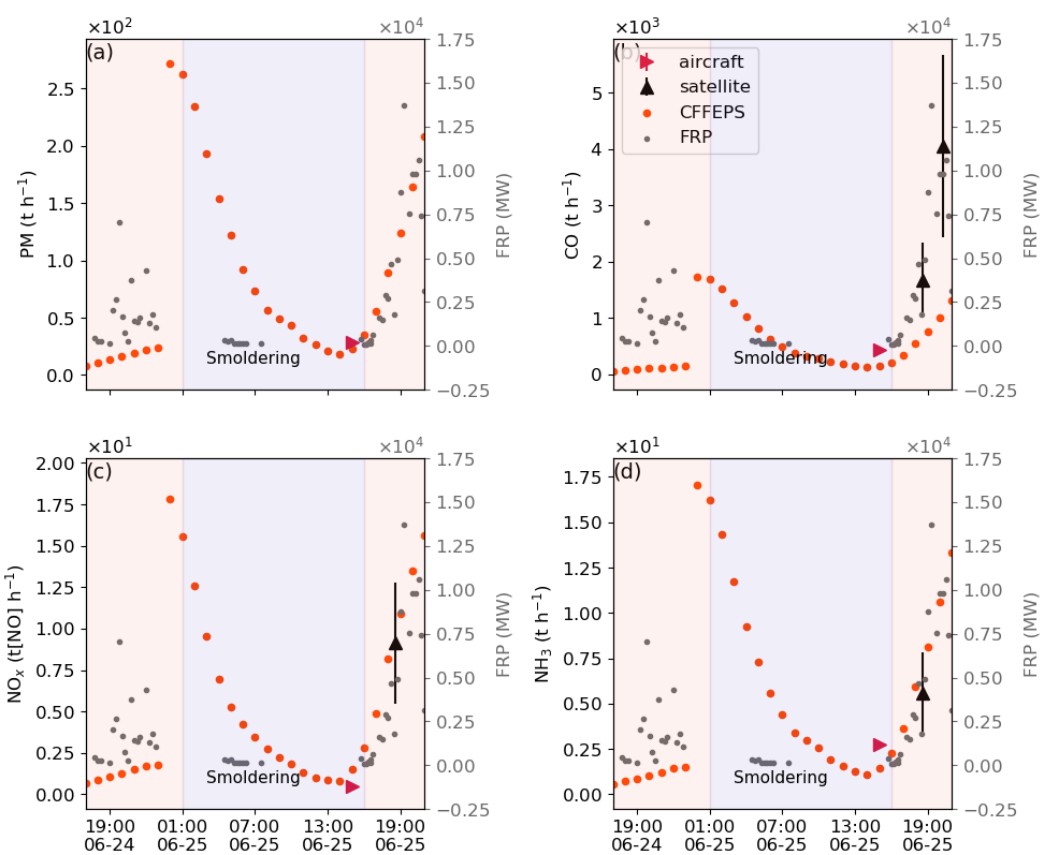

Figure 10. Fire radiative power (FRP; in MW) from GOES-R (grey dots) and emissions from
the CFFEPS model (orange dots) from 2018-06-24 17:00 UTC to 2018-06-25 21:00 UTC.
Aircraft-derived emission rates are shown for a) $PM_1$, b) CO, c) $NO_x$ (as NO) and d) $NH_3$ (in t h$^{-1}$
; red arrow) at 15:00 UTC when the aircraft flew closest to the fire. The corresponding
TROPOMI satellite-derived emission rates are also shown (in t h$^{-1}$; black arrows). Note, the
aircraft flight time occurred when the fire intensity reflected a surface, smoldering fire and the
satellite overpass time took place when the fire had transitioned to a crown (flaming) fire. The
smoldering and flaming time periods are coloured in blue and pink, respectively.





**Table A1**

Table S7.  Summary of emission factors (g kg$^{-1}$) (EF) and emission ratios (ppb ppm$^{-1}$;
particulates in µg m$^{-3}$ ppm$^{-1}$ and TGM in ng m$^{-3}$ ppm$^{-1}$) (ER) for the SP, NP, and the EF average
of the two plumes, grouped by particulate species, and inorganic and organic gas-phase species
(sorted by increasing molecular weight).  PM$_1$ is the sum of all the particulate species.  The CE
was 0.84±0.04 and 0.82±0.01 for the SP and NP, respectively.  For comparison, EFs are also
included from previously published literature: Andreae (2019)[1], Liu et al. (2017)[a]; Akagi et al.
(2011)[b]; and Simpson et al. (2011)[c]), Koss et al. (2018)[2] and Permar et al. (2021)[3].  See Table S8
for compounds that did not have exact matches for comparison to literature values.  For species
measured in mass concentration units, Eq. 2 was modified by converting TC to mass
concentrations using the measured temperature and pressure, and removing the molar mass ratio
term. * Indicates that the compound was 'calculated' (SI Sect 2.1.1) while the remaining
compounds were calibrated.  **Estimated, see text in Sect. 3.4.3. ***Uncertainty reflects the
standard deviation of the calibration.

| Molecular Weight | Compound | Compound Name | Instru-ment | Average EF (g kg$^{-1}$) | NP EF (g kg$^{-1}$) | SP EF (g kg$^{-1}$) | Literature EF (g kg$^{-1}$) | NP ER | SP ER |
|---|---|---|---|---|---|---|---|---|---|
| Particulate | | | | | | | | | |
| | PM$_1$ | particulate matter (<1µm) | AMS | 6.8±1.1 | 7.1±0.48 | 6.6±2.1 | 18.8±15.9[1] | 59±1.5 | 65±15 |
| | BC | black carbon | SP2 | 0.13±0.027 | 0.11±0.0098 | 0.14±0.052 | 0.43±0.21[1] 0.39±0.17[3] | 0.55±0.026 | 0.58±0.32 |
| | NH$_4$ | p-ammonium | AMS | 0.11±0.0097 | 0.11±0.0069 | 0.12±0.018 | 0.34±0.15[1, a] | 1.1±0.066 | 1.9±0.43 |
| | NO$_3$ | p-nitrate | AMS | 0.17±0.032 | 0.14±0.013 | 0.19±0.064 | 0.87±0.13[1, a] | 0.9±0.02 | 1.2±0.028 |
| | SO$_4$ | p-sulphate | AMS | 0.06±0.013 | 0.066±0.003 | 0.055±0.026 | 0.30±0.16[1, a] | 0.035±0.00077 | 0.054±0.015 |
| | OA | p-total organics | AMS | 6.6±1.1 | 6.9±0.33 | 6.3±2.1 | 24.3±0.21[1, a] | 58±0.6 | 62±15 |
| Gas Inorganic | | | | | | | | | |
| 17.031 | NH$_3$ | ammonia | LGR | 0.63±0.19 | 0.45±0.071 | 0.82±0.37 | 2.5±1.8[1] 0.68±0.19[2] | 5.8±0.9 | 13±4.7 |
| 27.026 | HCN | hydrogen cyanide | CIMS | 0.31±0.028 | 0.34±0.026 | 0.29±0.049 | 0.53±0.30[1] 0.28±0.06[2] 0.43±0.17[3] | 2.8±0.19 | 2.9±0.34 |
| 28.01 | CO | carbon monoxide | Picarro | 115.7±7.5 | 127±5.6 | 104.1±14.0 | 120±47[1] 99±20[3] | 110±67 | 130±100 |
| 30.006 | NO | nitric oxide | TECOs | 0.016±0.0072 | 0.017±0.0081 | 0.016±0.012 | 0.29[1] | 0.12±0.068 | 0.14±0.09 |
| 43.025 | HNCO | isocyanic acid | CIMS | 0.083±0.017 | 0.091±0.018 | 0.076±0.03 | 0.57±0.24[2] 0.16±0.036[3] | 0.46±0.074 | 0.47±1.8 |
| 43.025 | HNCO | isocyanic acid* | PTRMS | 0.021±0.0065 | 0.015±0.0025 | 0.026±0.013 | 0.57±0.24[2] | 0.078±0.011 | 0.16±0.066 |
| 44.009 | CO$_2$ | carbon dioxide | Picarro | 1496.3±35.8 | 1481.7±18.4 | 1510.9±69.2 | 1500±140[1] 1400±61[3] | 7400±360 | 9400±340 |
| 46.005 | NO$_2$ | nitrogen dioxide | TECOs | 0.11±0.037 | 0.076±0.0077 | 0.15±0.074 | | 0.37±0.052 | 0.83±0.37 |
| 46.005 | NO$_x$ | sum (NO+NO$_2$) | TECOs | 0.12±0.045 | 0.072±0.037 | 0.17±0.083 | 1.2±0.86[1] | 0.34±0.17 | 0.97±1.2 |





| | | | | | | | | | |
|---|---|---|---|---|---|---|---|---|---|
| 47.013 | $HNO_2$ | nitrous acid | CIMS | 0.01±0.0077 | 0.00048±0.0018 | 0.02±0.015 | 0.41[1] 0.6±0.20[2] | | 0.11±0.077 |
| 64.064 | $SO_2$ | sulphur dioxide | TECOs | 0.17±0.046 | 0.074±0.032 | 0.26±0.085 | 0.22±0.31[1] | 0.26±0.13 | 1.1±0.29 |
| 79.011 | $HNO_4$ | peroxynitric acid | CIMS | 0.00066±0.0021 | 0.00047±0.0042 | 0.00085±0.0059 | | 0.00089±0.012 | 0.0028±0.0019 |
| 108.009 | $N_2O_5$ | dinitrogen pentoxide | CIMS | 5.2E-5±0.00035 | 0.00025±0.00051 | -0.00015±0.005 | | 0.00048±0.001 | -0.0003±0.0012 |
| 200.59 | GEM | gaseous elemental mercury | Tekran | 8.7E-5±1.2E-5 | 8.2E-5±1.4E-5 | 9.2E-5±1.9E-5 | 0.00023±0.00031[1] | 0.00068±0.00012 | 0.00091±0.0001 |
| Gas Organic | | | | | | | | | |
| | ΣNMOG | non methane organic gases | PTRMS+CIMS+AWAS+cartridges | 25.8±3.2 | 26.2±2.1 | 25.4±5.8 | | | |
| | Estimate $NMOG_{Ty}$ | non methane organic gases | PTRMS+CIMS+AWAS+cartridges | 37.8 to 39.9** | | | 58.7[1, b] 25.0[2] 26.1[3] | | |
| | $NMOG_T$ (g **C**/kg) | Carbon fraction of non methane organic gases | Picarro | 31.2±3.8 | 36.8±5.1 | 25.5±5.6 | | 680±120 | 580±150 |
| 16.043 | $CH_4$ | methane | Picarro | 8.3±1.1 | 7.8±0.4 | 8.7±2.2 | 5.5±2.5[1] 5.9±1.8[3] | 110±1.3 | 150±30 |
| 26.038 | $C_2H_2$ | acetylene | AWAS | 0.27 | 0.20 | 0.34 | 0.31±0.17[3] | 2.2 | 4 |
| 27.046 | $C_2H_3$ | fragment vinyl/ethane* | PTRMS | 0.012±0.0017 | 0.012±0.0011 | 0.012±0.0032 | | 0.096±0.013 | 0.12±0.017 |
| 28.054 | $C_2H_4$ | ethene | AWAS | 1.49 | 1.29 | 1.69 | 1.5±0.66[1] 0.71±0.31[2] 1.5±1[3] | 12.94 | 18.31 |
| 30.026 | $CH_2O$ | formaldehyde | PTRMS | 1.0±0.22 | 1.1±0.14 | 0.93±0.43 | 1.8±0.4[1] 1.9±0.67[2] 1.9±0.43[3] | 8.1±0.68 | 8.9±3.3 |
| 30.07 | $C_2H_6$ | ethane* | AWAS | 1.3 | 1.3 | 1.4 | 1.1±0.84[3] | 12 | 14 |
| 32.042 | $CH_4O$ | methanol | PTRMS | 1.9±0.25 | 2.2±0.21 | 1.6±0.45 | 2.3±1.4[1] 0.9±0.35[2] 1.5±0.39[3] | 15±0.85 | 13±3.9 |
| 40.065 | $C_3H_4$ | fragments/propadiene* | PTRMS | 0.53±0.032 | 0.64±0.037 | 0.42±0.051 | 0.06±0.03[1] 0.088±0.041[2] | 3.5±0.062 | 2.8±0.21 |
| 41.053 | $C_2H_3N$ | acetonitrile | PTRMS | 0.44±0.034 | 0.48±0.028 | 0.4±0.062 | 0.31±0.099[1] 0.086±0.028[2] 0.31±0.15[3] | 2.6±0.066 | 2.6±0.1 |
| 42.037 | $C_2H_2O$ | acetic acid fragment* | PTRMS | 1.45±0.17 | 1.46±0.12 | 1.43±0.32 | | 7.67±0.32 | 9.05±1.08 |
| 42.041 | $CH_2N_2$ | cyanamide* | PTRMS | 0.064±0.0087 | 0.067±0.00038 | 0.061±0.017 | | 1.4±0.017 | 1.3±0.074 |
| 42.081 | $C_3H_6$ | propene | AWAS | 0.68 | 0.62 | 0.73 | 0.74±0.62[3] | 4.2 | 5.2 |
| 44.053 | $C_2H_4O$ | acetaldehyde | PTRMS | 1.1±0.11 | 1.2±0.074 | 1.0±0.2 | 0.81±0.23[1] 0.92±0.32[2] 1.7±0.43[3] | 6.3±0.42 | 6.3±0.48 |
| 44.097 | $C_3H_8$ | propane | AWAS | 0.52 | 0.53 | 0.50 | 0.46±0.18[3] | 3.4 | 3.4 |
| 46.025 | $CH_2O_2$ | formic acid | CIMS | 0.17±0.053 | 0.17±0.032 | 0.17±0.10 | 1±0.89[1] 0.28±0.14[2] 1.5±0.60[3] | 1.2±0.093 | 0.56±0.82 |



| | | | | | | | | | |
|---|---|---|---|---|---|---|---|---|---|
| 48.103 | CH$_4$S | methanethiol* | PTRMS | 0.014±0.0016 | 0.015±0.0021 | 0.013±0.0025 | | 0.068±0.0091 | 0.073±0.0081 |
| 50.057 | CH$_6$O$_2$ | methanol hydrate* | PTRMS | 0.028±0.0057 | 0.034±0.0023 | 0.022±0.011 | | 0.15±0.0077 | 0.12±0.065 |
| 52.076 | C$_4$H$_4$ | buten-yne/fragments* | PTRMS | 0.018±0.0016 | 0.02±0.0011 | 0.016±0.0031 | 0.057±0.032[2] 0.052±0.018[3] | 0.086±0.0043 | 0.081±0.0069 |
| 53.064 | C$_3$H$_3$N | acrylonitrile* | PTRMS | 0.036±0.0029 | 0.04±0.0024 | 0.032±0.0052 | 0.025±0.012[2] 0.044±0.015[3] | 0.17±0.0032 | 0.16±0.00054 |
| 54.048 | C$_3$H$_2$O | propynal* | PTRMS | 0.0087±0.0034 | 0.0045±0.0032 | 0.013±0.006 | 0.034±0.014[2] 0.037±0.015[3] | 0.018±0.013 | 0.062±0.024 |
| 54.092 | C$_4$H$_6$ | butadiene/fragments* | PTRMS | 0.15±0.016 | 0.15±0.004 | 0.15±0.031 | 0.089±0.03[1] 0.34±0.18[2] 0.27±0.096[3] | 0.62±0.014 | 0.73±0.067 |
| 54.092 | C$_4$H$_6$ | 1,3-butadiene | AWAS | 0.065 | 0.055 | 0.075 | 0.089±0.03[1] 0.34±0.18[2] 0.27±0.096[3] | 0.29 | 0.41 |
| 55.08 | C$_3$H$_5$N | propane nitrile* | PTRMS | 0.022±0.0017 | 0.025±0.0017 | 0.019±0.003 | 0.012±0.0051[2] 0.037±0.018[3] | 0.1±0.0038 | 0.094±0.0042 |
| 56.064 | C$_3$H$_4$O | acrolein | PTRMS | 0.28±0.035 | 0.29±0.025 | 0.26±0.065 | 0.335[1] 0.97±0.5[2] 0.4±0.18[3] | 0.82±0.04 | 0.83±0.038 |
| 56.108 | C$_4$H$_8$ | cis-2-butene | AWAS | 0.011 | 0.0061 | 0.015 | | 0.03 | 0.078 |
| 56.108 | C$_4$H$_8$ | isobutene | AWAS | 0.084 | 0.082 | 0.086 | | 0.41 | 0.45 |
| 56.108 | C$_4$H$_8$ | t-2,butene | AWAS | 0.0074 | 0.0026 | 0.012 | | 0.013 | 0.063 |
| 56.108 | C$_4$H$_8$ | 1,butene | AWAS | 0.13 | 0.12 | 0.14 | | 0.6 | 0.74 |
| 57.052 | C$_2$H$_3$NO | hydroxy acetonitrile | CIMS | 0.0035±0.00059 | 0.0025±0.00012 | 0.0044±0.0012 | 0.014±0.0048[2] 0.033±0.0087[3] | 0.026±0.0059 | 0.024±0.0047 |
| 57.052 | C$_2$H$_3$NO | methyl isocyanate* | PTRMS | 0.006±0.012 | 0.0068±0.0017 | 0.0052±0.0015 | 0.014±0.0048[2] 0.033±0.0087[3] | 0.026±0.0059 | 0.024±0.0047 |
| 57.096 | C$_3$H$_7$N | propene amine* | PTRMS | 0.0017±0.00073 | 0.0016±0.00086 | 0.0019±0.0012 | 0.0023±0.00099[2] 0.018±0.0082[3] | 0.0061±0.033 | 0.0086±0.0053 |
| 58.08 | C$_3$H$_6$O | acetone | PTRMS | 0.82±0.088 | 0.99±0.13 | 0.65±0.12 | 1.6±1.6[1] 0.34±0.12[2] 0.65±0.38[3] | 0.065±0.014 | 0.072±0.042 |
| 58.124 | C$_4$H$_{10}$ | isobutane | AWAS | 0.47 | 0.52 | 0.42 | 0.12±0.061[3] | 2.6 | 2.2 |
| 58.124 | C$_4$H$_{10}$ | n-butane* | AWAS | 0.15 | 0.16 | 0.14 | 0.12±0.061[3] | 0.79 | 0.73 |
| 59.068 | C$_2$H$_5$NO | acetamide* | PTRMS | 0.0016±0.0021 | 0.0054±0.0038 | -0.0023±0.0017 | 0.043±0.021[2] 0.04±0.012[3] | 0.02±0.014 | -0.01±0.0068 |
| 60.052 | C$_2$H$_4$O$_2$ | acetic acid | CIMS | 1.3±0.28 | 1.1±0.19 | 1.6±0.53 | 3.8±2.1[1] 2.2±0.89[2] 2.4±0.61[3] | 7.4±0.37 | 8.9±1 |
| 60.056 | CH$_4$N$_2$O | urea* | PTRMS | 0.078±0.012 | 0.079±0.015 | 0.076±0.019 | | 0.29±0.044 | 0.34±0.048 |
| 60.096 | C$_3$H$_8$O | propanol* | PTRMS | 0.0054±0.0017 | 0.0061±0.0024 | 0.0046±0.0025 | 0.0074±0.0058[3] | 0.022±0.0082 | 0.02±0.0097 |
| 61.04 | CH$_3$NO$_2$ | nitromethane* | PTRMS | 0.011±0.0021 | 0.01±0.0011 | 0.011±0.004 | 0.074±0.03[2] 0.078±0.0085[3] | 0.036±0.0049 | 0.048±0.013 |
| 62.068 | C$_2$H$_6$O$_2$ | ethylene glycol* | PTRMS | 0.0038±0.00047 | 0.004±0.00041 | 0.0036±0.00086 | | 0.014±0.0084 | 0.015±0.0036 |
| 62.13 | C$_2$H$_6$S | dimethyl sulfide | PTRMS | 0.011±0.0031 | 0.016±0.0061 | 0.0067±0.0014 | 0.0047[1,a] 0.0016±0.00084[2] 0.08±0.083[3] | 0.017±0.004 | 0.029±0.0065 |





| 64.04 | $CH_4O_3$ | methanetriol* | PTRMS | 0.0011±0.0025 | 0.0036±0.00059 | -0.0013±0.005 | | 0.013±0.0024 | -0.0049±0.02 |
|---|---|---|---|---|---|---|---|---|---|
| 64.087 | $C_5H_4$ | * | PTRMS | 0.003±0.00091 | 0.0042±0.0017 | 0.0017±0.00076 | | 0.014±0.0052 | 0.0074±0.003 |
| 66.103 | $C_5H_6$ | cyclopentadiene* | PTRMS | 0.032±0.0031 | 0.041±0.0047 | 0.023±0.0042 | 0.011±0.0049[3] | 0.14±0.012 | 0.096±0.014 |
| 67.091 | $C_4H_5N$ | pyrrole* | PTRMS | 0.026±0.0022 | 0.027±0.00062 | 0.025±0.0043 | 0.054±0.029[2] 0.039±0.021[3] | 0.09±0.0051 | 0.098±0.0067 |
| 68.075 | $C_4H_4O$ | furan* | PTRMS | 0.39±0.028 | 0.43±0.02 | 0.35±0.052 | 0.36±0.44[1] 0.36±0.11[2] 0.43±0.19[3] | 1.4±0.049 | 1.4±0.076 |
| 68.119 | $C_5H_8$ | isoprene | PTRMS | 0.41±0.10 | 0.64±0.078 | 0.19±0.06 | 0.074[1] 0.22±0.11[2] 0.082±0.095[3] | 2.1±0.22 | 0.47±0.61 |
| 69.083 | $C_4H_5O$ | * | PTRMS | 0.0043±0.00039 | 0.0047±0.00016 | 0.0038±0.00077 | | 0.015±0.0012 | 0.015±0.019 |
| 69.107 | $C_4H_7N$ | butane nitrile* | PTRMS | 0.0077±0.00064 | 0.0088±0.00061 | 0.0065±0.00011 | 0.011±0.0048[2] 0.02±0.01[3] | 0.028±0.00085 | 0.025±0.0018 |
| 70.091 | $C_4H_6O$ | methyl vinyl ketone, methacrolein, crotonaldehyde | PTRMS | 0.19±0.055 | 0.2±0.0039 | 0.18±0.11 | 0.11±0.12[1] 0.34±0.15[2] 0.39±0.15[3] | 0.66±0.013 | 0.68±0.37 |
| 70.135 | $C_5H_{10}$ | pentene/fragments* | PTRMS | 0.018±0.0017 | 0.019±0.0022 | 0.018±0.0025 | 0.046±0.025[1] 0.028±0.01[2] 0.015±0.0084[3] | 0.059±0.009 | 0.069±0.0053 |
| 70.135 | $C_5H_{10}$ | c-2-pentene | AWAS | 0.004 | 0.0033 | 0.0048 | 0.046±0.025[1] 0.028±0.01[2] 0.015±0.0084[3] | 0.013 | 0.021 |
| 70.135 | $C_5H_{10}$ | cyclopentane | AWAS | 0.0035 | 0.0038 | 0.0031 | 0.0035±0.0025[3] | 0.016 | 0.014 |
| 70.135 | $C_5H_{10}$ | 1-pentene | AWAS | 0.052 | 0.053 | 0.052 | 0.046±0.025[1] 0.028±0.01[2] 0.015±0.0084[3] | 0.21 | 0.22 |
| 70.135 | $C_5H_{10}$ | 2-methyl,1-butene | AWAS | 0.014 | 0.014 | 0.015 | 0.046±0.025[1] 0.028±0.01[2] 0.015±0.0084[3] | 0.056 | 0.062 |
| 70.135 | $C_5H_{10}$ | 2-methyl,2-butene | AWAS | 0.0019 | 0.0017 | 0.0022 | 0.046±0.025[1] 0.028±0.01[2] 0.015±0.0084[3] | 0.0068 | 0.0095 |
| 72.063 | $C_3H_4O_2$ | acrylic acid | CIMS | 0.096±0.0098 | 0.13±0.0091 | 0.062±0.017 | 0.22±0.082[3] | 0.25±0.0095 | 0.35±0.087 |
| 72.107 | $C_4H_8O$ | MEK + butanal + 2-methylpropanal | PTRMS | 0.18±0.015 | 0.22±0.012 | 0.14±0.027 | 0.16±0.036[1] 0.087±0.028[2] 0.21±0.063[3] | 0.67±0.01 | 0.54±0.063 |
| 72.151 | $C_5H_{12}$ | n-pentane | AWAS | 0.078 | 0.086 | 0.07 | 0.057±0.028[3] | 0.34 | 0.29 |
| 72.151 | $C_5H_{12}$ | 2-methylbutane | AWAS | 0.022 | 0.024 | 0.021 | 0.057±0.028[3] | 0.097 | 0.086 |
| 73.095 | $C_3H_7NO$ | dimethylformamide* | PTRMS | 0.001±0.0006 | 0.0018±0.00052 | 0.00024±0.0011 | | 0.0053±0.014 | 0.0011±0.0038 |
| 74.079 | $C_3H_6O_2$ | propionic acid | CIMS | 0.13±0.04 | 0.12±0.0042 | 0.14±0.08 | | 1±0.041 | 1.1±0.096 |
| 74.079 | $C_3H_6O_2$ | hydroxy acetone/ethyl formate* | PTRMS | 0.32±0.033 | 0.35±0.025 | 0.3±0.06 | 0.49±0.19[2] 0.57±0.2[3] | 1±0.041 | 1.1±0.096 |



| | | | | | | | | | |
|---|---|---|---|---|---|---|---|---|---|
| 78.114 | $C_6H_6$ | benzene | PTRMS | 0.41±0.027 | 0.47±0.021 | 0.36±0.05 | 0.57±0.21[1] 0.42±0.25[2] 0.5±0.14[3] | 1.3±0.0016 | 1.2±0.046 |
| 80.086 | $C_5H_4O$ | cyclopentadienone/isomers* | PTRMS | 0.011±0.0014 | 0.0093±0.00043 | 0.012±0.0028 | 0.13±0.075[2] 0.027±0.017[3] | 0.026±0.0017 | 0.04±0.0068 |
| 80.13 | $C_6H_8$ | cyclohexadiene/monoterpene fragment* | PTRMS | 0.14±0.01 | 0.17±0.011 | 0.1±0.018 | | 0.48±0.037 | 0.34±0.06 |
| 81.094 | $C_5H_5O$ | * | PTRMS | 0.0039±0.00084 | 0.004±0.00032 | 0.0037±0.0016 | | 0.011±0.00039 | 0.012±0.0045 |
| 81.118 | $C_5H_7N$ | pentene nitriles/methyl pyrrole* | PTRMS | 0.0047±0.00051 | 0.005±0.00012 | 0.0044±0.0001 | 0.02±0.011[3] | 0.014±0.00029 | 0.015±0.002 |
| 82.102 | $C_5H_6O$ | methyl furan* | PTRMS | 0.3±0.026 | 0.31±0.011 | 0.29±0.05 | 0.32±0.11[2] 0.28±0.13[3] | 0.84±0.047 | 0.96±0.041 |
| 82.146 | $C_6H_{10}$ | cyclohexene* | PTRMS | 0.054±0.0035 | 0.075±0.0029 | 0.033±0.0064 | | 0.2±0.012 | 0.11±0.013 |
| 83.09 | $C_4H_5NO$ | methyloxazole* | PTRMS | 0.003±0.00052 | 0.0039±0.001 | 0.002±0.00018 | | 0.01±0.0023 | 0.0066±0.00067 |
| 83.134 | $C_5H_9N$ | pentanenitriles* | PTRMS | 0.016±0.00096 | 0.019±0.00046 | 0.013±0.0019 | 0.021±0.011[3] | 0.049±0.0011 | 0.042±0.0048 |
| 84.074 | $C_4H_4O_2$ | | CIMS | 0.31±0.16 | 0.1±0.092 | 0.52±0.3 | | 0.42±0.026 | 0.48±0.072 |
| 84.074 | $C_4H_4O_2$ | furanone* | PTRMS | 0.16±0.02 | 0.16±0.017 | 0.15±0.036 | 0.4±0.15[2] 0.32±0.11[3] | 0.42±0.026 | 0.48±0.072 |
| 84.118 | $C_5H_8O$ | cyclopentanone/isomers* | PTRMS | 0.069±0.0056 | 0.073±0.004 | 0.065±0.011 | 0.12±0.04[2] 0.087±0.038[3] | 0.19±0.012 | 0.21±0.0077 |
| 84.162 | $C_6H_{12}$ | hexene* | PTRMS | 0.015±0.0036 | 0.02±0.0033 | 0.0098±0.0064 | 0.11[1] | 0.052±0.0064 | 0.031±0.0017 |
| 84.162 | $C_6H_{12}$ | cis-2-hexene | AWAS | 0.0020 | 0.0021 | 0.0020 | | 0.0069 | 0.0064 |
| 84.162 | $C_6H_{12}$ | cyclohexane | AWAS | 0.0022 | 0.0019 | 0.0026 | 0.008±0.014[3] | 0.0064 | 0.0097 |
| 85.062 | $C_3H_3NO_2$ | methyl cyanoformate * | PTRMS | 0.0011±0.0007 | 0.00068±0.00011 | 0.0016±0.00082 | | 0.0018±0.00029 | 0.0051±0.0025 |
| 85.106 | $C_4H_7NO$ | $C_4H_5N$ water cluster* | PTRMS | 0.00071±0.00037 | 0.00067±0.00015 | 0.00076±0.0072 | | 0.0017±0.0034 | 0.0024±0.0021 |
| 86.09 | $C_4H_6O_2$ | methacrylic acid | CIMS | 0.1±0.024 | 0.11±0.012 | 0.097±0.047 | | 0.33±0.034 | 0.41±0.051 |
| 86.09 | $C_4H_6O_2$ | butanedione/isomers * | PTRMS | 0.13±0.016 | 0.13±0.016 | 0.13±0.028 | 0.34[1] 0.42±0.17[2] 0.53±0.21[3] | 0.33±0.034 | 0.41±0.051 |
| 86.134 | $C_5H_{10}O$ | pentanone* | PTRMS | 0.046±0.0038 | 0.053±0.0058 | 0.038±0.0048 | | 0.0095±0.00061 | 0.008±0.0005 |
| 86.178 | $C_6H_{14}$ | n-hexane | AWAS | 0.049 | 0.053 | 0.044 | 0.054±0.035[1] 0.04±0.036[3] | 0.17 | 0.16 |
| 86.178 | $C_6H_{14}$ | 2,3 dimethylbenzene | AWAS | 0.0031 | 0.004 | 0.0022 | | 0.014 | 0.0066 |
| 86.178 | $C_6H_{14}$ | 2,3-methylpentane | AWAS | 0.01 | 0.0089 | 0.011 | 0.01±0.0065[3] | 0.032 | 0.039 |
| 88.062 | $C_3H_4O_3$ | pyruvic acid | CIMS | 0.72±0.34 | 0.56±0.13 | 0.89±0.66 | 0.012±0.0047[2] 0.019±0.008[3] | 0.022±0.012 | -0.0025±0.02 |



| Mass | Formula | Name | Instrument | | | | | | |
|---|---|---|---|---|---|---|---|---|---|
| 88.106 | $C_4H_8O_2$ | C4 saturated carboxylic acids | CIMS | 0.082±0.023 | 0.12±0.044 | 0.047±0.012 | | 0.13±0.053 | 0.16±0.052 |
| 88.106 | $C_4H_8O_2$ | methyl propanoate* | PTRMS | 0.07±0.0078 | 0.075±0.01 | 0.065±0.012 | 0.073±0.023[2] 0.081±0.036[3] | 0.19±0.019 | 0.2±0.014 |
| 88.168 | $C_4H_8OS$ | oxathiane* | PTRMS | 0.0031±0.0018 | 0.0023±0.0017 | 0.004±0.0032 | | 0.0058±0.0041 | 0.012±0.0088 |
| 90.122 | $C_4H_{10}O_2$ | MEK water cluster/butane diol* | PTRMS | 0.0017±0.00051 | 0.0017±0.00032 | 0.0016±0.00097 | | 0.0041±0.00084 | 0.005±0.0029 |
| 90.125 | $C_7H_6$ | * | PTRMS | 0.0069±0.0012 | 0.0064±0.0018 | 0.0074±0.0017 | | 0.016±0.0044 | 0.022±0.0032 |
| 90.184 | $C_4H_{10}S$ | diethyl sulfide, butanethiol | PTRMS | 0.077±0.0084 | 0.083±0.0077 | 0.071±0.015 | | 0.2±0.028 | 0.21±0.04 |
| 91.113 | $C_6H_5N$ | ethylnylpyrrole* | PTRMS | 0.0015±0.00069 | 0.0018±0.0013 | 0.0011±0.00031 | 0.0018±0.00088[2] 0.0091±0.0026[3] | 0.0044±0.0031 | 0.0033±0.00054 |
| 92.141 | $C_7H_8$ | toluene | PTRMS | 0.26±0.077 | 0.26±0.014 | 0.26±0.15 | 0.35±0.11[1] 0.25±0.13[2] 0.42±0.16[3] | 0.63±0.01 | 0.71±0.38 |
| 93.082 | $C_2H_7NO_3$ | * | PTRMS | 0.0025±0.00046 | 0.0025±0.00028 | 0.0025±0.00088 | | 0.006±0.0009 | 0.007±0.0019 |
| 93.085 | $C_5H_3NO$ | furancarbonitrile* | PTRMS | 0.02±0.0016 | 0.022±0.0018 | 0.018±0.0026 | 0.0026±0.001[2] 0.0088±0.0037[3] | 0.053±0.0033 | 0.053±0.0031 |
| 93.129 | $C_6H_7N$ | methyl pyridine* | PTRMS | 0.0017±0.00059 | 0.0017±0.0011 | 0.0017±0.00049 | 0.014±0.0073[2] 0.035±0.012[3] | 0.004±0.0025 | 0.0047±0.0011 |
| 94.113 | $C_6H_6O$ | phenol* | PTRMS | 0.12±0.018 | 0.12±0.014 | 0.12±0.033 | 0.57±0.36[2] 0.33±0.13[3] | 0.28±0.031 | 0.35±0.053 |
| 94.13 | $C_2H_6O_2S$ | dimethyl sulfone* | PTRMS | 0.0042±0.00079 | 0.0048±0.0013 | 0.0035±0.00086 | | 0.055±0.02 | 0.015±0.0021 |
| 94.157 | $C_7H_{10}$ | cycloheptadiene* | PTRMS | 0.021±0.0034 | 0.023±0.0028 | 0.02±0.0061 | | 0.053±0.0046 | 0.056±0.012 |
| 94.19 | $C_2H_6S_2$ | dimethyl disulfide* | PTRMS | 0.0041±0.00097 | 0.0044±0.0014 | 0.0039±0.0013 | 0.0024±0.0009[2] | 0.01±0.003 | 0.011±0.026 |
| 95.077 | $C_5H_3O_2$ | * | PTRMS | 0.0041±0.00031 | 0.0043±0.00039 | 0.0038±0.00048 | | 0.0099±0.0013 | 0.011±0.0016 |
| 95.101 | $C_5H_5NO$ | pyridinol* | PTRMS | 0.0022±0.0006 | 0.0021±0.00069 | 0.0023±0.00097 | 0.0099±0.0054[2] | 0.0048±0.0017 | 0.0063±0.0023 |
| 95.145 | $C_6H_9N$ | C2 pyrrole* | PTRMS | 0.0024±0.00023 | 0.0027±0.00034 | 0.0021±0.00031 | | 0.0063±0.00066 | 0.006±0.0046 |
| 96.085 | $C_5H_4O_2$ | furfural* | PTRMS | 0.65±0.08 | 0.67±0.044 | 0.64±0.15 | 0.61[1] 0.54±0.17[2] 0.53±0.21[3] | 1.5±0.045 | 1.8±0.23 |
| 96.129 | $C_6H_8O$ | C2-furan* | PTRMS | 0.087±0.011 | 0.086±0.0063 | 0.087±0.022 | | 0.2±0.018 | 0.24±0.033 |
| 96.173 | $C_7H_{12}$ | cycloheptene* | PTRMS | 0.022±0.0029 | 0.033±0.0047 | 0.011±0.0034 | | 0.076±0.01 | 0.031±0.0081 |
| 97.073 | $C_4H_3NO_2$ | * | PTRMS | 0.004±0.0022 | 0.0044±0.0035 | 0.0036±0.0028 | | 0.0098±0.0076 | 0.0096±0.0072 |
| 97.117 | $C_5H_7NO$ | * | PTRMS | 0.002±0.00039 | 0.0023±0.0004 | 0.0017±0.00066 | | 0.0054±0.011 | 0.0045±0.0014 |
| 97.161 | $C_6H_{11}N$ | hexanenitrile* | PTRMS | 0.004±0.00043 | 0.0041±0.00018 | 0.004±0.00083 | 0.0088±0.0047[3] | 0.0093±0.0029 | 0.011±0.014 |
| 98.057 | $C_4H_2O_3$ | maleic anhydride* | PTRMS | 0.07±0.013 | 0.072±0.011 | 0.068±0.023 | 0.14±0.072[3] | 0.16±0.018 | 0.18±0.044 |





| mass | formula | name | instrument | | | | | | |
|---|---|---|---|---|---|---|---|---|---|
| 98.101 | $C_5H_6O_2$ | furan methanol/isomers* | PTRMS | 0.058±0.01 | 0.061±0.0059 | 0.054±0.019 | 0.38±0.15[2] 0.09±0.043[3] | 0.14±0.009 | 0.15±0.038 |
| 98.145 | $C_6H_{10}O$ | methyl cyclopentanone/isomers* | PTRMS | 0.015±0.0014 | 0.017±0.0013 | 0.013±0.0025 | 0.022±0.0086[2] 0.034±0.015[3] | 0.038±0.0022 | 0.035±0.0039 |
| 98.163 | $C_5H_6S$ | methyl thiophene | PTRMS | 0.0059±0.0045 | 0.0069±0.006 | 0.0048±0.0068 | 0.021±0.012[2] | 0.017±0.015 | 0.013±0.017 |
| 98.189 | $C_7H_{14}$ | heptene* | PTRMS | 0.0039±0.0011 | 0.0059±0.0019 | 0.0019±0.00089 | | 0.013±0.0038 | 0.0054±0.002 |
| 100.117 | $C_5H_8O_2$ | unsaturated C5 carboxylic acids | CIMS | 0.072±0.017 | 0.1±0.015 | 0.045±0.029 | | 0.22±0.026 | 0.13±0.085 |
| 100.117 | $C_5H_8O_2$ | methyl methacrylate/isomers* | PTRMS | 0.036±0.009 | 0.035±0.0077 | 0.037±0.016 | 0.14±0.053[2] 0.11±0.045[3] | 0.078±0.016 | 0.098±0.036 |
| 100.161 | $C_6H_{12}O$ | hexanal/hexanones* | PTRMS | 0.0065±0.00079 | 0.0074±0.001 | 0.0057±0.0012 | 0.0046±0.0029[2] 0.013±0.0056[3] | 0.016±0.002 | 0.015±0.002 |
| 100.205 | $C_7H_{16}$ | 2,2,3-trimethylbutane | AWAS | 0.00062 | 0.00067 | 0.00057 | | 0.0018 | 0.0015 |
| 100.205 | $C_7H_{16}$ | 3,3-dimethylpentane | AWAS | 0.035 | 0.054 | 0.016 | | 0.15 | 0.052 |
| 100.205 | $C_7H_{16}$ | 3-methylhexane | AWAS | 0.0075 | 0.0090 | 0.0060 | 0.016±0.018[3] | 0.027 | 0.017 |
| 102.089 | $C_4H_6O_3$ | C4 oxo-carboxylic acids | CIMS | 0.74±0.22 | 0.57±0.1 | 0.92±0.43 | | 1.2±0.26 | 2.4±0.89 |
| 102.089 | $C_4H_6O_3$ | acetic anhydride* | PTRMS | 0.0075±0.0015 | 0.0078±0.00036 | 0.0072±0.003 | 0.089±0.034[2] 0.044±0.02[3] | 0.017±0.0064 | 0.019±0.0068 |
| 102.133 | $C_5H_{10}O_2$ | C5 saturated carboxylic acids | CIMS | 0.012±0.0052 | 0.018±0.004 | 0.0055±0.0096 | | 0.039±0.01 | 0.016±0.027 |
| 102.133 | $C_5H_{10}O_2$ | valeric acid* | PTRMS | 0.024±0.0044 | 0.027±0.0036 | 0.02±0.0081 | | 0.059±0.0067 | 0.052±0.016 |
| 102.177 | $C_6H_{14}O$ | hexanol* | PTRMS | 0.002±0.00032 | 0.0022±0.00054 | 0.0017±0.00033 | | 0.0047±0.0098 | 0.0045±0.00075 |
| 103.121 | $C_4H_9NO_2$ | * | PTRMS | 0.0069±0.0011 | 0.0074±0.00091 | 0.0064±0.0019 | | 0.016±0.0014 | 0.016±0.0039 |
| 103.124 | $C_7H_5N$ | benzonitrile* | PTRMS | 0.06±0.0044 | 0.065±0.0021 | 0.054±0.0086 | 0.021±0.0045[2] 0.055±0.022[3] | 0.14±0.0025 | 0.14±0.0088 |
| 103.165 | $C_5H_{13}NO$ | * | PTRMS | 0.0011±0.00028 | 0.0013±0.00013 | 0.00095±0.0054 | | 0.0028±0.0026 | 0.0025±0.0014 |
| 104.108 | $C_7H_4O$ | * | PTRMS | 0.0019±0.00043 | 0.0021±0.00024 | 0.0017±0.00083 | | 0.0045±0.005 | 0.0043±0.0018 |
| 104.149 | $C_5H_{12}O_2$ | pentanediol* | PTRMS | 0.0029±0.00082 | 0.0033±0.0013 | 0.0024±0.001 | | 0.0069±0.025 | 0.006±0.0022 |
| 104.152 | $C_8H_8$ | styrene* | PTRMS | 0.039±0.0027 | 0.056±0.0014 | 0.022±0.0053 | 0.088±0.056[2] 0.018±0.012[3] | 0.12±0.0024 | 0.058±0.0089 |
| 106.121 | $C_4H_{10}O_3$ | diethylene glycol* | PTRMS | 0.0048±0.0019 | 0.006±0.0022 | 0.0036±0.003 | | 0.012±0.0042 | 0.0088±0.0067 |
| 106.124 | $C_7H_6O$ | benzaldehyde* | PTRMS | 0.036±0.0035 | 0.042±0.0012 | 0.03±0.0069 | 0.095±0.053[2] 0.084±0.026[3] | 0.087±0.0029 | 0.077±0.088 |
| 106.168 | $C_8H_{10}$ | C8 aromatics | PTRMS | 0.075±0.0069 | 0.082±0.0065 | 0.068±0.012 | 0.12±0.052[2] 0.21±0.08[3] | 0.17±0.013 | 0.17±0.0099 |





| | | | | | | | | | |
|---|---|---|---|---|---|---|---|---|---|
| 107.112 | $C_6H_5NO$ | pyridine aldehyde* | PTRMS | 0.0012±0.00024 | 0.00094±0.00032 | 0.0015±0.00035 | | 0.0019±0.00067 | 0.0038±0.00063 |
| 107.156 | $C_7H_9N$ | dimethyl pryidine/heptyl nitriles* | PTRMS | 0.0016±0.00048 | 0.0015±0.00042 | 0.0018±0.00087 | 0.005±0.0033[2] | 0.0031±0.0075 | 0.0043±0.0019 |
| 108.096 | $C_6H_4O_2$ | benzoquinone/quinone* | PTRMS | 0.025±0.004 | 0.024±0.0027 | 0.025±0.0076 | 0.084±0.024[2] 0.077±0.02[3] | 0.049±0.007 | 0.062±0.013 |
| 108.14 | $C_7H_8O$ | methyl phenol/anisol/cresol* | PTRMS | 0.04±0.0055 | 0.04±0.0063 | 0.04±0.009 | 0.41±0.17[2] 0.23±0.11[3] | 0.083±0.015 | 0.099±0.011 |
| 108.184 | $C_8H_{12}$ | cyclooctadiene* | PTRMS | 0.015±0.0018 | 0.017±0.00037 | 0.013±0.0036 | | 0.034±0.0014 | 0.032±0.0052 |
| 109.104 | $C_6H_5O_2$ | * | PTRMS | 0.0055±0.00065 | 0.0055±0.00067 | 0.0055±0.0011 | | 0.011±0.0015 | 0.014±0.0028 |
| 109.128 | $C_6H_7NO$ | * | PTRMS | 0.002±0.0011 | 0.0017±0.00039 | 0.0024±0.0021 | | 0.0035±0.0075 | 0.0057±0.005 |
| 110.112 | $C_6H_6O_2$ | benzenediol/ methyl furfural* | PTRMS | 0.11±0.015 | 0.11±0.005 | 0.11±0.03 | 0.68±0.29[2] 0.25±0.12[3] | 0.21±0.0056 | 0.27±0.044 |
| 110.156 | $C_7H_{10}O$ | norcamphor/C3 furan* | PTRMS | 0.032±0.004 | 0.03±0.0029 | 0.034±0.0076 | 0.079±0.026[2] 0.046±0.024[3] | 0.059±0.0039 | 0.083±0.01 |
| 110.2 | $C_8H_{14}$ | cyclooctene* | PTRMS | 0.0088±0.0019 | 0.012±0.0027 | 0.0053±0.0026 | | 0.024±0.0047 | 0.014±0.0059 |
| 111.1 | $C_5H_5NO_2$ | dihydroxy piridine/methyl maleimide* | PTRMS | 0.0026±0.00081 | 0.0031±0.0004 | 0.0022±0.0016 | 0.0066±0.0023[2] 0.024±0.0084[3] | 0.0061±0.0069 | 0.0051±0.0036 |
| 111.144 | $C_6H_9NO$ | * | PTRMS | 0.0018±0.00032 | 0.0028±0.00056 | 0.00091±0.0003 | | 0.0054±0.0092 | 0.0023±0.00057 |
| 112.084 | $C_5H_4O_3$ | furoic acid/hydroxy furfural* | PTRMS | 0.041±0.0081 | 0.044±0.0041 | 0.038±0.016 | 0.11±0.043[2] 0.12±0.031[3] | 0.087±0.0049 | 0.089±0.029 |
| 112.128 | $C_6H_8O_2$ | cyclohexanedione* | PTRMS | 0.014±0.0019 | 0.014±0.0034 | 0.014±0.0017 | | 0.028±0.0081 | 0.033±0.0041 |
| 112.172 | $C_7H_{12}O$ | ethylcycloheptanone* | PTRMS | 0.0069±0.0017 | 0.0067±0.001 | 0.007±0.0032 | 0.012±0.0056[2] 0.014±0.007[3] | 0.013±0.0014 | 0.016±0.0061 |
| 112.19 | $C_6H_8S$ | dimethylthiophene | PTRMS | 0.023±0.0098 | 0.028±0.016 | 0.018±0.012 | | 0.055±0.029 | 0.042±0.025 |
| 112.216 | $C_8H_{16}$ | octene* | PTRMS | 0.0017±0.00079 | 0.0021±0.0014 | 0.0013±0.0082 | | 0.0042±0.025 | 0.0027±0.0021 |
| 114.144 | $C_6H_{10}O_2$ | sum of cyclic saturated and n-unsaturated C5 carboxylic acids | CIMS | 0.0018±0.00026 | 0.0029±0.00038 | 0.00065±0.00035 | | 0.0025±0.0075 | 0.0014±0.00085 |
| 114.144 | $C_6H_{10}O_2$ | caprolactone/ c6 esters/ c6 diketone isomers* | PTRMS | 0.0068±0.0011 | 0.0082±0.00098 | 0.0053±0.0021 | 0.034±0.014[2] 0.039±0.017[3] | 0.016±0.0014 | 0.013±0.0043 |
| 114.188 | $C_7H_{14}O$ | heptanone/heptanal/isomers* | PTRMS | 0.005±0.00078 | 0.006±0.0012 | 0.0039±0.001 | 0.0091±0.0045[2] 0.0072±0.0025[3] | 0.012±0.0019 | 0.009±0.0013 |
| 114.232 | $C_8H_{18}$ | 2,2-dimethylhexane | AWAS | 0.0029 | 0.0036 | 0.0022 | | 0.0095 | 0.0056 |





| Mass | Formula | Name | Method | | | | | | |
|---|---|---|---|---|---|---|---|---|---|
| 114.232 | $C_8H_{18}$ | 4-methylheptane | AWAS | 0.015 | 0.015 | 0.015 | | 0.037 | 0.041 |
| 116.116 | $C_5H_8O_3$ | C5 oxo-carboxylic acids | CIMS | 1.3±0.35 | 1.1±0.15 | 1.5±0.68 | | 2±0.34 | 3.4±1.2 |
| 116.143 | $C_8H_6N$ | * | PTRMS | 0.0028±0.00063 | 0.0035±0.0011 | 0.0021±0.00068 | | 0.0065±0.0018 | 0.0049±0.0011 |
| 116.16 | $C_6H_{12}O_2$ | C6 saturated carboxylic acids | CIMS | 0.0001±0.00041 | 0.00013±0.00075 | 7.5E-5±0.00032 | | 0.00019±0.0014 | 0.00024±0.00079 |
| 116.16 | $C_6H_{12}O_2$ | butyl acetate/c6 esters * | PTRMS | 0.0073±0.0015 | 0.0094±0.0013 | 0.0052±0.0028 | 0.012±0.0081[2] 0.011±0.0062[3] | 0.018±0.0018 | 0.012±0.0056 |
| 116.204 | $C_7H_{16}O$ | heptanol* | PTRMS | 0.0018±0.00043 | 0.0016±0.00037 | 0.0019±0.00078 | | 0.0031±0.0074 | 0.0043±0.0015 |
| 116.222 | $C_6H_{12}S$ | cyclohexanethiol* | PTRMS | 0.0032±0.0012 | 0.004±0.0012 | 0.0025±0.00021 | | 0.0075±0.02 | 0.0056±0.0042 |
| 118.088 | $C_4H_6O_4$ | succinic acid* | PTRMS | 0.0017±0.00029 | 0.0026±0.00049 | 0.00081±0.00031 | | 0.0048±0.0076 | 0.0018±0.00055 |
| 118.135 | $C_8H_6O$ | benzofuran* | PTRMS | 0.017±0.0017 | 0.018±0.00066 | 0.017±0.0034 | 0.037±0.02[2] 0.041±0.015[3] | 0.034±0.0027 | 0.038±0.003 |
| 118.179 | $C_9H_{10}$ | methylstyrenes/propenyl benzenes* | PTRMS | 0.018±0.0014 | 0.024±0.0012 | 0.011±0.0026 | 0.05±0.03[2] 0.037±0.019[3] | 0.046±0.0075 | 0.025±0.0047 |
| 119.123 | $C_7H_5NO$ | * | PTRMS | 0.00094±0.00021 | 0.0011±0.00013 | 0.0008±0.00041 | | 0.002±0.0016 | 0.0018±0.00077 |
| 119.167 | $C_8H_9N$ | * | PTRMS | 0.0018±0.00018 | 0.002±8.4E-5 | 0.0016±0.00036 | | 0.0037±0.0028 | 0.0035±0.00042 |
| 120.151 | $C_8H_8O$ | methylbenzaldehyde/tolualdehyde* | PTRMS | 0.025±0.0037 | 0.024±0.0011 | 0.026±0.0073 | 0.13±0.08[2] 0.082±0.03[3] | 0.044±0.0017 | 0.058±0.0097 |
| 120.195 | $C_9H_{12}$ | trimethylbenzene/C9 aromatics* | PTRMS | 0.052±0.0061 | 0.075±0.011 | 0.029±0.0056 | 0.051±0.02[2] 0.069±0.031[3] | 0.14±0.015 | 0.064±0.0079 |
| 120.195 | $C_9H_{12}$ | Isopropylbenzene | AWAS | 0.013 | 0.011 | 0.016 | 0.013±0.025[3] | 0.025 | 0.040 |
| 120.195 | $C_9H_{12}$ | n-propylbenzene | AWAS | 0.27 | 0.30 | 0.24 | 0.0064±0.0039[3] | 0.69 | 0.59 |
| 120.195 | $C_9H_{12}$ | 1,3,5-trimethylbenzene | AWAS | 0.062 | 0.081 | 0.043 | 0.0036±0.0027[3] | 0.19 | 0.11 |
| 120.195 | $C_9H_{12}$ | 1-methyl-2-ethylbenzene | AWAS | 0.81 | 0.83 | 0.78 | | 1.9 | 2.0 |
| 121.139 | $C_7H_7NO$ | * | PTRMS | 0.00095±0.00033 | 0.00091±0.00023 | 0.001±0.00061 | | 0.0016±0.0038 | 0.0022±0.0012 |
| 122.123 | $C_7H_6O_2$ | benzoic acid/hydroxybenzaldehyde* | PTRMS | 0.02±0.0038 | 0.021±0.00058 | 0.019±0.0075 | 0.079±0.035[2] 0.065±0.023[3] | 0.037±0.0016 | 0.04±0.013 |
| 122.167 | $C_8H_{10}O$ | xylenol/C2 phenol/methylanisole* | PTRMS | 0.015±0.0018 | 0.016±0.00096 | 0.013±0.0035 | 0.11±0.037[2] 0.1±0.057[3] | 0.029±0.0027 | 0.029±0.0051 |
| 122.211 | $C_9H_{14}$ | cyclohexylallene* | PTRMS | 0.0076±0.001 | 0.0083±0.0017 | 0.0068±0.0012 | | 0.015±0.0028 | 0.015±0.0091 |



| Mass | Formula | Name | Instrument | | | | | | |
|---|---|---|---|---|---|---|---|---|---|
| 124.095 | $C_6H_4O_3$ | hydroxy benzoquinone* | PTRMS | 0.0032±0.0012 | 0.0029±0.00083 | 0.0035±0.0022 | 0.073±0.018[2] 0.045±0.026[3] | 0.0051±0.0015 | 0.0075±0.0042 |
| 124.139 | $C_7H_8O_2$ | guaiacol* | PTRMS | 0.052±0.007 | 0.051±0.0086 | 0.053±0.011 | 0.37±0.12[2] 0.27±0.17[3] | 0.091±0.019 | 0.12±0.014 |
| 124.183 | $C_8H_{12}O$ | acetylcyclohexene* | PTRMS | 0.0078±0.0011 | 0.0087±0.0015 | 0.0068±0.0016 | | 0.015±0.0021 | 0.015±0.0017 |
| 124.227 | $C_9H_{16}$ | cyclononene* | PTRMS | 0.0022±0.00033 | 0.0022±0.00048 | 0.0022±0.00046 | | 0.0039±0.0072 | 0.0048±0.00061 |
| 126.111 | $C_6H_6O_3$ | hydroxymethylfurfural* | PTRMS | 0.0096±0.002 | 0.0094±0.0031 | 0.0098±0.0027 | 0.27±0.1[2] 0.064±0.026[3] | 0.016±0.0054 | 0.021±0.0035 |
| 126.155 | $C_7H_{10}O_2$ | unsaturated C6 cyclic carboxylic acid | CIMS | 0.012±0.0053 | 0.015±0.0073 | 0.0087±0.0076 | | 0.026±0.012 | 0.019±0.016 |
| 126.155 | $C_7H_{10}O_2$ | cyclohexene carboxylic acid * | PTRMS | 0.0064±0.0019 | 0.008±0.0035 | 0.0048±0.0016 | | 0.014±0.0056 | 0.01±0.00025 |
| 126.199 | $C_8H_{14}O$ | octenone* | PTRMS | 0.0032±0.00064 | 0.0037±0.00064 | 0.0027±0.0011 | | 0.0064±0.012 | 0.0057±0.0018 |
| 126.217 | $C_7H_{10}S$ | trimethylthiophene* | PTRMS | 0.011±0.00091 | 0.016±0.0013 | 0.0054±0.0013 | | 0.028±0.0027 | 0.012±0.019 |
| 126.243 | $C_9H_{18}$ | cis,trans,trans-1,2,4-trimethylcyclohexane | AWAS | 0.0019 | 0.0022 | 0.0016 | | 0.0046 | 0.0033 |
| 126.243 | $C_9H_{18}$ | 1-nonene | AWAS | 0.00010 | 4.7E-05 | 0.00016 | | - 0.00015 | 0.00039 |
| 128.127 | $C_6H_8O_3$ | di hydroxymethyl furan* | PTRMS | 0.0044±0.0009 | 0.0059±0.0014 | 0.0029±0.0012 | | 0.01±0.0025 | 0.0063±0.0026 |
| 128.171 | $C_7H_{12}O_2$ | C6 unsaturated carboxylic acids | CIMS | 0.0091±0.0032 | 0.011±0.0031 | 0.0077±0.0055 | | 0.018±0.0048 | 0.016±0.011 |
| 128.171 | $C_7H_{12}O_2$ | cyclohexanoic acid* | PTRMS | 0.0068±0.0017 | 0.0085±0.0029 | 0.005±0.0018 | | 0.015±0.0044 | 0.01±0.00028 |
| 128.174 | $C_{10}H_8$ | naphthalene* | PTRMS | 0.017±0.0035 | 0.018±0.0037 | 0.015±0.0059 | 0.078±0.056[2] | 0.031±0.0056 | 0.031±0.0096 |
| 128.215 | $C_8H_{16}O$ | octanone* | PTRMS | 0.0034±0.00044 | 0.0039±0.0006 | 0.0028±0.00065 | | 0.0068±0.0078 | 0.006±0.0012 |
| 128.259 | $C_9H_{20}$ | 3,3-diethylpentane | AWAS | 0.0075 | 0.0055 | 0.0094 | | 0.011 | 0.022 |
| 130.187 | $C_7H_{14}O_2$ | C7 saturated carboxylic acids | CIMS | 0.022±0.0069 | 0.026±0.013 | 0.018±0.0036 | | 0.043±0.021 | 0.037±0.0097 |
| 130.187 | $C_7H_{14}O_2$ | amyl acetate* | PTRMS | 0.0031±0.00078 | 0.0034±0.0013 | 0.0028±0.00088 | | 0.0056±0.002 | 0.0058±0.0014 |
| 132.159 | $C_6H_{12}O_3$ | C6 hydroxy-carboxylic acids | CIMS | 0.0016±0.00038 | 0.0027±0.00061 | 0.00053±0.0045 | | 0.0045±0.0088 | 0.0012±0.001 |
| 132.162 | $C_9H_8O$ | methylbenzofurans* | PTRMS | 0.01±0.0016 | 0.01±0.001 | 0.011±0.003 | 0.055±0.03[2] 0.046±0.021[3] | 0.017±0.0014 | 0.021±0.037 |
| 132.206 | $C_{10}H_{12}$ | ethylstyrene/methylpropenyl benzene* | PTRMS | 0.0083±0.0012 | 0.0083±0.0016 | 0.0083±0.0017 | 0.041±0.019[2] 0.04±0.026[3] | 0.014±0.0031 | 0.017±0.014 |
| 134.134 | $C_8H_6O_2$ | phthalic acid* | PTRMS | 0.0039±0.0011 | 0.0044±0.0016 | 0.0033±0.0014 | | 0.0071±0.023 | 0.0065±0.0024 |



| | | | | | | | | | |
|---|---|---|---|---|---|---|---|---|---|
| 134.178 | C$_9$H$_{10}$O | methylacetophenone* | PTRMS | 0.0059±0.00094 | 0.0062±0.00043 | 0.0056±0.0018 | 0.053±0.031[2] 0.045±0.019[3] | 0.01±0.00064 | 0.011±0.0026 |
| 134.222 | C$_{10}$H$_{14}$ | C10 aromatics* | PTRMS | 0.024±0.0019 | 0.035±0.002 | 0.013±0.0033 | 0.043±0.022[2] 0.04±0.021[3] | 0.058±0.0022 | 0.026±0.0052 |
| 134.222 | C$_{10}$H$_{14}$ | 1,2-diethylbenzene | AWAS | 1.3 | 1.6 | 1.1 | | 3.3 | 2.6 |
| 134.222 | C$_{10}$H$_{14}$ | 1,2-dimethyl-4-ethylbenzene | AWAS | 0.063 | 0.085 | 0.042 | | 0.19 | 0.094 |
| 134.222 | C$_{10}$H$_{14}$ | 1,4-dimethyl-2-ethylbenzene | AWAS | 0.11 | 0.089 | 0.13 | | 0.15 | 0.31 |
| 136.15 | C$_8$H$_8$O$_2$ | methyl benzoic acid* | PTRMS | 0.013±0.0013 | 0.014±0.0014 | 0.012±0.0022 | 0.081±0.03[2] 0.066±0.029[3] | 0.022±0.0014 | 0.023±0.0028 |
| 136.238 | C$_{10}$H$_{16}$ | monoterpenes | PTRMS | 0.39±0.034 | 0.49±0.0094 | 0.29±0.068 | 1.53[1] 0.87±0.72[2] 0.21±0.15[3] | 0.8±0.032 | 0.57±0.13 |
| 138.122 | C$_7$H$_6$O$_3$ | hydroxybenzoic acid* | PTRMS | 0.0026±0.00061 | 0.0039±0.0012 | 0.0014±0.00028 | | 0.0061±0.017 | 0.0028±0.00035 |
| 138.166 | C$_8$H$_{10}$O$_2$ | creosol/methyl guiacol* | PTRMS | 0.0073±0.0013 | 0.0077±0.0014 | 0.0069±0.0022 | 0.26±0.077[2] 0.14±0.11[3] | 0.012±0.0019 | 0.013±0.0033 |
| 138.21 | C$_9$H$_{14}$O | isophorone* | PTRMS | 0.0092±0.0016 | 0.0086±0.0015 | 0.0098±0.0028 | | 0.014±0.0027 | 0.019±0.0039 |
| 142.286 | C$_{10}$H$_{22}$ | 3,3-dimethyloctane | AWAS | 0.078 | 0.0020 | 0.15 | | -0.071 | 0.37 |
| 146.189 | C$_{10}$H$_{10}$O | dimethylbenzofuran/ethyl benzofuran* | PTRMS | 0.0048±0.00051 | 0.0052±0.00085 | 0.0045±0.00058 | 0.043±0.018[2] 0.051±0.028[3] | 0.0078±0.011 | 0.0083±0.00035 |
| 146.233 | C$_{11}$H$_{14}$ | * | PTRMS | 0.0035±0.0007 | 0.0037±0.00059 | 0.0034±0.00013 | | 0.0057±0.011 | 0.0061±0.0019 |
| 148.117 | C$_8$H$_4$O$_3$ | benzofurandione* | PTRMS | 0.0047±0.0012 | 0.0048±0.0013 | 0.0047±0.002 | | 0.0071±0.017 | 0.0082±0.003 |
| 148.161 | C$_9$H$_8$O$_2$ | cinnamic acid* | PTRMS | 0.0024±0.00062 | 0.0026±4.2E-5 | 0.0021±0.0012 | | 0.0039±0.0015 | 0.0037±0.0019 |
| 148.205 | C$_{10}$H$_{12}$O | benzylacetone/estragole* | PTRMS | 0.0023±0.00054 | 0.0022±0.00088 | 0.0024±0.00063 | 0.027±0.012[2] 0.025±0.015[3] | 0.0033±0.014 | 0.0044±0.00074 |
| 148.249 | C$_{11}$H$_{16}$ | C11 aromatics/pentamethylbenzene* | PTRMS | 0.0041±0.00069 | 0.0043±0.00057 | 0.0038±0.00013 | 0.014±0.0078[2] 0.014±0.0074[3] | 0.0064±0.0082 | 0.0069±0.0016 |
| 150.177 | C$_9$H$_{10}$O$_2$ | ethyl benzoate/vinyl guaiacol* | PTRMS | 0.0028±0.00045 | 0.0029±0.00052 | 0.0028±0.00073 | 0.14±0.076[2] 0.036±0.025[3] | 0.0043±0.0079 | 0.0049±0.0009 |
| 150.221 | C$_{10}$H$_{14}$O | carvone* | PTRMS | 0.0021±0.00049 | 0.0027±0.009 | 0.0015±0.00039 | | 0.0039±0.012 | 0.0027±0.00048 |
| 152.149 | C$_8$H$_8$O$_3$ | methoxybenzoic acid* | PTRMS | 0.0075±0.0019 | 0.0085±0.0025 | 0.0065±0.0029 | | 0.012±0.0032 | 0.011±0.0041 |
| 152.193 | C$_9$H$_{12}$O$_2$ | ethylguaiacol* | PTRMS | 0.0027±0.001 | 0.0031±0.0018 | 0.0022±0.00098 | | 0.0044±0.024 | 0.0039±0.0014 |
| 152.196 | C$_{12}$H$_8$ | acenaphthylene* | PTRMS | 0.0032±0.0011 | 0.0041±0.0021 | 0.0022±0.00089 | 0.01±0.0066[2] | 0.0059±0.0028 | 0.0038±0.0012 |
| 152.237 | C$_{10}$H$_{16}$O | camphor/isomers* | PTRMS | 0.011±0.0015 | 0.013±0.0022 | 0.0087±0.002 | 0.027±0.017[2] 0.025±0.014[3] | 0.02±0.0033 | 0.015±0.0026 |
| 154.165 | C$_8$H$_{10}$O$_3$ | syringol* | PTRMS | 0.0022±0.00045 | 0.0026±0.00056 | 0.0017±0.0007 | 0.022±0.0078[2] 0.017±0.0067[3] | 0.0037±0.0065 | 0.0029±0.001 |





| | | | | | | | | | |
|---|---|---|---|---|---|---|---|---|---|
| 154.209 | $C_9H_{14}O_2$ | norbornaneacetic acid* | PTRMS | 0.0022±0.00083 | 0.0023±0.0011 | 0.0022±0.0012 | | 0.0033±0.0017 | 0.0038±0.0018 |
| 154.212 | $C_{12}H_{10}$ | acenaphthene* | PTRMS | 0.0029±0.00052 | 0.0033±0.00033 | 0.0025±0.00099 | | 0.0046±0.003 | 0.0042±0.0014 |
| 154.253 | $C_{10}H_{18}O$ | terpine-4-ol/cineole/isomers* | PTRMS | 0.0018±0.00065 | 0.0019±0.00084 | 0.0017±0.00098 | 0.0056±0.0021[2] 0.0027±0.0017[3] | 0.0029±0.014 | 0.0028±0.0015 |
| 204.357 | $C_{15}H_{24}$ | sesquiterpenes* | PTRMS | 0.0021±0.00032 | 0.0024±0.00045 | 0.0017±0.00045 | 0.15±0.07[2] 0.029±0.028[3] | 0.0026±0.0038 | 0.0022±0.00047 |
| 239±61 | $C_{11}$ to $C_{25}$ | I/SVOCs – $C_xH_y$ | cartridge | 1.6±0.04 | | | | | |
| 255±61 | $C_{11}$ to $C_{25}$ | I/SVOCs – $C_xH_yO_1$ | cartridge | 0.9±0.09 | | | | | |
| 271±61 | $C_{11}$ to $C_{25}$ | I/SVOCs – $C_xH_yS_1$ | cartridge | 0.1±0.003 | | | | | |
