# Peer review of "Reconciling the total carbon budget for boreal forest wildfire emissions using airborne observations 3 4 Katherine L. Hayden1\*, Shao-Meng Li2, John Liggio1, Michael J. Wheeler1, Jeremy J.B. Wentzell1, Amy 5 Leithead1, Peter Brick"

_Atmospheric Chemistry and Physics, 2022_

## Author Comment (AC1)

Dear Editor,

We thank both reviewers for their detailed and thorough review of our work, which we believe has significantly improved the paper. We have made numerous changes throughout the manuscript, and outline our responses to comments and associated manuscript edits below. Our point-by-point responses to the reviewer comments (black) are provided below as blue text, and changes to the manuscript as italicized in blue text. Our responses to reviewer #1 is followed by reviewer #2.

Thank you for considering this manuscript for publication in ACP.

**Reviewer #1**

**This paper describes extensive measurements of organic species in two plumes from one boreal wildfire (WF). There is a significant amount of detail and the paper, and its Supplementary Information, are quite long. The truly unique aspect of the paper is the total organic carbon measurement, which to my knowledge has never been done before. I think the paper is acceptable for publication once the following general and specific comments/questions are dealt with.**

**General comments;**

**The total organic carbon measurement and associated budget is one of the most interesting aspects of the paper, but there is not a lot of detail about it. I'd like to see some close-up plots of what the total signal and the $CO_2+CO+CH_4$ signals looked like inside and outside the plume. This would give a sense of what the variabilities are, and ultimately what the uncertainties are in the quantity $NMOG_T$.**

We have segregated the TC and $NMOG_T$ into it's own portion in the SI Methods section (Line 52), and it now includes more detail on this method. In addition, a figure is also included in the SI (Fig. S2) (re-numbered the subsequent figures) which shows the uncertainties when differencing the TC (with catalyst) and the $\Sigma(CO_2+CO+CH_4)$ (without catalyst). The uncertainty for both is ±60 ppbv with a resultant propagated uncertainty of ±85 ppbv in $NMOG_T$.

**I assume that the numbers presented are by Carbon, but that should be specified somewhere.**

The TC and $NMOG_T$ values are presented as ppmv C. This was already stated (now at Line 132 and 135). We also added a statement at Line 137 to 'see SI Methods for more details', and in the SI Methods, where we have now separated the TC and $NMOG_T$ into it's own paragraph (Line 52), it was also already stated that units are ppmv C (now at 53).

**Several places the authors mention the absence of emissions data on boreal WFs. Simpson et al., (2011), present emissions data from the boreal Canadian WFs and only get mentioned towards the end of the paper. There are several other places that the Simpson et al., work should be mentioned, and the results of this work could be used to comment further on some of the detailed conclusions of Simpson et al., which were limited by the fact that they had to rely on whole air cannister samples.**

Thank you, we acknowledge the importance of the Simpson et al. (2011) paper and made the following revisions to better reflect that work (and earlier in the manuscript) in the context of our measurements.

Line 20 and 88– we changed 'a lack of' to 'limited'

Line 86, added text,

*'Until now, the most complete characterization of boreal forest wildfire emissions in Canada was provided by Simpson et al. (2011) which relied on whole air canisters with offline analysis for organic compounds.'*

Line 316 added text,

*'The quantification of this suite of compounds provides new and additional emission estimates to those reported in Simpson et al. (2011) and compiled in Andreae (2019) for the boreal forest ecosystem. Several sulphur-containing compounds and a few other VOCs were not detected (Table S6), and although not part of the measurement suite in the present study, Simpson et al. (2011) did not observe emissions of anthropogenic halocarbons from wildfires in the same boreal forest ecosystem.'*

At Line 349, added another reference to Simpson et al. (2011)

Line 364, added text, (and removed the sentence at Line 368 as it did not fit into the context)

*'For example, Simpson et al. (2011) found a similar distribution of compound classes with 57 measured NMOG species, based on discrete canister samples, in boreal forest wildfires. In that study, oxygenates (non-aromatic) comprised a smaller portion of NMOG (29 %) as major emitted species like acetaldehyde and acetic acid (Fig. 8) were not included.'*

**Specific Comments**

**Lines 25-27. The description of the $NMOG_T$ budget appears incorrect. As it reads, the 7.4% attributed to I/SVOCs was part of the 46.2% NMOG, when later on in the paper it reads that the 7.4% is in addition to the NMOG. That would make sense because 100-46.2-7.4 is the 46.4% that the paper says in unidentified. In addition, these numbers should have uncertainties.**

Thank you, this was an error. We have revised our $NMOG_T$ budget to reflect a range of I/SVOC emission estimates related to the I/SVOC background, and discussed in detail below and in response to Reviewer #2's comments. The revisions resulted in ΣNMOG is **50±3 to 53±3 %** of $NMOG_T$ which includes the 7 to 10 % range from I/SVOCs (revised Fig. 6). This has been adjusted and clarified in the abstract at Line 26-27, in Sect. 3.3.3 specifically at Line 462-463.

**Lines 77-80. The Simpson et al., 2011 work should be mentioned here.**

As the Simpson et al. (2011) work is included in the Andreae (2019) literature review paper of boreal forest emission factors, it wasn't referenced explicitly here. However, as indicated earlier we added some text/reference to the Simpson et al. (2011) paper at Line 86,

*'Until now, the most complete characterization of boreal forest wildfire emissions in Canada was provided by Simpson et al. (2011) which relied on whole air canisters with offline analysis for organic compounds.'*

**Lines 116-117. Here it would be good to have a bit more information on the catalyst system, e.g. amount of Pt, geometry, flow rate through the catalyst.**

More details on the TC method including information on the catalyst system is now provided in the SI in a new paragraph on TC and NMOG$_T$ at Line 52.  See next response.

**Lines 119-120. As noted above, this method is worth describing in more detail, perhaps with its own section in the SI.**

More details on the TC method, including a figure (Fig. S2) is now provided in the SI in a new paragraph on TC and NMOG$_T$ at Line 52 with text as,

*'**TC and NMOG$_T$**   A second Picarro G2401-m instrument was used to measure total carbon (TC, in units of ppmv C) by passing the sample air through a heated (650 °C) platinum catalyst (Shimadzu), adapted from Stockwell et al. (2018) and Veres et al., (2010) which converted all carbon species to $CO_2$. Approximately 4 g of platinum catalyst (https://www.elementalmicroanalysis.com/product_details.php?product=B1605&description=High%20Sensitivity%20Catalyst%20630-00996) was enclosed in a resistively heated ½" O.D. x 12" long stainless steel tube.  As the catalyst assembly was mounted on the roof exterior to the aircraft, with no unheated portion of inlet, TC losses were expected to be negligible.  NMOG$_T$ mixing ratios in units of ppmv C were quantified by subtracting the ambient $CH_4$, CO and $CO_2$ measurements (instrument without the catalyst) from the TC measurements (Fig. S2).  Calibrations using two different mixing ratio standards of CO, $CO_2$ and $CH_4$, traceable to NOAA GMD standards, were performed for both Picarro instruments during flight (at the beginning and end) to assess instrument drift and sensitivity.  No significant drift was observed during each flight.  NMOG$_T$ was averaged to 10 sec (from the 2 sec native time resolution) to increase the signal to noise ratio.  The uncertainty of each instrument was assessed in flight by overflowing the inlet with a constant flow of calibration gas or an ultra pure nitrogen gas stripped of $CO_2$ via two NaOH pellet traps in series (https://www.sigmaaldrich.com/CA/en/product/supelco/503215).  In both cases, this resulted in an uncertainty of approximately 60 ppbv C at the 3σ level for each TC channel (dominated by the precision of the $CO_2$ measurement) (Fig. S2).  These uncertainties were added in quadrature resulting in a 3σ uncertainty of ±85 ppb C for NMOG$_T$.  Laboratory experiments indicated that the conversion efficiency of ethane across the catalyst was ~100 %, which is expected to be the most challenging species to combust aside from methane, which is concurrently measured.  Additional laboratory experiments using a range of hydrocarbons ($>C_2$) including aromatics also exhibited ~100 % conversion efficiency (Li et al., 2021; Li et al., 2019).  The catalyst material was changed after approximately every 5 flights to further ensure minimal changes in efficiency.'*

**Lines 136-138. The lack of coverage of N-containing compounds in this method is an issue and certainly bears on whether or not these measurements can be considered comprehensive. There are certainly significant N-containing I/SVOCs in WF emissions, see for example Tomaz, et al., (2018).**

In the manuscript, we acknowledged that we were not able to quantify N-containing I/SVOCs because of a lack of available standards (now at Line 160 in the SI), but we do not suggest that these measurements were comprehensive.  In fact, we had discussed in the manuscript that the unidentified carbon mass likely represents unmeasured I/SVOCs i.e. at Line 470, *'...Nevertheless, a portion of the unidentified species likely consisted of challenging-to-measure-VOCs and larger I/SVOCs that were highly functionalized or contained molecular features like reduced nitrogen groups (e.g. amines)...',* and that improved speciation of these lower volatility compounds is needed (Line 500).  With this adsorbent tube method, we observed a number of N-containing species (eg. Ditto et al. 2021 Figure S5B and S5D and S11B), however, they are not quantifiable yet, due to the lack of comprehensive authentic standards to convert

signal to mass concentration. The Tomaz et al. (2018) paper does not appear to quantify N-containing I/SVOCs in wildfire emissions. We further address this comment on nitrogen-containing compounds in a subsequent response.

**Line 169. Should be "depositional"**

This sentence has been removed. We re-worded and moved this discussion to a new section 2.5 Emissions Uncertainties at Line 257.

**Line 214. I don't think Liu et al., 2022 is referenceable as a paper since it hasn't even been submitted. It would be a personal communication.**

We removed the Liu et al (2022) reference in the reference list, and all references to it in the manuscript.

**Line 255. Here is a place where more details on the NMOG$_T$ measurement would be helpful. Was 100ppbv the detection limit for the NMOG$_T$ measurement?**

As indicated in a previous comment, we have added more information in the SI at Line 52 related to the NMOG$_T$ measurement. Additionally, the detection limit is now shown in Table S1 as 85 ppbv C at the $3\sigma$ level at 10 sec. We also changed 100 ppbv to 375 ppbv for the background to match with the average background identified in Table A1.

**Line 280-281. I assume these numbers are as Carbon, not by weight?**

At Line 352, it is indicated that these numbers in Figure 5 are mixing ratios.

**Section 3.3.2 This section needs to address the effect of not measuring N-containing species.**

We added some text at Line 341 to further address N-containing organic compounds that were likely not measured with our instrumentation suite.

*'Nitrogen-containing organics were detected in the present study totalling 3.9 ppbv and 18 % of ΣN$_r$ (Fig. 4), however, other such compounds that were not included with the instrument suite used in this study were also likely emitted. Such compounds could include organic nitrates, amines, amides, heterocyclic compounds, nitriles and nitro compounds that have been found in biomass burning emissions (Roberts et al., 2020; Lindaas et al., 2020; Andreae 2019; Koss et al., 2018; Tomaz et al., 2018; Stockwell et al., 2015).'*

We also added text at Line 479 to emphasize that the unidentified portion of NMOG$_T$ could be comprised of nitrogen-containing organics and possible estimates of that portion.

*'The unidentified portion may also have been comprised of nitrogen-containing organics (Sect. 3.1). Studies that included measurements of a larger range of nitrogen-containing organics in biomass burning emissions estimated that they comprised < 5-6 % of the total nitrogen budget (Lindaas et al. 2020; Gilman et al., 2015), and thus, an even smaller fraction of NMOG$_T$.'*

**Lines 341-344. What is the impact of the lack of background samples on these calculated ERs? Did you correct for CO outside the plume? Does this mean you essentially assume the VOC concentrations outside the plume are zero?**

To address this point and correct for the influence of I/SVOC background concentrations, we selected the cartridge collected during the upper 2 transects, which minimally intercepted the plume (average CO mixing ratios in these transects were near background levels, ~0.14 ppmv) (see Table A1, CO background average 0.119±0.005 ppmv), and we treated this as a background sample. Since there was some slight plume interception during these transects, this sample represents a contaminated background, and thus provides a conservative lower limit of I/SVOC concentrations when subtracted from the cartridge sample collected during the lowest 3 transects. We also present, as originally shown, the EF and ER values when no background sample is subtracted (i.e., the background I/SVOC concentrations are assumed to be zero), which represents an upper limit to the I/SVOC concentrations. Text at Line 219 reflects these changes,

*'For the integrated cartridges, samples were collected over the lower set of aircraft transects ('LOW') and higher set of transects ('HIGH'), resulting in two integrated cartridge samples for each screen. The HIGH sample was used as the background. The HIGH sample was collected largely outside the wildfire plume, but may have been influenced to some extent from emissions. However, this impact is expected to be minimal as average CO mixing ratios during the HIGH sample were at background levels (~0.14 ppmv). Nevertheless, to address the potential for influence of the plume in the HIGH sample, the ERs are presented as ranges with the lower estimates derived by subtracting the HIGH background sample, and the upper estimates without subtracting the HIGH sample. This calculation is described in Eq. 2 where Cartridge$_{LOW}$ represents the LOW cartridge sample measurements, Cartridge$_{BKGD}$ is the background derived from the HIGH cartridge sample measurements, and CO$_{LOW}$, CO$_{BKGD}$ are the average CO concentrations during the respective LOW and HIGH cartridge integration time periods. The uncertainty with this bounding analysis is acknowledged, but the I/SVOCs ERs within a plume are likely to vary similar to other work (Hatch et al., 2018).*

$$ER = \frac{Cartridge_{LOW} - Cartridge_{BKGD}}{CO_{LOW} - CO_{BKGD}} \ to \ \frac{Cartridge_{LOW} - 0}{CO_{LOW} - CO_{BKGD}}$$ '

**Line 347. What is the nature of this variation in sampling efficiency? How large was it, what caused it?**

We note that the concentrations estimated for the cartridge samples may be sensitive to variations in sampling efficiency within the under-wing sampling pod across $C_{10}$-$C_{25}$, though these effects are expected to be minimal for the QBTX adsorbent tubes used in this study, where breakthrough testing with similar sampling conditions by Sheu et al. (2018) determined that analyte trapping efficiency in this carbon number range was generally greater than 85% (Sheu et al., 2018). We added text at Line 436,

*'...though these effects are expected to be minimal for the adsorbent tubes used in this study (Ditto et al., 2021; Sheu et al. 2018)'*

Text is also added in the SI at Line 164,

*'Prior breakthrough testing with QBTX adsorbent tubes and similar sampling conditions to those used in this study showed that analyte trapping efficiency in the same carbon number range was generally greater than 85 % (Sheu et al., 2018).'*

Sheu et al reference: https://doi.org/10.1016/j.chroma.2018.09.014

**Line 367-368. These numbers need to have uncertainties and a description of how they were arrived at.**

We revised the I/SVOC EFs EF and ER estimates to reflect a low and high range as described in a previous response (and in responses to Reviewer #2). We present the low/high ranges in Fig 6 (and Fig.

S10 (was Fig. S8)) representing the uncertainties in the I/SVOC values. We also revised our uncertainties for all of our other measurements, and updated Fig. 6 to reflect the revised percent contributions to NMOG$_T$ and their uncertainties. Our revised uncertainties are described in detail in responses to Reviewer #2, and at Line 257 in an added section, 2.5 Emission Uncertainties, and reads as,

*'2.5 Emissions Uncertainties*
*There is the potential for inherent uncertainties using a plume integration method for calculating EFs and ERs as the ratios derived this way represent the average plume composition and ignore the spatial heterogeneity in wildfire plumes (Palm et al., 2021; Decker et al., 2021; Garofalo et al., 2019), chemical transformation processes, and can also be affected by changing background levels. Pollutants released by wildfires can be influenced by photochemical and physical changes that may take place between the time of emission and the time of measurement, particularly for more reactive compounds (e.g. Palm et al., 2021; Lindaas et al., 2020; Peng et al., 2020; Akagi et al., 2011). Although controlled laboratory studies are well suited to examine direct emissions with minimal aging, they cannot reproduce realistic burning conditions. Field measurements are critical to understand emissions that are impacted by factors such as complex burning dynamics, fuel moisture, temperature and winds (Andreae 2019). Recognizing the challenges of measuring primary emissions by aircraft, at 10 km (<1 hr) away from the fire source, Screen 1 measurements represent some of the freshest emissions measured under wildfire conditions, thus providing best estimates of initial conditions.*

*Uncertainties in the EFs and ERs are estimated by summing in quadrature the standard error of the average EF (or ER) and the propagated measurement uncertainties. The standard error is used as description of the uncertainty on the average EF (and ER) characterizing repeated transects across the SP and NP for a total of 20 min of in-plume sampling. The standard error is expected to at least partially capture uncertainties associated with plume aging and vertical plume heterogeneity. As many compounds exhibited significant in-plume enhancements above background levels, uncertainties in the integrated ΔX, ΔCO and ΔTC values were assumed to be dominated by instrumental (measurement) uncertainties (Table S1, S2). Emissions are not reported for compounds where the average mixing ratios were within 1σ of the background average. The low and high I/SVOCs EFs (and ERs) are provided as estimates of their uncertainties (as described in Sect. 2.3). The derivation of AWAS and cartridge EFs (and ERs) may have potential limitations as they rely on a limited number of samples, with the potential of the AWAS discrete samples capturing only part of a plume.'*

**Line 442. These numbers have too many significant figures, 3 would be the most that are warranted.**

Agreed. At Line 523-526 significant figures are fixed.

**Line 531. This is confusing, doesn't similar average MCEs imply similar fires stages were sampled?**

The average MCE between the Simpson et al. (2011) of 0.89 closely matches our study. Despite the similar average MCEs, the Simpson et al. (2011) measurements were taken near large and active wildfires during the mid day, while our measurements reflect a smoldering surface fire taken at 9am LT. At Line 659 we removed 'with similar average MCEs'. At Line 672, we added,

*'...combustion state, despite having similar study-averaged MCEs.'*

At Line 679, we had indicated that the Simpson et al. (2011) study sampled earlier and more intense fire stages.

**Lines 647-648. This doesn't make any sense; these ERs differ by almost a factor of 10.**

There was an error in the ER and it has been corrected at Line 817 to read as:

*'For the two satellite overpasses during the flaming phase of the fire, the $R_{CO/FRP}$ values were within the uncertainties (19:06 UTC $R_{CO/FRP}$ =0.47±0.25 t $h^{-1}$ $MW^{-1}$; 20:48 UTC $R_{CO/FRP}$ = 0.43±0.23 t $h^{-1}$ $MW^{-1}$).'*

**Line 651. I think at most 3 significant figures are warranted here.**

Agreed, and we now include uncertainties. We have changed the text at Line 813 to:

*'If the fire is assumed to have continued burning when GOES did not detect any fire hot spots (between 22:00 - 04:00 UTC and 07:00 - 15:00 UTC, with an FRP of 150 MW (~GOES detection limit; Roberts et al., 2015), the emissions increase to 24,000±9600, 106±43 and 98±39 tonnes, respectively, providing an upper limit of emissions.'*

**Figure 8b, Did you really measure peroxynitric acid? Could this be an artefact due to $IO^-$ ions reacting with $HNO_3$ in the IMR?**

We agree that the signal could be due to $(I)HO_2NO_2^-$ and/or $(IO)HNO_3^-$. However, we have confirmed that in order to generate $(IO)HNO_3^-$ in our instrument configuration, we require >100 ppbv of $O_3$ and tens of ppbv's of $HNO_3$ present. At the $O_3$ and $HNO_3$ levels we measured in these plumes (<100 ppbv and <0.5 ppbv $HNO_3$, respectively), we expect the signal is largely due to $(I)HO_2NO_2^-$. No changes made to the manuscript.

**Table A1. This is a long table. It would be helpful if there was a heading bar on each page to make it easier to read the details.**

Headers are now at the top of each page for Table A1.

**Supplementary Information**

**Lines 40-41. Wouldn't the heated inlet line for $NH_3$ volatilize $NH_4NO_3$? How might that impact the measurements?**

The inlet for $NH_3$ is mounted in rear-facing (SI Line 23) which minimizes the sampling of particles. We added text at SI Line 24,

*'The rear-facing inlet minimizes the sampling of particles'.*

Although particulate $NH_4$ and $NH_3$ abundances were of similar magnitude, and an equilibrium at 90°C would strongly favour the gas phase, the rate constants for evaporation/volatilization ($NH_4NO_3 => NH_3 + HNO_3$) are quite slow (> minutes) even at these temperatures (Makar et al., 1998), compared to the sampling time from the inlet to the $NH_3$ instrument which was on the order of 0.25 seconds. Therefore we are confident that volatilization of $NH_4NO_3$ made no significant contribution to the measured $NH_3$ concentrations. No other change made in the manuscript.

Makar, P. A., Wiebe, H. A., Staebler, R. M., Li, S. M., and Anlauf, K. (1998), Measurement and modeling of particle nitrate formation, *J. Geophys. Res.*, 103( D11), 13095– 13110, doi:10.1029/98JD00978.

**Line 60. The Li et al., references are not in the SI ref list.**

Added Li et al. (2019) and Li et al. (2021) references to the SI.  Removed Li et al. (2019), Li et al. (2021) and Li et al. (2017) from the manuscript as they were not referenced.

**Line 82. Do you mean it provides a stronger signal at higher masses to provide a better mass calibration?**

The permeation tube with trichlorobenzene was used during sampling to calibrate higher masses because we needed a signal that was constant and measurable during all sampling periods.  When the instrument is sampling zero air, the catalyst removes VOCs from the air and it is difficult to detect and identify masses in the higher mass range.  This adds uncertainties in the mass calibration when switching between zero and ambient air.  Adding the permeation tube to the sample flow provides us with a constant mass signal that can be used for mass calibration while the instrument is sampling both zero and ambient air.

We made a minor change to the text at Line 97 as follows:

*'A permeation tube with 1,2,4-trichlorbenzene was placed at the inlet to improve the accuracy of the mass calibration for higher masses.'*

**Line 188. Does the 3000 particle/s limit bias the measurements?**

The concentration limit on the UHSAS is not expected to bias the results. The UHSAS data were used to determine the collection efficiency of the AMS measurements. While we expect the collection efficiency to be composition dependent we do not anticipate a significant change in collection efficiency between the highest concentration periods within the plume where UHSAS data were unavailable and the other periods in the plume. As the AMS data shows that the aerosol composition is dominated by organic aerosols throughout the entirety of the plume, the collection efficiency determined when there was valid UHSAS data is expected to be the same as that when no UHSAS data were available. The missing UHSAS data accounts for approximately 7% of the data collected in the first screen of the flight.  No change made to the manuscript.

**Line 192. Is this a composition-weighted proportional density?**

Yes.  This has been added to the text at Line 191,

*'Total mass was calculated from the UHSAS measurements based on the composition-weighted proportional density determined from the AMS*

**Figure S1. It is impossible to read much of the text on the map.**

The map in Fig. 1 has been improved so that the text is much clearer, including when zooming in.

**References**

I.J. Simpson, S. K. Akagi, B. Barletta, N. J. Blake, Y. Choi, G. S. Diskin, A. Fried, H. E. Fuelberg, S. Meinardi, F. S. Rowland, S. A. Vay, A. J. Weinheimer, P. O. Wennberg, P. Wiebring, A. Wisthaler, M. Yang, R. J. Yokelson, and D. R. Blake, Boreal forest fire emissions in fresh Canadian smoke plumes: $C_1$-$C_{10}$ volatile organic compounds (VOCs), $CO_2$, CO, $NO_2$, NO, HCN and $CH_3CN$, Atmos. Chem. Phys., 11, 6445–6463, https://doi.org/10.5194/acp-11-6445-2011, 2011.

Tomaz, S., T. Cui, Y. Chen, K.G. Sexton, J.M. Roberts, C. Warneke, R.J. Yokelson, J.D. Surratt, and B.J. Turpin, Photochemical cloud processing of primary wildfire emissions as a potential source of secondary organic aerosol, *Environ. Sci. Technol.,* 52, 11027-11037, doi.org/10.1021/acs.est.8b03293, 2018

**Reviewer #2**

**This paper uses airborne measurements of 250 compounds from 15 instruments to analyze the total carbon budget from a boreal forest wildfire that was sampled on June 25, 2018. The results were also used to derive emission factors (EFs) and compare with satellite observations and modelled emissions.**

**Analyzing and presenting such a large suite of measurements is challenging and I appreciate the work the authors have done. However there are still a large number of issues to address, including rigorous uncertainty analysis. Several results are given to 0.1% or to 5 sig figs without error bars. The AWAS measurement uncertainty is 40%, but AWAS EFs don't have error bars. Some EFs differ from the literature by 40x or more, which needs much more discussion. HONO and the butanes seem to have a measurement issue, and isoprene could have furan interference. Intercomparisons for compounds measured by multiple instruments should be presented.**

We have rigorously reviewed our data and our uncertainty analyses for all of our measurements and provide details in subsequent responses. Although our measurements go through rigorous QA/QC procedures, we have nonetheless re-assessed our data to confirm their robustness as per the reviewer's suggestions. We now provide uncertainties on the AWAS EFs and ERs which were inadvertently left off. We have added additional text to the manuscript to discuss differences from the literature, with specifics outlined in subsequent responses. We also now provide intercomparisons for those compounds that can be directly compared between the PTRMS and CIMS and between the PTRMS and AWAS.

Specific to the intercomparisons, an intercomparison for isoprene measured by the PTRMS and the AWAS is shown in Fig. S3a with additional text in the SI Sect 1.1.4 Overlapping compounds (at Line 256) as,

*'Comparisons between these two methods are challenging due to the influence of isomers in the PTRMS signal, and the fact that a number of PTRMS compounds are determined using calculated sensitivities (i.e. not directly calibrated with a standard) with estimated uncertainties of 50 %. These factors limit a comparison between the AWAS and PTRMS for isoprene ($C_5H_8$) which is shown in Fig. S3a. The comparison for Screens 1-3 shows good agreement with an $r^2=0.87$. When including only Screen 1 data, there are two data points (in the SP) where the PTRMS is a factor of 2.5 to 3 higher than the AWAS, resulting in a lower $r^2=0.23$. Although the PTRMS isoprene signal is known to have interferences from cycloalkanes, these compounds are not expected to be emitted from wildfires. 2-methyl-3-buten-2-ol (MBO) produces a fragment at m/z 69.070 that is not separated in the PTRMS, and can also interfere with the isoprene measurement. We do not have measurements to confirm the impact of MBO on the isoprene signal. However, Permar et al. (2021) reported that PTRMS derived isoprene measurements during their study were approximately 2x higher than the AWAS isoprene while sampling smoke, but MBO which was measured during that study was considered too low to account for the higher than expected isoprene. In the present study, it is possible that there were contributions from other unknown isomers to the PTRMS signal in the fresh smoke plumes along Screen 1. Due to these uncertainties, and the comparatively fewer in-plume AWAS samples, EFs and ERs for isoprene are reported from both the PTRMS and AWAS, and isoprene from the PTRMS was used in the carbon budget (Table S4).'*

Intercomparisons are also provided for overlapping compounds between the PTRMS and CIMS (Fig. S3b, c and d). Text has been added at SI Line 294 as,

*'Although comparisons of exact masses between the PTRMS and CIMS are complicated because of the influence of isomers in the PTRMS signal, four exact masses were identified as the same compound*

*including acetic acid, acrylic acid, formic acid, and isocyanic acid. Figure S3 shows a comparison for the first three compounds, but excludes isocyanic acid as this compound can hydrolyze in the PTRMS drift tube. The CIMS provided measurements of pyruvic acid, but the PTRMS signal at the same mass is likely affected by inlet line losses. Acetic acid and acrylic acid show good agreement with $r^2 > 0.8$ with the PTRMS in-plume measurements ~20 % higher than the CIMS (Fig. S3a, b); this is likely due to additional contributions to the PTRMS signal at these respective exact masses. The comparison for formic acid (Fig. S3c) is poor ($r^2 =0.3$) likely because the PTRMS measurements are noisy and have a high detection limit of 2 ppbv, whereas the CIMS detection limit is 0.097 ppbv. The CIMS measurements were also directly calibrated (Table S2), whereas the PTRMS formic acid sensitivity (and other compounds) were calculated, and as such, the CIMS measurements for the overlapping compounds were retained for analysis (Table S5).'*

Note that we have removed the PTRMS-derived acetic acid fragment at m/z 42.037 in Table A1 and report only the CIMS-derived value at m/z 60.052. We also corrected an entry in Table S5 at m/z 60.052 as the PTRMS at this mass is the parent ion, not a fragment; removed 'fragment' and the Table caption related to this.

**Another issue is that EFs for I/SVOCs were done without background subtractions, which are a fundamental part of the calculation. If there aren't suitable background values, the calculations shouldn't be presented as EFs. This also impacts the 7.4% I/SVOC contribution presented in the abstract. Specific comments are given below.**

As also described in a subsequent response, to address this point, and correct for the influence of I/SVOC background concentrations, we provide a range of EF and ER estimates reflecting low and high limits. We selected the cartridge collected during the upper 2 transects (higher altitudes), which minimally intercepted the plume (average CO mixing ratios at the higher altitudes) were near background levels, ~0.14 ppmv) (see Table A1 for average background CO levels of 119±5 ppbv), and we treated this as a background sample. Since there was some slight plume interception during these transects, this sample represents a contaminated background, and thus provides a conservative lower limit of I/SVOC concentrations when subtracted from the cartridge sample collected during the lowest 3 transects. We also present, as originally shown, the EF and ER values when no background sample is subtracted (i.e., the background I/SVOC concentrations are assumed to be zero), which represents an upper limit to the I/SVOC concentrations. EF and ER estimates use the background-subtracted average CO (Eq 2) and TC (Eq 3) mixing ratios (converted to mass concentrations), respectively, across the cartridge sampling time period. Text at Lines 218 reflects these changes. The percent contribution of I/SVOCs are now 7 % to 10 % representing the low and high estimates, and revised in Fig. 6, and in the manuscript.

**L25: Here and throughout, please use error bars. What is the uncertainty on 46.2%? With >200 compounds and multiple instruments, the uncertainties are sure to be considerable.**

We have revised our EF and ER uncertainties for all of our measurements as discussed in detail in a subsequent response. We now provide uncertainty estimates for the percent contributions to $NMOG_T$ and Fig. 6 and Sect. 3.33 has been updated accordingly.

**L26: If I understand well, 46.2 + 7.4 + 46.4% = 100%, so it's confusing to say 46.2% … 'of which' 7.4% are I/SVOCs. Please clarify.**

Thank you, this was an error. The ΣNMOG is now estimated as a range (reflecting the I/SVOC estimates) of 50±3 % to 53±3 % of $NMOG_T$ which includes the low and high I/SVOC estimates of 7 and 10 %, respectively. We have updated our $NMOG_T$ budget to reflect the range of I/SVOC EFs now

presented and their percent contribution, as well as our revised VOC uncertainty estimates. Changes were made accordingly in the abstract, Fig. 6, and in Sect. 3.3.3, as well as in the SI Fig S10.

**L112-114: The GEM measurements are described using the same sentence here and in the SI. The manuscript is already long and the SI should supplement the main paper, not repeat it. Same comment on L114-116, L118-120, etc. Please check the rest of the Methods and SI for this.**

Repetition in describing the measurements between the manuscript and SI was checked and modified as indicated below.

- In the SI at Line 25, we have removed the first sentence and '..and GEM' in the heading.
- At Line 37 in SI, removed the sentence about GEM.
- Moved sentences at Lines 125-128 in main manuscript to the end of the NO, $NO_2$ and $SO_2$ section of the SI (Line 39).
- At Line 43 in SI, removed first sentence.
- At Line 77 in SI, removed part of CIMS description.
- At Lines 91 and 100-103 in SI, removed part of PTRMS description.
- At Line 107 in SI, removed part of AWAS description.
- Added some additional description of the AWAS at Lines 138-142.
- At Lines 144-149 and Lines 153-156 in SI, removed some cartridge description.
- Added some more description of the cartridges at Lines 145-146, Line 156, Lines 160-161, and Lines 163-166.
- At Line 202-205 in SI, removed part of BC description.
- At Line 216 in SI, removed part of UHSAS description.

**L143: In order to compare $PM_1$ from this study with $PM_{2.5}$ EFs from other studies, the aerosol mass between 1 and 2.5 μm was estimated to be 10% (SI L170-173). What is the uncertainty in this estimate? In Table A1, does this mean that the $PM_1$ EF from the literature is actually $PM_{2.5}$? (I see later on L442 that it does.) This should at very least be clearly stated in the caption, but I'm wondering what the value is of forcing this intercomparison.**

Upon further review, our ability to estimate the difference in mass between $PM_1$ and $PM_{2.5}$ from our measurements directly is not appropriate. We have therefore, rewritten the SI at Line 196, as shown below.

*'The $PM_1$ EF from this study is compared with the $PM_{2.5}$ EF from the Andreae (2019) literature review of boreal forest wildfire studies. While these measurements represent PM over different size ranges (<1μm vs <2.5μm), the difference is not expected to be significant based on typical size distributions of wildfire emissions (Andreae, 2019; Reid and Hobbs, 1998; Reid et al., 2005). This approach has been used previously in literature reviews of EFs for PM from wildfire emissions (Andreae, 2019; Akagi et al., 2011). '*

In Table A1, the PM EF is for $PM_{2.5}$ from Andreae (2019), but we have also now included the $PM_1$ EF from Liu et al. (2017). We have included the following text in the Table A1 caption to be more explicit, *'The Andreae (2019) PM EF represents $PM_{2.5}$.'*

Despite the difference in aerosol size, based on the estimate of aerosol size distribution from wildfire emissions, and that this approach is taken in other literature reviews of PM EFs, we consider such a comparison useful. There are few measurements of PM in wildfires (or even lab studies) reported in

general, especially for boreal forest ecosystems, and what is reported exhibits a large range of values i.e. Andreae (2019), $18.7\pm15.9$ g kg$^{-1}$, n=5.  Placing our measurements into the context of these previous measurements continues to show the large variability in PM emissions.  We revised and added to the text in Section 3.4.1 Particle species, at Line 524,

*'The PM$_1$ EF of $6.8\pm1.1$ g kg$^{-1}$ (accounting for estimated mass differences due to particle diameters (SI Sect. 1.1.2)) falls in the lower end of the large range previously observed for boreal forest wildfires ($18.7\pm15.9$ g kg$^{-1}$; Fig. 9b).  The few PM EFs for BFF19 (n=5) over a limited range of MCEs (i.e. 0.89 to 0.93) shows significant variability consistent with previous work (Jolleys et al., 2015; Akagi et al., 2011; Cubison et al., 2011; Hosseini et al., 2013).'*

We also expanded the discussion in this section on comparisons of OA EFs with literature which addresses a subsequent comment.

**L168: How many AWAS samples were taken in Screen 1, which was used to determine emissions? How many plume samples and how many background? I'm guessing the sample size was small. How does this impact uncertainty in the emission calculations?**

There were 3 AWAS samples take in the SP (transect 1, 2 and 3) and 2 AWAS samples taken in the NP (transect 2 and 3).  There were 2 background samples (transect 4).  These samples were used to derive emissions.  To be more explicit, text was added at Line 216,

*'ERs for the AWAS compounds were determined using the average mixing ratio of 3 samples taken in the SP and two in the NP, and the average mixing ratio of two background samples.  CO mixing ratios were averaged across the AWAS sample time period.'*

Text at Line 246 was also added,

*'The EFs for the AWAS and the cartridge samples were derived using the average measurements as discussed for the ER, but with TC as the denominator.'*

A separate section on Uncertainties (2.5 Uncertainties) at Line 257 was added.  In this section, text was included to acknowledge the limited number of AWAS samples which may have potential limitations, at Line 280,

*'The derivation of AWAS and cartridge EFs (and ERs) may have potential limitations as they rely on a limited number of samples, with the potential of the AWAS discrete samples capturing only part of a plume.'*

**L170: Please further discuss losses (photochemistry, deposition) in the 42 mins (SP) and 72 mins (NP) from emission to sampling. Even if they're the some of the freshest emissions, losses over this time (and the uncertainty it introduces) should be discussed, especially for the most reactive species.**

We expanded our discussion of the impacts of photochemistry and deposition (physical changes) on the derivation of emissions and address the uncertainties that these processes can introduce, including references that highlight the issue of photochemistry and losses between emission and time of measurement.  Text has been included in an added section, 2.5 Emissions Uncertainties at Line 257,

*'There is the potential for inherent uncertainties using a plume integration method for calculating EFs and ERs as the ratios derived this way represent the average plume composition and ignore the spatial*

*heterogeneity in wildfire plumes (Palm et al., 2021; Decker et al., 2021; Garofalo et al., 2019), chemical transformation processes, and can also be affected by changing background levels. Pollutants released by wildfires can be influenced by photochemical and physical changes that may take place between the time of emission and the time of measurement, particularly for more reactive compounds (e.g. Palm et al., 2021; Lindaas et al., 2020; Peng et al., 2020; Akagi et al., 2011). Although controlled laboratory studies are well suited to examine direct emissions with minimal aging, they cannot reproduce realistic burning conditions. Field measurements are critical to understand emissions that are impacted by factors such as complex burning dynamics, fuel moisture, temperature and winds (Andreae 2019). Recognizing the challenges of measuring primary emissions by aircraft, at 10 km (<1 hr) away from the fire source, Screen 1 measurements represent some of the freshest emissions measured under wildfire conditions, thus providing best estimates of initial conditions.'*

We also added text at Line 274,

*'The standard error is expected to at least partially capture uncertainties associated with plume aging and vertical plume heterogeneity.'*

We moved text at Line 185-188 to 2.5 Emissions Uncertainties as it fits into the discussion on uncertainties.

More specifically, we discuss the potential impacts of photochemistry on EFs for specific organics in subsequent comment related to comparisons of EFs between the NP and SP.

**L179: Please add detection limits to Table S1, and plume and background concentrations to Table A1, so we can see their values and how they compare. What percent of background data were below detection and how was this handled for the background subtraction?**

We added average in-plume and background mixing ratios (concentrations) to Table A1. Detection limits are added to Table S1 and S2 (for the CIMS). The detection limits are reported as three times the standard deviation of the blank/zero. Table A1 shows the Screen average in-plume mixing ratios (concentrations) based on the start/stop times used for the compound-specific integrations. The background values are selected for a time period outside and away from the plumes prior to the start of the first transect for all compounds. Note that background averages specific for the integrated values from which the EFs and ERs were determined were selected at the same altitude as the plume transect, typically on the farthest edges of the track distant from the plumes, as indicated at Line 198. All of the QC/QA'd data was included in the analysis, including data below the detection limit for the determination of background averages. We did not remove any data, nor did we replace values below the detection limit. The percent of background data that were below the detection limit varied by compound; we calculate that 23 % of our measured compounds had detection limits that were greater than the background average plus 3σ. The background averages spanned across ~2 to 3 min of 1 sec data (or 10 sec for the AMS and NMOG data), that would have a resulting standard error that is lower than the single point 3σ detection limit.

**L194: Please quantify 'well above', 'co-varied well' and 'majority of measurements.' At this point we still don't know what the 250 compounds are (Table A1 isn't mentioned until L342).**

We quantified 'well above' and removed 'co-varied well with the majority of measurements'. At Line 212, we modified the text to

*'In this study, ERs were calculated using CO as it was enhanced above a background of ~0.159±0.020 ppmv for the plumes measured, there were no other significant CO sources in the study area, and CO is a particularly good tracer for smoldering fires (e.g. Simpson et al., 2011).'*

At the beginning of Section 3.2 General plume features, at Line 313, we now introduce the compounds measured referencing Table A1,

*'Table A1 shows mixing ratios (or concentrations) and background levels of 193 pollutants that were enhanced in the fire plumes.'*

**L326: Later on L342 we learn that there are no I/SVOC background values for background subtractions, which ERs also require (Eq. 1). So what was done for the ER calculations in Table S7?**

In our revisions, we now provide a range of ER (and EF) estimates reflecting lower and upper limits.  We selected the cartridge collected during the upper 2 transects to use as a background sample.  This sample represents a slightly contaminated background, and thus provides a conservative lower limit of I/SVOC concentrations when subtracted from the cartridge sample collected during the lowest 3 transects.  We also present, as originally shown, the ER values when no background sample is subtracted.  ER estimates use the background-subtracted average CO (Eq 2) mixing ratios (converted to mass concentrations) across the cartridge sampling time period.  Text at Lines 218-231 reflects these changes.  The new version of Table S7 shows a range of ER values: the lower limit, which accounts for background I/SVOC concentrations, as well as background CO concentrations, and the high limit, which assumes background I/SVOC concentrations are zero.

**L342: This doesn't make sense. The EF requires background-subtracted mixing ratios (L201). So how then were the I/SVOC EFs calculated? It's not possible to avoid the issue of background then present and discuss EFs anyways.**

As described in detail in an earlier response, to address this point and correct for the influence of I/SVOC background concentrations, we provide a range of EF estimates reflecting low and high limits.  We selected the cartridge collected during the upper 2 transects, which minimally intercepted the plume (average CO mixing ratios were near background levels, ~0.14 ppmv), and we treated this as a background sample. Since there was some slight plume interception during these transects, this sample represents a contaminated background, and thus provides a conservative lower limit of I/SVOC concentrations when subtracted from the cartridge sample collected during the lowest 3 transects. We also present the EF values when no background sample is subtracted (i.e., the background I/SVOC concentrations are assumed to be zero), which represents a high limit to the I/SVOC concentrations.  EF estimates use the background-subtracted average TC mixing ratios (Eq 3) (converted to mass concentrations) across the cartridge sampling time period.  Text at Lines 246-247 reflects these changes specific for EFs.

**L358: This section will need to be reworked. If I/SVOC EFs can't be properly calculated, then TC partitioning based on derived EFs is also affected (Fig. 6, Fig. S8). From Figure 6, I/SVOCs contribute 7.4%, but this isn't known to 0.1%. On L360, the EF is given to 5 sig figs without an error bar. On L364 the 46.2% hinges on the 7.4% for I/SVOCs, which has uncertainty.**

The I/SVOC EFs can be properly calculated and we have revised our EF and ER estimates to reflect a low and high range as described in previous responses.  We present the low/high ranges in Fig 6 (and Fig. S10 (was Fig. S8)) representing the uncertainties in the I/SVOC values.  We have adjusted the percent

contributions to NMOG$_T$ in Fig. 6 (TC partitioning) to reflect the uncertainties in the I/SVOCs (as low/high ranges) and our revised uncertainties for the other NMOGs as described in the next response.

We have corrected the significant figures and added the uncertainty at Line 456-457 (was Line 360).

**L373: Uncertainty needs to be treated holistically in this paper. If some compounds are uncertain to a factor of 2, it's not realistic to present percentages to 0.1%. Each uncertainty needs to be quantified and carried through.**

We reviewed in detail our uncertainty analyses. Our uncertainties were originally presented as the standard deviation of the average EF (or ER) for each plume and for both plumes together. Although this approach has been used before to describe EF uncertainties (e.g. Permar et al., 2021; Liu et al., 2017; Simpson et al., 2011), we decided, for a more holistic approach, to also combine the measurement uncertainties into overall uncertainty estimates given some of the large measurement uncertainties e.g. 50% for PTR calculated compounds, similar to that described in Lindaas et al. (2020). Overall uncertainties in the EFs (and ERs) are now estimated by summing in quadrature the standard error of the average EF (or ER), and the measurement uncertainties. We use the standard error to describe the uncertainty of the average EF (and ER) based on the repeated transect measurements across the SP and NP. The measurement uncertainties are estimated by propagating the measurement uncertainties associated with the ΔX, ΔTC, and ΔCO mixing ratios across all transects for each plume. To be conservative, we also chose not to report compounds that were close to background levels defined as within 1σ of the background variation. These revisions resulted in the removal of 57 compounds from analysis in this manuscript. Table A1 and related figures (Figs 5, 6, 7, 8, 9 and SI Figs. 7, 8, 9, 10, 11, 12, 13 and 14) have been updated. Text has been modified to reflect the reduced number of compounds being reported ie from 250 to 193 compounds. We have also revised our percentages to more appropriately reflect the uncertainties in Fig. 6, as well as our percentages in the pie charts and the text.

A new section, 2.5 Emission uncertainties at Line 257 has now been added with a discussion on uncertainties and a description of how the uncertainties in the EFs and ERs are estimated. The text related specifically to how the uncertainties are estimated is as follows at Line 271,

*'Uncertainties in the EFs and ERs are estimated by summing in quadrature the standard error of the average EF (or ER) and the propagated measurement uncertainties. The standard error is used as description of the uncertainty on the average EF (and ER) characterizing repeated transects across the SP and NP for a total of 20 min of in-plume sampling. The standard error is expected to at least partially capture uncertainties associated with plume aging and vertical plume heterogeneity. As many compounds exhibited significant in-plume enhancements above background levels, uncertainties in the integrated ΔX, ΔCO and ΔTC values were assumed to be dominated by instrumental (measurement) uncertainties (Table S1, S2). Emissions are not reported for compounds where the average mixing ratios were within 1σ of the background average. The low and high I/SVOCs EFs (and ERs) are provided as estimates of their uncertainties (as described in Sect. 2.3). The derivation of AWAS and cartridge EFs (and ERs) may have potential limitations as they rely on a limited number of samples, with the potential of the AWAS discrete samples capturing only part of a plume.'*

**L409: In Fig. S11, I was surprised by some of the differences between the NP (older) and SP (younger). Long-lived acetylene is in the SP top 25 but not the NP. Reactive isoprene is in the NP top 25 but not the SP. The NP isoprene value (0.6 g/kg) is high compared to boreal forest fire literature (0.07-0.15; Andreae, 2019, Akagi et al., 2011), which could be possible furan interference (e.g., Santos et al., 2018, https://doi.org/10.5194/acp-18-12715-2018). Please discuss these issues.**

Subsequent to EF revisions as discussed in previous responses, acetylene is now 19[th] in the SP and 27[th] in the NP. Previously, acetylene was 23[rd] for the SP and 29[th] for the NP. In addition, of the top 25 organics, isoprene is 11[th] in the NP and 25[th] for SP; previously it was 12[th] in the NP and 28[th] in the SP. We would expect that EFs for long-lived species would be in agreement between the NP and SP. The NP-SP EFs for acetylene is just shy of agreeing within their uncertainties (NP=0.20±0.05 g kg$^{-1}$; SP=0.34±0.11 g kg$^{-1}$). We also agree that it is surprising that the EF for reactive isoprene is larger in the NP compared to the SP (factor of ~3) given that the NP appears to be ~30 min older than the SP (Line 183). Although the reasons for these differences are unknown at this time, we speculate that differences in photochemistry (oxidant levels) between the NP and SP preferentially influenced these more reactive species. This is consistent with the observation of higher O$_3$ mixing ratios in the SP (52.4±3.0 ppbv) compared to the NP (44.7±3.6 ppbv).

We included some discussion at Line 611,

*'EFs for the NP and SP generally agreed within their uncertainties with larger differences for some of the more reactive species like isoprene, monoterpenes, and furan. For example, the SP EF for isoprene was a factor of 3.4 lower than that for the NP (0.64±0.3 g kg$^{-1}$) (Fig. S13). Although the reasons for these differences are not yet known, observations of higher O$_3$ in the SP (52.4±3.0 ppbv) compared to the NP (44.7±3.6 ppbv) suggest the influence of higher oxidant chemistry in the SP emissions compared to the NP.'*

We also acknowledge that the NP EF (0.64±0.34 g kg$^{-1}$) for isoprene is quite a bit higher compared to the boreal forest average of (0.074-0.15 g kg$^{-1}$; n=2). We note, however, in the manuscript (Line 662) that these comparisons are limited by few values available. If we expand our comparisons to other reported values, isoprene emissions vary widely, e.g. in Koss et al. (2018) the isoprene EF varies from 0.18 to 0.46 g kg$^{-1}$ depending on the fuel type used in the lab experiments. In Hatch et al. (2017), the isoprene EF for ponderosa pine is 0.404 g kg$^{-1}$ and for black spruce is 0.279 g kg$^{-1}$. Although these values are still lower than our NP EF value, these results taken together highlight that isoprene emissions can vary widely, and even within the same ecosystem (NP vs SP this study). Discussion around why our EF isoprene values differ from that reported in Simpson et al. (2011) (most closely comparable in terms of ecosystem type) is already provided at Line 665.

We added text at Line 667,

*'The difference in EFs for isoprene would be even greater if only the NP EF (0.64±0.34 g kg$^{-1}$) is compared (if it is assumed that isoprene emissions were influenced by photochemical losses in the SP).'*

We include additional discussion in a subsequent response related to differences in EF between our study and other studies.

As for furan, furan was measured in the PTRMS as C$_4$H$_4$OH$^+$ at m/z 69.033 and fully separated from the isoprene C$_5$H$_8$H$^+$ peak at m/z 69.070. Interferences between isoprene and furan are likely only an issue with a quadrupole-PTR. In the Santos et al. paper, a PTRMS was used with a quadrupole detector that was not able to separate furan and isoprene. However, given the possibility of contributions from other unknown isomers to the PTRMS signal in the fresh smoke plumes along Screen 1 (see intercomparison Fig. S3), we now report the EFs for isoprene from both the PTRMS and AWAS (Table A1). These EFs agree within their uncertainties for the SP, NP and average values (the average EF from the PTRMS is 0.42±0.26 g kg$^{-1}$ and from the AWAS is 0.30±0.18 g kg$^{-1}$).

**L430: Akagi et al. (2011) and Andreae (2019) also present PM results from boreal forest wildfires. Please state which specific particle species results are the first here.**

Andreae (2019) shows PM EFs but not for PM chemical species, except BC, and Akagi et al. (2011) only for $PM_{2.5}$. We modified the text at Line 536, to add

*'...chemically-speciated..."*

And at Line 538, added *'...(except BC)'.*

**L434: How fresh were the plumes in Liu et al. (2017)? Could different plume aging between the two studies also play a role?**

The plumes used to derive PM composition EFs in Liu et al (2017) ranged from 20-120 min. Our plume ages of ~42 to 72 min, lie within the range of their measurements suggesting that plume ages are not a dominant factor in PM differences. Their combustion efficiency ranged from 0.877-0.935, while ours was lower at 0.82. With a lower combustion efficiency in our study, it might be expected that the $EF_{PM}$ (or $EF_{OA}$) would be higher compared to Liu et al. (2017) as per the anticorrelation of $EF_{OA}$ with combustion efficiency shown in Fig. 4 of Liu et al (2017). However, this relationship can be impacted by factors like OA loading, gas-particle partitioning related to dilution, and fuel moisture content (May et al. 2014). Although differences in fuel composition may explain some variability in these comparisons, other factors will also influence these comparisons. The Liu et al. (2017) study presented comparisons with prescribed burns which were lower by factors of 2.2 to 8.7 for $EF_{OA}$; OA was $24.3\pm6.1$ g kg$^{-1}$ for western wildfires, and $2.8\pm1.6$ to $11.2\pm2.7$ g kg$^{-1}$, which they speculate may be due to differences in fuel content, and condition, burning condition and fire size and intensity. The results presented here ($EF_{OA}=6.6\pm1.1$ g kg$^{-1}$), are lower than the Liu et al. (2017) western wildfires (factor of 3.7), but are within the range for prescribed burns. This suggests that the wildfire burning conditions in present study may be more similar to that of the prescribed burns in Liu et al. (2017) which are typically more controlled and contained meteorological and fuel moisture conditions.

We have modified the text to incorporate the above discussion at Line 544,

*'The lower OA emissions under smoldering conditions in the current study compared to Liu et al. (2017) with higher combustion efficiencies (0.877 to 0.935) conflicts with some findings showing increased OA emissions with lower fire intensities (Liu et al., 2017, Burling et al., 2011). However, the relationship between $EF_{OA}$ and combustion efficiency can be impacted by multiple factors such as OA loading, gas-particle partitioning related to dilution, and fuel moisture content (May et al., 2014). The $EF_{OA}$ in the current study ($6.6\pm1.1$ g kg$^{-1}$) lies in the range of $EF_{OA}$ reported for prescribed burns across three temperate ecosystems ($2.8\pm1.6$ to $11.2\pm2.7$ g kg$^{-1}$) (May et al., 2014). This may imply that the low intensity, surface, smoldering wildfire conditions in the present study (Sect. 3.1) may be similar to prescribed burn conditions which are typically low intensity fires that are restricted to the forest floor and understory, and conducted under controlled and consistent meteorological and fuel moisture conditions (Yokelson et al., 2013; Carter and Foster, 2004).'*

We have also made modifications to the text at Line 524,

*'The $PM_1$ EF of $6.8\pm1.1$ g kg$^{-1}$ (accounting for estimated mass differences due to particle diameters (Sect. 2.1.2)) falls in the lower end of the large range previously observed for boreal forest wildfires ($18.7\pm15.9$ g kg$^{-1}$; Fig. 9b). The few PM EFs for BFF19 (n=5) over a limited range of MCEs (i.e. 0.89 to 0.93)*

*shows significant variability consistent with previous work (Jolleys et al., 2015; Akagi et al., 2011; Cubison et al., 2011; Hosseini et al., 2013).'*

We also expanded our discussion of OA EF comparisons with Liu et al. (2017) at Lines 541-554,

*'Although differences in fuel type burned between the present study (mature Jack pine, boreal spruce, boreal mixed-wood) and Liu et al. (2017) (mixed conifer, grass, brush and chaparral) may influence the chemical composition of emissions, these large differences suggest the importance of other factors in controlling OA emissions. The lower OA emissions under smoldering conditions in the current study compared to Liu et al. (2017) with higher combustion efficiencies (0.877 to 0.935) conflicts with some findings showing increased OA emissions with lower fire intensities (Liu et al., 2017, Burling et al., 2011). However, the relationship between $EF_{OA}$ and combustion efficiency can be impacted by multiple factors such as OA loading, gas-particle partitioning related to dilution, and fuel moisture content (May et al., 2014). The $EF_{OA}$ in the current study (6.6±1.1 g kg$^{-1}$) does lie in the range of $EF_{OA}$ reported for prescribed burns across three temperate ecosystems (2.8±1.6 to 11.2±2.7 g kg$^{-1}$) (May et al., 2014). This may imply that the low intensity, smoldering wildfire conditions in the present study (Sect. 3.1) may be similar to prescribed burn conditions which are typically low intensity fires that are restricted to the forest floor and understory, and conducted under controlled and consistent meteorological and fuel moisture conditions (Yokelson et al., 2013; Carter and Foster, 2004). Inorganic PM emissions, however, are likely more dependent on fuel elemental composition than combustion efficiency (Liu et al., 2017).'*

Carter, M.C., and Foster, C.D., Prescribed burning and productivity in southern pine forests: a review: Forest Ecol. Mgmt., 191, 93-109, 2004.

**L434, 439: Different boreal versus temperate forest fuels are used to help explain EF differences, but their vegetation types can also be similar (L421). Please state the major vegetation species for Permar et al. (2021) and Liu et al. (2017) so their fuels can be compared with this study.**

We have now included the major vegetation species for Liu et al. (2017) at Line 542 and for Permar et al (2021) at Line 518, as well as a reminder of the present study vegetation at Line 541.

**L444: The paper still needs to present the sample duration used for EF calculations. From Fig. 3 (Screen 1), it looks like up to 20 minutes of in-plume sampling (10 mins each for NP and SP) over 1 hour of flight time. Please discuss sample size/duration as a potential limitation or source of uncertainty.**

There was approximately 20 min of in-plume sampling for the real-time measurements. For the SP, transects 1 to 4 ranged from 2:24 min to 3:46 min in-plume for a total of 11:31 min in-plume. For the NP, transects 1 to 3 ranged from 2:26 min to 3:23 min in-plume for a total of 8:24 min. Both in-plume durations together totalled 19:55 min on Screen 1. This is now specified at Line 274, and the number of AWAS samples is indicated at Lines 216. We had acknowledged in the Summary and Implications section at Line 852 that these results represent only one smoldering fire.

In addition, in a new Section, 2.5 Estimates of Emissions Uncertainties, text has been added to acknowledge the limited sample number of AWAS and cartridge samples at Line 280,

*'The derivation of AWAS and cartridge EFs (and ERs) may have potential limitations as they rely on a limited number of samples, with the potential of the AWAS discrete samples capturing only part of a plume.'*

**L451: Here and elsewhere, ensure that the average and its uncertainty match after the decimal place. So 0.31 ± 0.028 should be 0.31 ± 0.03. In Table A1 the BC error isn't right (0.11 ± 0.0098). And so on.**

Table A1 and associated text now appropriately reflects uncertainties.

**L458: What would explain HONO differing by a factor of 41 (!) from Andreae (2019)? Could it be a measurement issue? HONO EFs across all vegetation types range from about 0.2-1.2 (Akagi et al., 2011; Andreae, 2019), so the EF reported here (0.01, Table A1) is far off and unlikely to be explained by limited studies (L459).**

We have revisited the CIMS HONO measurements. We have re-checked the mass spectra and fits and re-ran the calculations by hand and obtained the same results. There was nothing obviously diagnostically wrong with the CIMS instrument. If there was an artifact present, we would have expected it to be $NO_2$ reacting on the inlet (to form HONO) to bias the numbers high, however, this is clearly not the case. The signal is low at the (I)HONO peak. Having ruled out measurement/instrument issues, we report the HONO EF of 0.02±0.012 g kg$^{-1}$. Note that as discussed in an earlier response, we removed the HONO EF for the NP as the in-plume mixing ratios were within 1σ of the background values.

There, in fact, does not appear to be any reported HONO EFs for the boreal forest ecosystem in Andreae (2019) (the previous value of 0.41 g kg$^{-1}$ indicated doesn't appear to be related to any publication in the supplemental table), thus, we compare only with the Koss et al. (2018) value of 0.60±0.20 g kg$^{-1}$ (Table A1 updated to reflect this). Although the comparison differs by a factor of 30, and it is a useful exercise to compare them, we consider this comparison to be limited by having only a single, relevant reported HONO EF in the literature.

Most importantly, with respect to HONO, or any other reported species, we contend that comparisons to literature values for EFs/ERs must be done with caution. Wildfires are complex and highly variable owing to a number of both fuel and environmental related factors. There is no reason to expect that EFs should agree well between fires in different locations (or even the same location) given all of the variables. Certainly, differing EFs compared to literature across studies is not a reason to negate one EF over any other in the literature. We do not consider our HONO EF to be so different than that of the Koss et al. (2018) value (or any other reported value) to then negate it, particularly with limited statistics, as well as differences in deriving EFs between lab and field measurements as discussed in Hodshire et al. (2019). As such, we believe both estimates are equally valid.

We modified the text at Line 581 to read,

*'However, EFs for most other gaseous inorganic species were lower than the BFF19 EF average including $NH_3$, HONO, and $NO_x$ by factors of 4.0, 20, and 7.1, respectively(Fig. 9a). There are only a limited number of studies reporting EFs for these compounds in the BFF19 category. For example, there are only 4 previously reported BFF19 EFs for $NH_3$ (2.5±1.8 g kg$^{-1}$) showing a large range of values. Although these comparisons are limited by the few reported values in the literature, the differences indicate a strong sensitivity towards factors like fire intensity, chemical reactivity, fuel type and moisture, and meteorology.'*

**L458: What would explain $NO_x$ being lower by a factor of 15? Also check if 15x is an error (Table A1): 1.2 (Andreae) divided by 0.12 (this study) = 10x not 15x? The $SO_2$ EF is 0.22 (Andreae) and 0.17 (this study) so better agreement than 4.7x? Please check all calculations.**

Thank you, the $NO_x$ EF is actually 7x lower, and $SO_2$ EF is not lower as we are now reporting just the SP EF (as discussed in a response on uncertainties). This has been corrected in the text at Line 582. There is a large range of $NO_x$ EF reported for the boreal forest ecosystem ie $1.18\pm0.86$ g kg$^{-1}$ (Andreae 2019), and our $NO_x$ EF of $0.17\pm0.04$ g kg$^{-1}$, indicates additional variability, and isn't necessarily expected to agree with previous reports given the complexities identified in the previous response. Changes were already made at Line 581 in the previous response to also address $NO_x$.

**L469: Something is wrong with the butanes in the top 25. The EF of n-butane should be 2-3 higher than isobutane (Akagi et al., 2011; Andreae, 2019), yet only isobutane is in the top 25 (Fig. 8). The isobutane EF (0.47) is about 10x higher than the boreal literature (0.042-0.052), also suggesting a measurement issue.**

We have spent considerable effort revisiting the AWAS data. The AWAS data have gone through a rigorous QA/QC process. Gas reference standards were obtained from the National Physical Laboratory UK ($C_2$ to $C_9$) and Apel-Reimer Environmental ($C_7$ to $C_{10}$) containing all compounds reported with stated uncertainties of 2% to 5% and were used without dilution. Mixing ratios in the standards were between 0.2 and 20 ppb(v). Calibration curves were generated by varying the volume of standard gas injected and converting this to equivalent mixing ratios on an Area Count*pptv$^{-1}$*mL$^{-1}$ basis for the sample injections. Full calibrations with all standards were carried out at the start of the study and whenever the instrument had been adjusted, cleaned, or repaired. A secondary reference gas (high pressure cylinder filled with wintertime dry air from the Canadian Arctic) was analyzed daily to monitor any peak response changes (drift in the MSD), chromatographic performance, and overall instrument function. Gas flows were determined from measurements with MesaLabs Dry-Cal flow meters of various measurement ranges. Instrument gas pressures were checked and recorded daily. We have also re-evaluated the data by means of re-calculating the mixing ratios using a different method (in this case, entirely by hand) for selected data from the instrument raw sample and calibration signals (area counts). In addition, the identification of these peaks was visually re-verified despite this being done during the initial analysis of the data. No errors were found.

Our n-butane EF of $0.15\pm0.04$ g kg$^{-1}$ agrees within the standard deviation of the boreal forest average of $0.11\pm0.06$ g kg$^{-1}$. However, we agree that the EF for i-butane of $0.47\pm0.16$ g kg$^{-1}$ is unusually high compared to previous literature (boreal average $0.052\pm0.052$ g kg$^{-1}$) and also higher than n-butane, which has not been previously reported for any ecosystem. We scoured the literature for possible explanations for the elevated i-butane, but could not find a reasonable explanation. Given our reduced confidence in the i-butane data with these inconsistent results, we have chosen, conservatively, to remove i-butane from our analyses. We have adjusted the relevant figures to reflect this ie. Figs. 5, 6, 7, 8, 9 and Table A1, Table S4, Figs. S8, S9, S10, S11, S14.

**L472: In Table A1, why are temperate forest EFs (Permar et al.) being used for the butanes, rather than boreal forest EFs (Akagi et al., Andreae)? (0.042-0.052 for isobutane, 0.11-0.12 for n-butane). Is the same value for both (0.12) in Table A1 a typo?**

The table now includes both boreal forest and temperate forest values for the butanes with correction made (typo).

**L516: What might explain the much higher EF for pyruvic acid compared to the literature (37x)? How were measurement issues ruled out? When there are such huge discrepancies, it doesn't make sense to then cite acids as 10.3% of NMOGs (since the errors aren't as small as 0.1%).**

Pyruvic acid was measured by the CIMS as its cluster with Iodide ($I(C_3H_4O_3^-)$). The signal was calibrated using an 8 component certified standard solution provided by Caledon Labs (caledonlabs.com). The 8 components (each 10 mM) were Butyric Acid, Hexanoic Acid, Pyruvic Acid, Methacrylic Acid, Levulinic Acid, 3-Hydroxybutyric Acid and Oxalic acid. A gas phase standard was generated from this solution by diluting the stock solution and nebulizing it using a Liquid Calibration Unit (Ionicon) which had been modified to allow for the addition of extra dry dilution gas. The resulting calibration factor at 25 degrees C and 30 % RH agree very well with the values obtained by Lee et al (2014, Table S1) using the same ionization scheme. In their study they determined a value of 0.35 ncps/pptv whereas ours was 354.2 ncps/ppbv. In addition, there are no other possible isomeric species with any appreciable sensitivity at the same exact m/z of the pyruvic acid-iodide cluster. While we agree the pyruvic acid EF values are high relative to some literature, there is nothing to suggest an instrumental cause or calibration error as reason for the difference. Regardless, as noted previously, differences between the sparse literature data that exists and our data is not a reason to assume that it is a discrepancy given the complexity of factors that control wildfire emissions. As such, we have retained the pyruvic acid data and associated emission factors/ratios. Note that the pyruvic acid EF, 0.72±0.71 g kg$^{-1}$ has a high uncertainty, largely due to the variability from transect to transect; this may reflect effects of chemistry.

Veres et al. (2010) suggest that the lignin content of the fuel may be responsible for the release of organic acids in biomass burning and thus, the fuel content may be an important factor in assessing such emissions. We also expect the differences in EFs to be related to other factors such as type of fire, water content, fire age, fuel type, and environmental conditions. We provide such an explanation at Line 651,

*'Differences in fuel type may be an important factor in the variability of these comparisons. Based on laboratory experiments, Veres et al. (2010) found a large range (factor of 5 to 13) of organic acid emissions with different fuel types suggesting that the lignin content of the fuel could be a source of biomass burning organic acid emissions.'*

Lee, B. H., Lopez-Hilfiker, F. D., Mohr, C., Kurtén, T., Worsnop, D. R., & Thornton, J. A. (2014). An Iodide-Adduct High-Resolution Time-of-Flight Chemical-Ionization Mass Spectrometer: Application to Atmospheric Inorganic and Organic Compounds. Environmental Science & Technology, 48(11), 6309-6317. http://dx.doi.org/10.1021/es500362a

We also compared pyruvic acid between the PTRMS and CIMS, however, while other acids are tightly correlated (Fig. S3), pyruvic acid is not. This is likely due to the PTRMS measuring less volatile compounds and inlet line losses. In addition, the PTRMS sensitivity is calculated rather than directly calibrated, and was not corrected for inlet losses. As a result, we haven chosen to retain the CIMS pyruvic acid only. We removed pyruvic acid from Table S5 overlapping compounds.

We have modified the percent contribution of acids to the ΣNMOGs which now more appropriately reflects the uncertainty of this estimate. We have also modified the percent contribution of categorized species in Figs. 4, 5, and in the SI Figs. 7, 8 and 9.

**L518: As well as HONO, isocyanic acid is also much lower than the literature. There is also a factor of 4 difference between the two isocyanic acid instruments in this study (Table A1). Please present and discuss instrument intercomparisons.**

We have decided not to report the PTRMS HNCO, as the PTRMS HNCO measurements may have potential interferences (e.g. cyanic acid, fulminic acid). We are retaining the CIMS measurements as they are selective towards HNCO only, as demonstrated in the literature. The sensitivity (ie: calibration factor)

for the CIMS HNCO data is within the range obtained for the same ionization scheme in other studies (Lee et al., 2014). Calibration factors for HONO are also well within the range reported for this type of CIMS. As with pyruvic acid (and all of our measurements), we have gone over our data in great detail and found nothing to suggest an instrumental cause or calibration error as reason for the differences. We acknowledge that the CIMS HNCO EF is lower than previous reported EFs with the largest difference with the laboratory study (factor of 6.9 with Koss et al., 2018) and a smaller difference with the field study (factor of 1.9 with Permar et al., 2021). Our HONO EFs are also lower by a factor of 30 compared to Koss et al. (2018). We note that there are still very few measurements of HNCO and HONO in emissions from wildfires or laboratory studies, and none available for the boreal forest ecosystem. Again, note that the previous value of 0.41 g kg$^{-1}$ indicated doesn't appear to be related to any publication in the supplemental table. Consequently, there is no reason to expect agreement between the few studies that exist for these compounds. While there is value is conducting these comparisons as they provide some context and added statistical information, especially where there are still few measurements, drawing any conclusions from these comparisons is not possible. The behaviour of any fire (and the various conditions/parameters associated with a fire) is likely to mean that a direct comparison between EFs here and in the literature is complex, as we have stated on several occasions above. We contend that such comparisons, while important, do not negate the measurements of EFs from either study(s) in the absence of any instrumental/calibration error. No changes in the manuscript have been made in this case.

**L579: Again, what would explain lower EFs in this study for so many compounds, including lower by up to 49-57x? These are huge differences. Please discuss, and also check your calculations.**

As discussed in detail in earlier responses, we have spend considerable effort going through all of our measurements and calculations including assessing mass spectra, chromatograms, calibrations and emissions calculations. We have excluded measurements where they were close to background levels. We also re-assessed and revised our overall uncertainties to better reflect measurement uncertainties. We consider our reported data and uncertainties to be robust. There is value is conducting comparisons with previous reported literature, as they provide some context, and increase the statistics to help constrain emissions. Placing our measurements in the context of previous measurements highlights where emissions may be considerably variable (e.g. PM, NH$_3$, pyruvic acid, isoprene) or where there may be better agreement (e.g. HCN). We do not assume that newly derived EFs from this study should necessarily agree or disagree with the literature, particularly for compounds with limited reported EFs. The behaviour of any wildfire is likely to mean that a direct comparison between EFs here and in the literature is complex. We contend that such comparisons, while important, do not negate the measurements of EFs from either study(s) in the absence of any instrumental/calibration error. We have improved text at various points to discuss differences as per the reviewer's comments. This includes PM at Line 526, OA at Line 541, NO$_x$ and HONO at Line 586, and pyruvic acid (organic acids) at Line 651. Note that we have also discussed similarities such as HCN at Line 588, furan and furfural at Line 691, CO$_2$ and CO at Line 579.

We also provide additional discussion related to differences in EFs between our study and others at Line 696,

*'These comparisons provide context for the emissions reported in the present study and moves towards improved statistics to better constrain wildfire emissions. Additional factors are considered to explain variability in emissions between this study and other reported values, as well as within this study (NP vs SP). Differences and variability in burn conditions (e.g. fire intensity, winds, fuel density, flame dynamics, fuel moisture) likely influence these comparisons; the Screen 1 measurements in the present study were taken from 9-10 am LT when the fire was in a low intensity, smoldering state, while those in Permar et al. (2021) and Simpson et al. (2011) took place during mid-day under active wildfire*

*conditions. Aircraft measurements in general have a higher probability of sampling variable burn conditions compared to laboratory studies (Hodshire et al., 2019), and as such, aircraft-derived EFs are likely to reflect variability for reactive species as speculated earlier with isoprene. Particularly for reactive species that can exhibit complex variation across plumes, EFs (and ERs) derived by integrating across plumes can be biased low, (Sect. 2.5; Peng et al., 2021; Decker et al., 2021). Also, EFs derived using TC in this study may result in lower, albeit small, EFs compared to reported values that do not account for all the carbon (estimated to be 1-2 % (Akagi et al., 2011)).*

**L587: Throughout avoid qualitative wording ('much lower', 'slightly higher', 'slightly different'). Use percent differences. ALK5 is so very far off in Fig. S12 (70x?) - what caused this?**

The measurement derived emission SAPRC-11 model speciation profile shown in Table S9 and the associated Fig. S14 (previously Fig. S12) were updated to reflect the changes in EF in this revised manuscript. We have revised the discussions with following at Line 739:

*'For a research version of the FireWork system, the component speciation is mapped to the SAPRC-11 chemical mechanism species (Carter and Heo, 2013) with detailed oxygenated compounds and aromatic species, largely to better represent SOA formation processes. For comparison with the measurement derived speciation profile in this study, EFs were first mapped to SAPRC-11 species and normalized by the total identified mass species fraction without unknowns to obtain mass fractions of relevant model mechanism species (Table S9). Comparing the normalized mass fractions for similar mechanism species (Fig. S14) showed a substantially lower fraction of reactive alkanes (ALK5) with an estimated 5 % in this study compared to 28 % in the SPECIATEv4.5 wildfire smoldering profile. Mass fractions in this study are notably higher for the ACYL, ETHE, and ISOP lumped model species by factors of 13, 7 and 51. The mass fraction of CH$_4$ is also different with 24 % of TOG in this study compared to 4 % from the SPECIATE4.5 profile. The measurement derived chemical speciation profile is expected to be different from the average speciation profile from EPA's SPECIATEv4.5 due to differences in chemical species identification, fuel type, fire and measurement conditions, and uncertainties on how measured compounds are mapped to lumped mechanism species. The emissions profile developed in the present study can be used to improve predictions of wildfire smoldering emissions specific to the Canadian boreal forest.'*

**L593: 'more representative' – this study has large differences from the literature, and a number of results point to possible measurement issues. These need to be resolved before drawing this conclusion.**

We have addressed these questions posed by the reviewer in previous responses. We have modified the text at Line 753,

*'The emissions profile developed in the present study can be used to improve predictions of wildfire smoldering emissions specific to the Canadian boreal forest.'*

**L617: Please provide more detail for the mass balance calculation. What parameters were used and how were the error bars were derived?**

More detail has been provided. Section 1.2 Mass balance method for estimating aircraft-derived emission rates has been added to the SI. Main manuscript text modified at Line 782 to,

*'Emission rates in metric tonnes per hour (t h$^{-1}$) were derived from selected aircraft measurements using a mass balance method that was designed to estimate pollutant transfer rates through virtual screens using aircraft flight data (Gordon et al., 2015) (see SI Methods).'*

**L631: The UTC time isn't intuitive in Fig. 10. Use local time instead (or as well) to clearly indicate when local morning/evening is. Include some local time conversions in the text.**

We have included a conversion to local time in the figure caption. *Local time = UTC – 6 hrs.*

**L647: Please avoid 'very similar' and 'well within the uncertainties.' Provide error bars for both emission rates. Is there a typo? The two values are 10x different.**

Word 'very similar' and 'well within uncertainties' have been removed. In this Section, we made some modifications to the text to improve clarification of ratios (now specified as $R_{species/FRP}$) and to differentiate from emission ratios and emission rates. Error bars provided for the $R_{CO/FRP}$ values and the first value has been corrected (typo). Text starting at Line 817 is modified as,

*'For the two satellite overpasses during the flaming phase of the fire, the $R_{CO/FRP}$ values were within the uncertainties (19:06 UTC $R_{CO/FRP}$ =0.47±0.25 t h$^{-1}$ MW$^{-1}$; 20:48 UTC $R_{CO/FRP}$ = 0.43±0.23 t h$^{-1}$ MW$^{-1}$).'*

**L648: Avoid using 'ER' (which is emission ratio, L405) for 'emission rate.' On L648 and L649, do not use 'ratio' if you mean emission rate. On L651, add error bars and do not cite to 5 sig figs.**

In this Section, we made some modifications to the text to improve clarity of ratios (now specified as $R_{species/FRP}$) and to differentiate between emission ratios and emission rates. We also added error bars to the reported total emissions and reduced significant figures at Line 820 and 826.

**Figure 1 doesn't indicate scale or orientation. Please include axis labels, a distance scale, and North direction. Is the 'large arrow' missing? Please label the five screens, especially since Screen 5 isn't shown in Fig. 2.**

Fig. 1 has been updated to include a distance scale, axis labels, a north direction arrow, a red arrow for plume direction, and labels for all five screens.

**SI L92: A measurement uncertainty of 40% for AWAS grab samples (Table S1) is incredibly high for this type of measurement. What's contributing to this? If it's this high, how does it impact the EFs based on AWAS? (which so far don't have error bars in Fig. 8 and Table A1)**

A revised uncertainly value of 25 % has been assessed for compounds analyzed by flame ionization detection. These include all $C_2$ to $C_5$ compounds and $C_6$ alkanes. This value is slightly above the value (20 %) suggested for alkenes (the most unstable light hydrocarbons) in the data quality objectives of the WMO 1$^{st}$ GAW-VOC Intercomparison study for canister and in-situ measurements (Rappengluck *et al.*, 2006). This elevated uncertainty estimate for alkanes, alkenes, and alkynes measured by GC-FID is based on analytical issues caused by incomplete water vapour management in the sample gas stream. The majority of water in the sample was intentionally switched onto a column known to be relatively insensitive to moisture perturbations (RTX-QS used for $C_2$ and $C_3$ species) by means of a column switching valve downstream of a pre-column (SPB-1) which separated these compounds, water, and $CO_2$ from higher molecular weight species. The higher molecular weight compounds were switched to a second column (HP-AL/S) connected to a second FID. However, sample water vapour was found post-analysis to cause retention time shifts in the pre-column itself, and some peak broadening effects. It was possible to manually adjust the integration method to properly identify the peaks with drifting retention times, however, this caused additional area count uncertainty. Therefore a value of 25 % was considered appropriate for the two FID's data under these operating conditions.

A higher uncertainty estimate was applied to the compounds analyzed by MSD compared to those from the two FIDs. These constituted the $C_6$ alkenes, and the $C_7$ to $C_{10}$ species. There were also retention time shifts for these compounds due to incomplete water vapour management in the sample air. While it was possible to correctly identify the compounds with the retention time drift experienced, this had the potential to cause less accurate quantitation of the compounds. In each set of 12 samples (each AWAS module), one sample canister was analyzed in duplicate. The replicate results indicated that 40 % is a reasonable uncertainty level to assign given that the most variable compounds approached this level of imprecision.

We added text to the SI at Line 138 to address this,

*'Toluene, benzene and xylenes were not quantified because the method and columns used were not optimized for these compounds. Analytical issues due to incomplete water vapour management in the sample gas stream resulted in retention time shifts and some peak broadening effects resulting in elevated uncertainties. Uncertainties are estimated at ±25 % for $C_2$ to $C_5$ and $C_6$ alkanes detected by FID, and ±40 % for C6 alkenes, and the $C_7$ to $C_{10}$ species detected by MS.'*

We have now included overall propagated uncertainties on the reported EFs (and ERs) in Table A1 and in Fig. 8 for the AWAS values which is the combined standard error and measurement uncertainties, as described in more detail in earlier responses.

Rappengluck, B., Apel, E., Bauerfeind, M., Bottenheim, J.W., Brickell, P., Cavolka, P., Cech, J., Gatti, L., Hakola, H., Honzak, J., Junek, R., Martin, D., Noone, C., Plass-Dulmer, C., Travers, D., and D. Wang. 2006. The first VOC intercomparison exercise within the Global Atmosphere Watch (GAW). Atm. Env. 40: 7508-7527, doi:10.1016/j.atmosenv.2006.07.016

**SI L375-377 (Table S7): It doesn't make sense to talk about EFs in a table that presents ERs. Do you actually mean ERs here? Using concentrated plume values to 'approximate' an ER (or EF) isn't OK. Without a background subtraction the ER can't be presented as an ER.**

We have removed reference to EFs in this table that presents ERs. In our revisions, we selected the cartridge collected during the first screen across the highest 2 transects to use as a background sample (see previous responses). We revised Table S7 to reflect this. The new version of Table S7 shows a range of ER values: the lower limit, which accounts for background I/SVOC concentrations, as well as background CO concentrations, and the upper limit, which assumes background I/SVOC concentrations are zero (our original estimates). The figure caption has been modified accordingly.

**ADDITIONAL COMMENTS. Lots of details to address. Please proof the paper carefully.**

**L20, L80: Because boreal forest fires have been studied, change 'a lack of' to 'limited'**

Changed at Line 20 and 88.

**L54: Wording is very similar to L18.**

This is intentional, no change.

**L58, L61, L597, L608, L611: Please provide references.**

References provided (web site links) for first two instances at Line 61 and Line 66.
Third instance added Chen et al. (2019) at Line 759.
Fourth instance – Added a reference at Line 772.
Fifth instance – this smoldering vs flaming distinction is for this fire specifically (as shown in Fig. 10)

**L69: Should 'and ratios' be deleted here?**

Deleted at Line 74.

**L94-95: Please use different wording in the abstract, introduction and conclusions.**

We modified text in the introduction and Summary and Implications to reduce repetition in the abstract, introduction and Summary and Implications. The following text was revised in the Introduction at Line 55,

*'The severity and frequency of wildfires is expected to increase in response to climate change (Bush and Lemmen, 2019; Seidl et al., 2017; Whitman et al., 2019) with evidence to suggest that such impacts are expected to be most pronounced in the boreal biome (Seidl et al., 2017; Whitman et al., 2019).'*

And at Line 91,

*'In the summer 2018, a research aircraft was deployed to measure emissions and subsequent transformation processes from a boreal forest wildfire in western Canada (Fig. 1; Fig. S1). In this paper, measurements of a comprehensive suite of gas- and particle-phase compounds are used to provide a detailed characterization of smoldering wildfire emissions. The highly speciated non-methane organic gas (NMOG) measurements are described by broad chemical classes and across a range of volatilities extending from VOCs to SVOCs. The wide range of measured NMOGs, along with concurrent total NMOG carbon ($NMOG_T$) measurements, provides a unique opportunity to reconcile the total carbon budget. Emission factors are derived for 193 compounds which represents the most extensive chemical speciation of wildfire emissions to date, almost tripling the number of reported values for the boreal forest ecosystem in the Andreae (2019) compilation paper. Emission estimates are also combined with those from satellite observations to evaluate modelled diurnal variability. The purpose of this work is to provide relevant emissions information for boreal forest wildfires to ultimately contribute towards improved emissions quantification and chemical speciation representations in air quality models.'*

The text was also modified in the Summary and Implications, at Line 836,

*'Consistent with previous results, highly…'*

And at Line 844,

*'builds on previous work (e.g. Simpson et al., 2011; Andreae 2019) and…'*

**L104: Is 'intervals' the correct word here? Do you mean time resolution?**

Yes, we mean time resolution. Intervals changed to 'time resolution'.

**L104: Typo: change to 'measurement methods.'**

Typo corrected at Line 119.

**L130 and SI L92 give AWAS fill times of 20-30 sec, but SI L96 states approximately 15 sec (30 sec max). Which is it?**

Fill times were 20-30 sec.  SI corrected at Line 111.

**L132 says that AWAS measured <C10 hydrocarbons, but SI L229 says <C9. Which is it? (from Table A1 it seems to be ≤C10)**

The AWAS measured ≤C10 hydrocarbons.  Added this to Table S1.  Text in manuscript at Line 149 and SI at Line 260 fixed.

**L169: Typo: 'depositional'**

Changed sentence at Line 261 to,

*'Pollutants released by wildfires can be influenced by chemical and physical changes that may take place between the time of emission and the time of measurement…'*

**L172, L194: What's the basis for these statements? Is it just based on absence of upwind cities and industries?**

The text at Line 189 has been modified for clarity,

*'There are no significant anthropogenic sources like upwind urban or industrial areas impacting the Screen 1 measurements.'*

**L203: TC was already defined on L115.**

At Line 241, changed to TC.

**L239: Quantify 'low $NO_x$ levels.' Fig. S2 shows up to 5 ppbv in the plumes, which is much higher than background.**

We removed the portion of the sentence referring to low $NO_x$ levels at the Lapina et al. (2008) reference; Line 307.

**L258: If OA is 276 µg/m3, please change the scale in Fig. 3 so the data don't go off-scale.**

The scale on Fig. 3 has been changed to maximize at 280 ug m$^{-3}$

**L263: How do you get 85%? The reduced compounds in Fig. 4 sum to 79.4% (oxidized are 20.6%).**

Thank you. This was an error and should be 79.4 % which we are revising to 79 % at Line 335.

**L281: Hydrocarbons (defined on L279) sum to 27.2 + 19.3 + 3.1 = 49.6% (Fig. 5). How do you get 52.8%?**

After making our revisions, the sum is 19% (alkanes) + 29% (alkenes) + 3% (alkynes) + 2% (portion of aromatics that are CxHy) = 53 % (as per Fig. S8). Text as Line 356 modified as,

*'Hydrocarbons (i.e. $C_xH_y$, including some aromatics) were responsible for just over half of the ΣNMOG (53 %) (Fig. S8), with 29 % identified as alkenes such as ethene, propadiene, and propene, 19 % alkanes, predominantly ethane, and 3 % alkynes, almost entirely acetylene.'*

**L282: Please reference Fig. 5 here. The text goes back and forth between Fig. S6 and Fig. 5.**

Fig. 5 is already referred to in the first sentence of this section, at Line 353.

**L313: The text here is qualitative. Please include concentrations and error bars.**

The text in this paragraph is describing emissions/emission factors related to furfural and phenol emissions. We have now revised the text at Line 389 as,

*'In this study, furfural was the most abundant oxygenated aromatic compound and a factor of 5 times higher than that of phenol. Although Koss et al. (2018) found that phenol and furfural emissions were similar for most fuels tested in the laboratory, furfural emissions derived from multiple wildfires sampled in Permar et al. (2021) were similar to those in the present study, and a factor of 1.6 higher for phenol.'*

**L326: Table S7 lists C11 CH compounds with an ER of 0. Is this a typo?**

This is correct. Some species in the CH/CHO/CHS distributions did show ERs of zero, when their concentrations were below instrument detection limits or when there was signal observed in the blank sample for the same compound (in these cases, the signal of the blank was subtracted out). Sheu et al. (2018) discusses LOD calculations for these methods, and LODs are typically less than 1 ppt.

**L333: The CH compounds peak at C18-C23 in Table S7, not C20-C25.**

Thank you. We changed the text at Line 415 to $C_{20}$-$C_{23}$.

**L342: Here and on L1410 change 'Table A1' to 'Table 1.' On L1412 delete 'Table S7'**

Kept Table A1 at Line 428 as this is correct. Deleted the word 'Table S7' (Line 1611).

**L364: Here and throughout, define all error bars and how they were derived.**

We added an explanation of how the uncertainty was derived for the ΣNMOG at Line 458,

*'The ΣNMOG uncertainties were estimated by summing in quadrature the individual compound EF uncertainties for the SP and NP separately (Fig. S11), with these uncertainties subsequently summed in quadrature to derive the average ΣNMOG uncertainty (Fig. 6).'*

We reviewed in detail our uncertainty analyses and provide details on revisions in a previous response. Also, a new section, 2.5 Estimates of emission uncertainties at Line 257 has now been added with a description of how the uncertainties in the EFs and ERs are estimated. Uncertainties are provided throughout.

**Grammar: change 'SVOCs to VOCs categories' to 'SVOC to VOC categories'**

Changed at Line 492.

**L405: Define EFs and ERs earlier, not here.**

At Line 195, added (ER) to define emission ratios. At Line 234, emission factors are already defined. At Line 504, removed the definition for EFs and ERs.

**L415: L152 states that Jack pine, not lodgepole pine, is a dominant fuel in this study.**

Changed text to: '...*a similar fuel type..*' at Line 515.

**L429: Quantify 'generally similar' (within what %?)**

At Line 534, changed text to,

*'Although the magnitude of EFs between the SP and NP are within their derived uncertainties, the ERs showed differences by up to 70 % for $NH_4$ (Fig. S10) suggesting some differences in photochemistry between the two plumes.'*

**L442: Too many sig figs**

Reduced the sig figures to be more appropriate at Line 524

**L456: Quantify 'very close.' In Table A1, the $CO_2$ EF has too many sig figs.**

Changed text to *'...comparable within uncertainties..'* at Line 580.

Reduced $CO_2$ EF significant figures in Table A1.

**L458: The text uses HONO but Table A1 uses $HNO_2$. Use one notation for easier cross-referencing.**

Changed Table A1 to HONO. Changed Table S2 to HONO notation.

**L458: Add HONO to Table S1 (to show its measurement uncertainty).**

Uncertainties and detection limits have been added to Table S2 for all the CIMS compounds which also includes HONO. HONO has a measurement uncertainty of 30% and a detection limit of 12 pptv.

**L464: Quantify 'fairly well', 'do not vary widely'. On L550 quantify 'fairly good'**

At Line 588, changed text to

*'In contrast, EFs for HCN derived in the current study ($0.31\pm0.03$ g $kg^{-1}$) lie within the range of BFF19, LAB18 and TFF21 values ($0.28\pm0.06$ to $0.53\pm0.30$ g $kg^{-1}$), (Figs 9a, b, c, respectively)...'*

At Line 691, changed text to,

*'Agreement within uncertainties was found with BFF19 for furfural, and furan (Fig 9a).'*

**L481: The value of 57% may change depending on what you find for isobutane.**

We have removed isobutane from our analysis, as well as 57 compounds that were within $1\sigma$ of the background average as discussed earlier. The original value of 57 % was incorrect as it included $CH_4$ in the denominator and should be have been 75 %. Our revised data, specifically for Fig. 8, indicates that that this percentage is now 81 %. The text in the manuscript has been updated at Line 608,

*'In the present study, the top 24 NMOG compounds accounted for 81 % of the $\Sigma$NMOG by total molecular mass with lower emissions from the remaining measured compounds.*

**L482: Typo: 'lower lower'**

Fixed at Line 608.

**L531, 543: Please discuss how possible interference from furan was checked or ruled out.**

Furan was measured in the PTRMS as $C_4H_4OH^+$ at m/z 69.033 and fully separated from the isoprene $C_5H_8H^+$ peak at m/z 69.070. Interferences between isoprene and furan are likely only an issue with a quadrupole-PTR. Any interferences with the isoprene signal would need to be from another compound. Other compounds found in literature that are known to interfere with the isoprene signal include fragments from cylcoalkanes, however cycloalkanes are not expected downwind of the forest fire plume, and there was no evidence of cycloalkanes from other PTR traces in the fire plume. Fragmentation of cycloalkanes did interfere with the isoprene signal in our measurements downwind of the oil sands facilities. 2-methyl-3-buten-2-ol (MBO) produces a fragment at m/z 69.070 that is not separated in the PTRMS and can interfere with the isoprene measurement. We do not have additional measurements during our project to confirm the impact of MBO on the isoprene signal. In Permar et al. (2021), they measured MBO by TOGA and determined that the levels of MBO in the smoke were likely too low to account for the higher than expected isoprene. Permar et al. (2021) reported that isoprene measurements during their study from the PTRMS were approximately 2x higher than the TOGA and AWAS isoprene while sampling smoke. They were unable to identify the source of the additional fragments in the smoke. No changes made in the manuscript related to the question of furan interferences in the isoprene signal.

**L565: It looks like you could add 'NMOG' to 'CO and CH$_4$' since it's also comparable in Fig. 9d.**

Thank you, yes, we added NMOG at Line 720.

**L601: Typo: change 'CFFEPs' to 'CFFEPS'**

Corrected at Line 762.

**L614: 'corresponds with' doesn't make sense … how can 0.29 MW/ha correspond with >0.4 MW/ha?**

Changed text at Line 778 to clarify as

*'This energy density threshold for smoldering <0.29 MW ha$^{-1}$ found in this study is in agreement with O'Brien et al. (2015) who found flaming combustion at >0.4 MW ha$^{-1}$ for lower intensity flaming fires and smoldering combustion at lower energy densities.'*

**L642: The text states 23:00 UTC but Fig. 10 indicates 21:00 UTC (L1402). Is 23:00 correct? If so why aren't the data from 21:00-23:00 shown in Fig. 10?**

The fire continues until June 25, 23:00 UTC as FRP can be observed until this time. Figure 10 stops at 21:00 UTC as rain started, and the discrepancy between the model emissions and FRP increased as a result of rain that is not considered in the model bottom-up emission estimates. Text has been included at Line 767 as an explanation,

*'After 21:00 UTC, the discrepancy between the CFFEPS-predicted emissions and FRP increased as a result of rain that passed through the area that is not considered in the model bottom-up emission estimates, and is not shown here.'*

**L643: Do you mean 'emission factor' rather than 'emission ratio' here?**

We actually mean ratio ($R_{species/FRP}$). As indicated in a previous response, the text has been clarified to better distinguish between ratios and emission ratios/factors. At Line 822, the text has been modified to

*'Total emissions were then estimated by integrating the GOES FRP over the period 2018-06-24 17:00 UTC to 2018-06-25 23:00 UTC (after which no more hot spots were detected by GOES and the fire presumably extinguished), and applying the derived smoldering and flaming $R_{species/FRP}$ values.'*

**L648: Typo: change '2000' to '20:00'**

Changed to 20:48 at Line 820.

**L650: Please state the date and time that the fire 'went out'**

Date and time that GOES did not observe any hot spots (ie. fire went out) is now indicated (July 25 at 23:00 UTC). Text is modified as indicated in next response.

**L654: Add error bars (which will give the actual upper limits) and do not use 5 sig figs.**

Error bars have been added and the sig figs have been fixed. Text revised at Line 825,

*'Assuming that the fire went out when GOES did not observe any hot spots (i.e. after July 25 at 23:00 UTC), total emissions for this fire of CO, NO$_x$ and NH$_3$ are estimated at 22,000±8700, 104±42, and 84±33 tonnes, respectively. If the fire is assumed to have continued burning when GOES did not detect any fire hot spots (between 22:00 - 04:00 UTC and 07:00 - 15:00 UTC, with an FRP of 150 MW (~GOES detection limit; Roberts et al., 2015), the emissions increase to 24,000±9600, 106±43 and 98±39 tonnes, respectively, providing an upper limit of emissions.'*

**L666: Typo: change 'I/SIVOCs' to 'I/SVOCs'**

Corrected at Line 843.

**L1418: Why are both letters and numbers used for superscripts? Just use one. Delete the extra bracket.**

We are now using just numbers for superscripts. We removed the reference to Akagi et al. (2011) and Simpson et al. (2011) as these values were in Andreae (2019) anyway, and added a superscript of 4 for Liu et al. (2017). Extra bracket deleted.

**Figure 2: The distance scale is faint and only includes one label (16.9 km). Please increase the font size in the CO legend**

Fig. 2 has been revised to increase the font size in the CO legend and increase the clarity of the distance scale.

**Figure 9: This Figure is very busy. Can the labeling be simplified?**

Fig 9 has been simplified by removing compound labels that are not explicitly referenced to this figure in the manuscript.

**SI L17: Begin the SI numbering at 1.0 rather than 2.0 (otherwise it seems cut-and-pasted)**

Fixed as suggested.

**SI L99: Please quantify 'as soon as possible.' Hours? Days?**

The analysis was performed between 5 and 9 days after the flight. Text in the SI at Line 114 was modified as follows:

*'The samples were analysed between 5 and 9 days after the flight with an analytical system installed at the Fort McMurray International Airport.'*

**SI L126: Does this mean two cartridge samples were used for emission calculations?**

There were two integrated samples taken in Screen 1, but only 1 sample was used for emission calculations. The text has been removed. More clarity has been added to the main manuscript at Lines 218-231 and Lines 246-247 describing the cartridge samples used for emissions calculations.

**SI L217: Change 'ethyl benzene' to 'ethylbenzene'**

Changed at Line 248.

**Table S1: Replace 'Canister grab samples' with 'AWAS.' Change 'Grab' to a range in seconds.**

Changed to AWAS and indicated 20-30 sec.

**Table S2: Capitalize 'c4h4o2' in the third column (also in Fig. S11)**

Fixed in Table S2 and Fig. S13 (was Fig. S11)

**Table S6: Use subscripts for C8H16, C9H20, C10H14, C10H22. Check the manuscript for this.**

Changed to subscripts in Table S6.

**Table S7: Add error bars to the ERs.**

Lower and upper ranges are added to Table S7 to best reflect the uncertainties in the integrated cartridge measurements, and described at Lines 218-231.

**Table S9: Some molecular weights have as many as 10 significant figures (IVOC, 227.3333333)**

Fixed the sig figures.

**Figure S12: Label the units on each axis with what's plotted ('Hayden et al.' is not an axis label)**

The labels have been modified. The y-axis is now 'Normalized mass species fraction of total organic', and the x-axis is 'SPECIATE4.5 (#95428) normalized mass speciation fraction of total organics'. The text in the figure caption (Fig. S14) has been modified to,

*'Figure S14. Comparison of the normalized organic gas speciation profile derived from this study with that from the EPA's SPECIATE4.5 (#95428) profile. EFs in the present study were mapped to the SAPRC-11 model mechanism species and normalized to total organic gas (which does not include the unidentified mass fraction), to create a total organic gas mass speciation profile. The total organic mass speciation profile is plotted against the similarly treated mass speciation profile from the EPA SPECIATEv4.5 #95428 for wildfire smoldering emissions. Note that for comparison purposes the non-standard SAPRC-11 species in the present study are lumped, such that SESQ is summed with TERP, and IVOC, WSOC and NVOL are summed with NROG.'*